# Anthraquinones and Their Analogues from Marine-Derived Fungi: Chemistry and Biological Activities

**DOI:** 10.3390/md20080474

**Published:** 2022-07-25

**Authors:** Salar Hafez Ghoran, Fatemeh Taktaz, Seyed Abdulmajid Ayatollahi, Anake Kijjoa

**Affiliations:** 1Phytochemistry Research Center, Shahid Beheshti University of Medical Sciences, Tehran 16666-63111, Iran; s_hafezghoran@yahoo.com (S.H.G.); majid_ayatollahi@sbmu.ac.ir (S.A.A.); 2Medicinal Plant Breeding & Development Research Institute, University of Kurdistan, Sanandaj 66177-15175, Iran; 3Department of Advanced Medical and Surgical Sciences, University of Campania “Luigi Vanvitelli”, 80138 Naples, Italy; f.taktaz@gmail.com; 4Department of Biology, Faculty of Sciences, University of Hakim Sabzevari, Sabzevar 96179-76487, Iran; 5ICBAS-Instituto de Ciências Biomédicas Abel Salazar and CIIMAR, Rua de Jorge Viterbo Ferreira 228, 4050-313 Porto, Portugal

**Keywords:** anthraquinones, hydroanthraquinones, bianthraquinones, marine-derived fungi, *Aspergillus* sp., *Penicillium* sp., antibacterial activity, cytotoxicity

## Abstract

Anthraquinones are an interesting chemical class of polyketides since they not only exhibit a myriad of biological activities but also contribute to managing ecological roles. In this review article, we provide a current knowledge on the anthraquinoids reported from marine-derived fungi, isolated from various resources in both shallow waters such as mangrove plants and sediments of the mangrove habitat, coral reef, algae, sponges, and deep sea. This review also tentatively categorizes anthraquinone metabolites from the simplest to the most complicated scaffolds such as conjugated xanthone–anthraquinone derivatives and bianthraquinones, which have been isolated from marine-derived fungi*,* especially from the genera *Apergillus*, *Penicillium*, *Eurotium*, *Altenaria*, *Fusarium*, *Stemphylium*, *Trichoderma*, *Acremonium*, and other fungal strains. The present review, covering a range from 2000 to 2021, was elaborated through a comprehensive literature search using the following databases: ACS publications, Elsevier, Taylor and Francis, Wiley Online Library, MDPI, Springer, and Thieme. Thereupon, we have summarized and categorized 296 anthraquinones and their derivatives, some of which showed a variety of biological properties such as enzyme inhibition, antibacterial, antifungal, antiviral, antitubercular (against *Mycobacterium tuberculosis*), cytotoxic, anti-inflammatory, antifouling, and antioxidant activities. In addition, proposed biogenetic pathways of some anthraquinone derivatives are also discussed.

## 1. Introduction

Phylogenetically and functionally, fungi are ubiquitous organisms living in associations with almost all viable resources such as plants and animals to complement the nutrient cycling in various ecosystems on Earth [1]. Currently, marine mycology has been somewhat neglected, and most of the fungal diversity normally refers to their terrestrial counterparts. To this respect, although ca. 75% of the Earth’s surface is occupied by seas and oceans, harboring small organisms to the largest ones, there are still only about 1000 fungal species that are derived from terrestrial ancestors [2]. Marine-derived fungi are found in diverse habitats and have significant ecological functions. According to Kohlmeyer et al., the filamentous fungi in the marine environment are generally divided into two groups and ecotypes: (i) obligate species, which are originally living in the salt-free waters or estuarine, and (ii) facultative species, which transited from terrestrial and freshwater milieus needing to have a physiological adaptation for survival [3]. Therefore, the marine fungal habitats involve deep-sea sediments, hydrothermal vents, arctic ice and snow, sandy and tidal regions, driftwood, seagrasses, mangroves, and coastal salt marshes. These organisms are more likely to adapt to live on or inside other living organisms such as phytoplanktons, marine mammals, algae, corals, sponges, invertebrates and even dinoflagellates and diatoms (primary producers) to either balance or manage the global carbon cycles [1]. The first report of marine-derived fungi, collected in marine environments, dates back to 19th-century research when the use of microscopes and culture media blossomed [4]. Over the last few decades, the interest in mycochemistry of marine-derived fungi has increased dramatically since the 1990s due to the discovery of bioactive compounds with possible pharmaceutical applications [5]. To date, a large number of fungi that are isolated from the marine ecosystems belong to a few genera including *Aspergillus*, *Penicillium*, *Cladosporium*, *Aureobasidium*, *Cryptococcus*, and *Malassezia*, which were isolated from various environmental niches ranging from the deep sea all the way to surface waters [6]. Furthermore, marine-derived fungi have a versatile biosynthetic machinery capable of biosynthesizing a myriad of secondary metabolites of different chemical classes such as alkaloids, polyketides, meroterpenoids, terpenoids, steroids, and peptides [7,8]. Among fungal secondary metabolites, polyketides are the most structurally diverse and pharmacologically relevant natural products with low toxicity and high efficacy, many of which exhibit cytotoxic effects on cancer cells [9,10]. These include anthraquinones, hydroxyanthraquinones, naphthalenes, naphthoquinones, macrolides, polyenes, tetracyclines, and tropolones that are responsible for a broad range of bioactivities viz. antimicrobial, antifungal, antiviral, antioxidant, anti-inflammatory, anti-fouling, cytotoxicity, inhibition of various enzymes including protein kinases and enzymes related to diabetes [11,12].

By increasing the body of marine-derived fungi literature, this comprehensive review aims to present an update of the previous reviews [11,13], providing the latest classification of all the anthraquinones isolated thus far from the marine-derived fungi. In the following subsections, the review starts with the biosynthesis of anthraquinones. Subsequently, chemistry and some relevant structural features, and species-specific anthraquinones from marine-derived fungi are discussed. The databases used to search for anthraquinone metabolites and the keywords were Google Scholar, PubMed, Scopus, and Web of Science.

## 2. Biosynthesis of Anthraquinone Scaffold

The biosynthesis of anthraquinones in plants is different from that in fungi since there are two distinct pathways in plants, i.e., the shikimate and the acetate–malonate pathways, while the acetate–malonate pathway is a uniquely reported pathway in fungi for the biosynthesis of these polyketides. The biosynthesis of anthraquinones is regulated by non-reducing polyketide synthases (NR-PKSs) comprising the acyl carrier protein (ACP), which provides the regioselective cyclization of a β-polyketide chain to yield various aromatic metabolites. The polyketide pathway providing anthraquinones consists of successive Claisen condensations of malonyl-CoA units (extender units) with acetyl-CoA (a starter unit), leading to the β-ketoacyl-*S-*ACP intermediate as the product template (PT) domain. Depending on the regioselectivity of the cyclization of the first ring and the size of the final product, PT undergoes either C4-C9 or C6-C11 cyclizations, followed by aldol reaction, enolization, oxidation, and decarboxylation, resulting in the formation of the anthraquinone scaffold (Figure 1) [14,15].

By using suitable cultivation methods, many fungal species have been isolated from the submerged areas such as sea water, sediments, sponges, algae, mangrove plants, etc., that are routinely producing anthraquinone compounds. Some of the predominant fungal strains viz. *Aspergillus* sp., *Penicillium* sp., *Eurotium* sp., *Fusarium* sp., and *Alternaira* sp. have been reported in aquatic ecosystems [6]. To a lesser extent, other species that are able to biosynthesize biologically active anthraquinones such as *Acremonium* sp., *Amorosia* sp., *Chaetomium* sp., *Cladosporium* sp., *Guignardia* sp., *Curvularia* sp., *Emericella* sp., *Engyodonitum* sp., *Geotrichum* sp., *Gliocladium* sp., *Halorosellinia* sp., *Microsphaeropsis* sp., *Microsporum* sp., *Monodictys* sp., *Neosartorya* sp., *Nigrospora* sp., *Paecilomyces* sp., *Phoma* sp., *Phomopsis* sp., *Scopulariopsis* sp., *Sporendonema* sp., *Stemphylium* sp., *Talaromyces* sp., *Thermomyces* sp., *Trichoderma* sp., and *Xylaria* sp. have also been isolated from the marine environments ranging from decayed plants to living macro-organisms. In the following sections, we have tentatively categorized marine fungal anthraquinones from the simplest to the most complex structures.

## 3. Anthraquinoid Polyketides and Their Analogues from Marine-Derived Fungi

A comprehensive literature survey of anthraquinoid polyketides covering the period from 2000–2021 was undertaken. In order to facilitate the discussion of the reported anthraquinones from marine-derived fungi, they are classified according to the complexity of the substituents on the anthracene-9,10-dione scaffold as follows: anthraquinones (**I**), tetrahydroanthraquinones (**II**), 5,8-anthraquinones (**III**), tetrahydro-5,8-anthraquinones (**IV**), anthrones (**V**), tetrahydro-9-hydroxyanthrones (**VI**), anthrols (**VII**), 9,10-dihydroxyanthracenes (**VIII**), azaanthraquinones (**IX**) (Figure 2), dimeric anthraquinones, and anthraquinone analogues fused with xanthone and chromone derivatives.

### 3.1. Anthraquinones

#### 3.1.1. Simple Anthraquinones

In general, anthraquinones (9,10-dioxoanthracene or anthracene-9,10-dione) represent the type of pigments possessing a *p*-quinone moiety as a central ring of the anthracene scaffold. Replacing each hydrogen atom of the benzene rings with simple substituents such as hydroxyl, methoxy, methyl or its oxidation analogs (hydroxymethyl, formyl and carboxyl groups), prenyl group, and other substituents leads to diverse anthraquinoid compounds [11].

Buttachon et al. reported the isolation of emodin (**1**) (Figure 3) from the culture extract of *Aspergillus candidus* KUFA0062, isolated from a marine sponge *Epipolasis* sp., which was obtained from the coral reef at the Similan Island National Park, Phang-Nga province, Thailand [16]. Compound **1** was also obtained from the ethyl acetate (EtOAc) extract of the culture of *Penicillium ochrochloron*, isolated from the underwater sea sand, which was collected from the North Sea in St. Peter-Ording, Germany [17]. In another study, Wang et al. isolated **1** and questin (**2**) (Figure 3) from a solid culture extract of *A. flavipes* HN4-13, obtained from a Lianyungang coastal sediment from Jiangsu Province, China [18]. Liu et al. also isolated **1** from the culture extract of a marine-derived *Aspergillus* sp. LS57, isolated from a marine sponge *Haliclona* sp., which was collected at Lingshui, Hainan Province, China [19]. The mycelial extract of *A. terreus* DTO 403-C9, isolated from the leaves of an unidentified mangrove tree, which was collected at Khanh Hoa Province, Vietnam, furnished **2** and a new naturally occurring 1,2,5-trihydroxy-7-methyl-9,10-anthraquinone (**3**) (Figure 3) [20].

A DPPH^•^ radical scavenging activity-guided fractionation of the culture extract of *A. europaeus* WZXY-SX-4-1, isolated from a marine sponge *Xestospongia testudinaria*, which was collected on Weizhou Island, Guangxi Province, China, resulted in the isolation of 1-methyl emodin (**4**) and dermolutein (**5**) (Figure 3) [21]. The culture extract of *A. glaucus* HB1-19, isolated from a marine sediment collected in Fujian Province, China, furnished **1** and **2**, together with physcion (**6**), catenarin (**7**), and rubrocristin (**8**) (Figure 3) [22]. Compound **6** is a common fungal anthraquinone since it was obtained from several sources such as the culture extract of *A. wentii* EN-48, isolated from a marine brown alga *Sargassum* sp. [23], the EtOAc extract of the culture of *Eurotium repens*, isolated from a marine sponge *Suberites domuncula*, collected near Zelenyi Island (Kuril Islands) [24], the fermentation extract of *E. cristatum,* isolated from a marine sponge *Mycale* sp., which was collected from Wonnapa Beach, Bangsaen, Chonburi Province, Thailand [25], the culture broth extract of *Microsporum* sp. MFS-YL, isolated from a marine red alga *Lomentaria catenata*, which was collected from Guryongpo, NamGu, PoHang, Republic of Korea [26], and the fermentation extract of *Penicillium* sp. ZZ901, isolated from a sample of a wild bivalve *Scapharca broughtonii* (Schrenck), which was collected from the Sea Shoal, China [27].

The culture extract of *Aspergillus tritici* SP2-8-1, isolated from a soft coral, *Galaxea fascicularis*, which was collected at Port Dickson, Malaysia, yielded besides **1**, 3-hydroxy-1,2,5,6-tetramethoxyanthracene-9,10-dione (**9**), 3-hydroxy-2-hydroxymethyl-1-methoxyanthracene-9,10-dione (**10**) and 1,2,3-trimethoxy-7-hydroxymethylanthracene-9,10-dione (**11**) (Figure 3) [28].

The culture extracts of the algicolous fungi, viz. *Aspergillus wentii* (pt-1), *A. ustus* (cf-42), and *A. versicolor* (dl-29 and pt-20), possessing algicidal property, afforded 1,5-dihydroxy-3-methoxy-7-methylanthraquinone (**12**), 1,3,5-trihydroxy-7-methylanthraquinone (**13**), and 5-hydroxy-2,4-dimethoxy-7-methylanthraquinone (emodin-6,8-dimethyl ether; **14**) (Figure 3) [29]. Compound **14** was also reported from the culture extract of the strain *A. wentii* EN-48, isolated from a marine macroalga *Sargassum* sp. [30].

The culture extract of *Eurotium chevalieri* KUFA0006, isolated from the inner twig of a mangrove plant, *Rhizophora mucronata* Poir, which was collected in the Eastern Seaboard of Thailand, yielded **1**, **2**, **6** and questinol (**15**) (Figure 3) [31]. Fractionation of the culture extract of an algicolous fungus, *Chaetomium globosum*, isolated from the inner tissue of a marine red alga, *Polysiphonia urceolata*, which was collected from the Qingdao coastline, resulted in the isolation of **7** and erythroglaucin (**16**) (Figure 3) [32].

Fallacinol (**17**) (Figure 3) was isolated, together with **1** and **15** (Figure 3), from the fermentation extract of *Talaromyces stipitatus* KUFA 0207, obtained from a marine sponge *Stylissa flabelliformis*, which was collected at a depth of 10–15 m from the coral reef at Samaesarn Island in the Gulf of Thailand [33].

The culture extract of *Aspergillus versicolor*, isolated from the inner tissue of a green alga, *Halimeda opuntia*, which was collected at a depth of 5–8 m from the coast of Rass Mohamed of Red Sea (South Sinai, Egypt), furnished evariquinone (**18**) and 7-hydroxyemodin-6,8-dimethyl ether (**19**) (Figure 3), in addition to **1** and **4** (Figure 3) [34].

2-(Dimethoxymethyl)-1-hydroxyanthracene-9,10-dione (**20**), 1-hydroxy-2-methylanthracene-9,10-dione (**21**), 2-methylanthracene-9,10-dione (**22**), damnacanthal (**23**), rubiadin (**24**), xanthopurpurin (**25**), rubianthraquinone (**26**) and 6-hydroxyrubiadin (**27**) (Figure 4) were isolated, together with **1** (Figure 3), from the fermentation extract of *A. versicolor*, obtained from a deep-sea sediment [35].

Further anthraquinones viz. citreorosein (**28**), chrysophanol (**29**) and aloe-emodin (**30**) (Figure 4) were isolated, together with **1** (Figure 3), from the mycelial extract of *Penicillium oxalicum* 2HL-M-6, which was obtained from a sea mud in Bohai Bay, China [36]. Khamthong et al. reported the isolation of **1** (Figure 3), **28** and **29** (Figure 4) from the fermentation extract of *P. citrinum* PSU-F51, which was obtained from a gorgonian Sea fan (*Annella* sp.), collected at the Similan Islands, Phangnga Province, Thailand [37].

Ren et al., in the screening for cytotoxic agents against human myeloid leukemia K562 cell line, described the isolation of **1** and **28** from the culture extract of *Gliocladium* sp. T31, which was isolated from a marine lichen collected from the South Pole [38]. Compounds **1** (Figure 3), **28**, and **29** (Figure 4) were reported from the culture broth extract of a gorgonian coral-derived *Penicillium* sp. SCSGAF0023 [39], whereas **28** and **29** were also reported from the fermentation extract of *Fusarium equiseti*, isolated from a brown alga, *Padina pavonica,* collected from the Red Sea [40].

Chemical investigation of the fermentation extract of an endophytic fungus, *Penicillium citrinum* HL-5126, isolated from the mangrove plant, *Bruguiera sexangula* var. *rhynchopetala*, which was collected in the South China Sea, resulted in the isolation of **2** (Figure 3) and **28** (Figure 4) [41]. Compounds **1** (Figure 3) and **29** (Figure 4) were also reported from the EtOAc extract of the mycelial extract of *Paecilomyces* sp. (Tree1-7), isolated from a mangrove saprophytic bark from the Taiwan Strait [42], and from the culture extract of *Aspergillus candidus* KUFA0062, isolated from a marine sponge, *Epipolasis* sp., which was obtained from the coral reef at the Similan Island National Park, Phang-Nga province, Thailand [16].

Purification of the culture extract of *Trichoderma* sp. (H-1), isolated from a surface muscle of a sea cucumber, which was collected from Chengshantou Island, Yellow Sea, China, also afforded **1** (Figure 3) and **29** (Figure 4) [43]. Compound **29** (Figure 4) was also reported from the culture extract of an unidentified marine red alga-derived fungus (strain F-F-3C), which was collected at the coast of Tarama Island, Okinawa, Japan [44].

The chloroform-soluble portion of the methanol (MeOH) extract of a culture of *Penicillium* sp. strain F01V25, isolated from a marine alga, *Dictyosphaeria versluyii*, collected near Dravuni, Fiji, yielded carviolin (**31**) (Figure 4) [45]. Purification of a MeOH-soluble extract of *Penicillium* sp. SCSIOsof101, isolated from sediment samples, collected in the South China Sea (2448 m depth), also afforded an emodin derivative, named emodic acid (**32**) (Figure 4) [46]. Compounds **1** (Figure 3) and **32** (Figure 4) were also isolated from the culture extract of *Eurotium rubrum*, which was obtained from the inner tissue of a semi-mangrove plant, *Hibiscus tiliaceus,* collected from Hainan island, China [47].

Macrosporin (**33**), 1,7,8-trihydroxy-3-methoxy-6-methylanthraquinone (**34**), and 1-hydroxy-3-methoxy-6-methylanthraquinone (**35**) (Figure 4) were isolated from the mycelial extract of *Penicillium* sp., obtained from a soft coral, *Sarcophyton tortuosum,* which was collected in the South China Sea [48]. Compound **33** was also reported from the culture extract of a marine-derived fungus, *Altenaria* sp. ZJ-2008003, which was isolated from a soft coral, *Sarcophyton* sp., collected from the South China Sea [49], as well as from the solid-rice culture extract of an endophytic fungus, *Stemphylium* sp. 33231, isolated from a mangrove plant, *Burguiera sexangula* var. *rhynchopetala*, which was collected in the South China Sea [50], as well as from the solid-rice culture extract of *S. lycopersici*, isolated from the inner tissue of a gorgonian soft coral, *Dichotella gammacea*, collected from the South China Sea [51]. Compounds **33** and **35** (Figure 4) were also isolated from the fermentation extract of *Phomopsis* sp. PSU-MA214, which was obtained from the leaves of a mangrove plant, *Rhizophora apiculata* Griff. Ex T. Anderson, collected from Songkhla province, Thailand [52].

Chemical investigation of *Eurotium chevalieri* MUT2316, isolated from the Atlantic sponge, *Grantia compressa*, afforded cinnalutein (**36**) (Figure 4), together with **6** (Figure 3) [53], while the culture extract of *E. chevalieri* KUFA0006, isolated from the inner twig of a mangrove plant, *Rhizophora mucronata* Poir, which was collected in the Eastern Seaboard of Thailand, yielded acetylquestinol (**37**) (Figure 4), in addition to **1**, **2**, **6** and **15** (Figure 3) [31]. Compound **37** was also obtained from the culture extract of *Neosartorya spinosa* KUFA 1047, isolated from a marine sponge, *Mycale* sp., which was collected from a coral reef at Samae San Island, Chonburi province, Thailand [54].

Pachybasin (**38**), phomarin (**39**), 1-hydroxy-3-hydroxymethylanthraquinone (**40**), and ω-hydroxydigitoemodin (**41**) (Figure 4) were isolated, together with **1** (Figure 3) and **29** (Figure 4), from the fermentation extract of *Trichoderma harzianum* (XS-20090075), isolated from the inner tissue of a soft coral, which was collected from the coral reef at Xisha Island in the South China Sea [55]. A defatted culture extract of *Trichoderma* sp. strain SCSIO41004, isolated from a marine sponge, *Callyspongia* sp., which was collected from the sea area near Xuwen County, Guangdong province, China, furnished 1,3,6-trihydroxy-8-methylanthraquinone (**42**) (Figure 4) [56].

She et al. described the isolation of 1,4-dihydroxy-2-methoxy-7-methylanthracene-9,10-dione (**43**) (Figure 4) from the culture extract of an estuarine fungus, *Halorosellinia* sp. (no. 1403). The structure of **43** was confirmed by a single-crystal X-ray diffraction analysis [57].

Mycelial and broth extracts of a mangrove endophytic fungus, *Halorosellinia* sp. (no. 1403), isolated from a decayed woody tissue of a mangrove tree, *Kandelia candel* (L.) Druce., which was collected from Mai Po, Hong Kong, yielded 1,4,6-trihydroxy-2-methoxy-7-methylanthracene-9,10-dione (**44**), demethoxyaustrocortirubin (**45**), 1-hydroxy-3-methyl-9,10-anthraquinone (**46**), and austrocortinin (**47**) (Figure 5) [58]. Compound **47** (Figure 5) was also isolated from the broth culture extract of *Fusarium* sp. PSU-F14, isolated from a gorgonian sea fan, which was collected near Koh Hin Ran Pet, Suratthani Province, Thailand [59].

El-Beih et al., reported the isolation of monodictyquinone A (**48**) (Figure 5), together with **1** (Figure 3), **29** and **38** (Figure 4), from the EtOAc-soluble fraction of the ethanol (EtOH) extract of *Monodictys* sp., which was isolated from a sea urchin, *Anthocidaris crassispina*, collected from Toyama Bay in the Sea of Japan [60].

Rheoemodin (**49**) (Figure 5) was isolated, together with **1**, **15**, **17** (Figure 3) and **28** (Figure 4), from the fermentation extract of *Talaromyces stipitatus* KUFA 0207, which was obtained from a marine sponge, *Stylissa flabelliformis*, collected from the coral reef at Samaesarn Island in the Gulf of Thailand [33].

Marcrospin (**50**) (Figure 5) and **6** (Figure 3) were isolated from the mycelial extract of *Altenaria* sp. ZJ9-6B, isolated from fruits of a mangrove tree, *Aegiceras corniculatum*, collected in Zhanjiang mangrove, Guangdong province, China [61]. Chemical investigation of the broth culture extract of *Altenaria* sp. (SK11), isolated from the root of a mangrove tree, *Excoecaria agallocha*, from Shankou, Guangxi province, China, yielded 6-methylquinizarin (**51**) (Figure 5), together with **47** [62]. 6-*O*-Methylalaternin (**52**) (Figure 5) was obtained from a culture extract of *Altenaria tenuissima* DFFSCS013, which was isolated from a sediment collected at a depth of 2403 m from the South China Sea [63].

Jadulco et al. reported the isolation of lunatin (**53**) (Figure 5) from the culture extract of *Curvularia lunata*, which was isolated from a marine sponge, *Niphates olemda*, collected in the Bali Bata National Park in Indonesia [64]. By using bioctivity-guided purification approach, Ren et al. also isolated **1** (Figure 3), **28** (Figure 4), and **53** (Figure 5) from the fermentation extract of *Gliocladium catenulatum* T31, isolated from marine sediment samples [65].

1,3-Dihydroxy-6-hydroxymethyl-7-methoxyanthraquinone (**54**) and 1,3-dihydroxy-6-methyl-7-methoxyanthraquinone (**55**) (Figure 5) were isolated from a defatted culture extract of *Thermomyces lanuginosus* Tsikl KMM 4681, obtained from a marine sediment from the South China Sea, Vietnam [66], while 7-methoxymacrosporin (**56**) and 7-(γ,γ)-dimethylallyloxymacrosporin (**57**) (Figure 5) were isolated from the culture extract of an endophytic fungus, *Phoma* sp. L28, obtained from the roots of a mangrove plant, *Myoporum bontioides* A. Gray, which was collected in Leizhou peninsula, Guangdong province, China [67].

3,5,8-Trihydroxy-7-methoxy-2-methylanthracene-9,10-dione (**58**) and **47** (Figure 5) were obtained from the culture extract of *Nigrospora* sp. ZJ-2010006, isolated from an unidentified sea anemone, which was collected from the Weizhou coral reef in the South China Sea. In order to evaluate the antibacterial activity of their analogs, **47** and **58** were acetylated to give a series of acetylated anthraquinones viz. 8-acetoxyaustrocortirubin (**47a**), 8-acetoxy-3,5-dihydroxy-7-methoxy-2-methylanthracene-9,10-dione (**58a**), 5-acetoxy-3,8-dihydroxy-7-methoxy-2-methylanthracene-9,10-dione (**58b**), 3-acetoxy-5,8-dihydroxy-7-methoxy-2-methylanthracene-9,10-dione (**58c**), 5,8-diacetoxy-3-hydroxy-7-methoxy-2-methylanthracene-9,10-dione (**58d**), 3,8-diacetoxy-5-hydroxy-7-methoxy-2-methylanthracene-9,10-dione (**58e**), 3,5-diacetoxy-8-hydroxy-7-methoxy-2-methylanthracene-9,10-dione (**58f**), and 3,5,8-triacetoxy-7-methoxy-2-methylanthracene-9,10-dione (**58g**) (Figure 5) [68]. The fermentation extract of the same fungus, isolated from the inner tissue of the zoathid, *Palythoa haddoni* (GX-WZ-20100026), collected from coral reefs in the South China Sea, also furnished both **47** and **58** (Figure 5) [69].

1,6,8-Trihydroxy-4-benzoyloxy-3-methylanthraquinone (**59**) (Figure 5) was isolated, together with **2**, **6** and **7** (Figure 3), from the culture extract of *Eurotium* sp. SCSIO F452, obtained from sediment samples collected from the South China Sea [70].

Brauers et al. reported the isolation of three 1,3,6,8-tetrahydroxyanthraquinone analogues, **60**–**62** (Figure 6), from the culture extract of *Microsphaeropsis* sp., obtained from fresh samples of a marine sponge, *Aplysina aerophoba*, which was collected from Banyuls-sur-Mer in Southern France. The absolute configuration of the stereogenic carbon of the substituent on C-2 of **60**–**62** was established as *R* by comparison of their calculated and experimental electronic circular dichroism (ECD) spectra [71,72]. Fractionation of a defatted culture extract of a marine sponge-associated fungus, *Trichoderma* sp. strain SCSIO41004, led to the isolation of 7-acetyl-1,3,6-trihydroxyanthracene-9,10-dione (**63**) and ZSU-H85 (**64**) (Figure 6) [56].

Zhao et al., in their screening program to search for metabolites with anti-phytopathogenic bacterial and fungal activities, as well as cytotoxicity, found that the culture extract of *Fusarium equiseti*, isolated from the intertidal marine plants of the Yellow Sea in Qingdao, China, showed interesting bioactivities. Further fractionation of the culture extract led to the isolation of **63** and (11*S*)-1,3,6-trihydroxy-7-(1-hydroxyethyl) anthracene-9,10-dione (**65**) (Figure 6). The absolute configuration of the stereogenic center (C-11) in **65** was determined as *S* by comparison of its calculated and experimental ECD spectra [73]. Compound **65** (Figure 6) was also isolated from the culture extract of *Cladosporium* sp. HNWSW-1, isolated from fresh roots of a mangrove plant, *Ceriops tagal*, which was collected from Dong Zhai Gang Mangrove Reserve in Hainan province, China [74].

The mycelial extract of *Fusarium* sp. (strain no. b77), obtained from the Shenzhan coast, Guangzhou, China, provided 5-acetyl-2-methoxy-1,4,6-trihydroxyanthraquinone (**66**) and 1-acetoxy-5-acetyl-2-methoxy-4,6-trihydroxyanthraquinone (**67**) (Figure 6) [75], while isorhodoptilometrin (**68**) (Figure 6) was isolated from the mycelial extract of a sea mud-derived *Penicillium oxalicum* 2HL-M-6 [36]. Ren et al. isolated **68** from the active extract of a marine lichen-derived *Gliocladium* sp. T31 [38]. Compound **68** was also obtained from the fermentation extract of a sea sediment-derived *G. catenulatum* T31, using antitumor activity-guided purification approach [65].

The isorhodopilometrin derivative, (−)-2′*R*-1-hydroxyisorhodopilometrin (**69**) (Figure 6), was obtained from the culture extract of *Penicillium* sp. OUCMDZ-4736, isolated from a sediment surrounding the roots of a mangrove plant, *Acanthus ilicifolius*, collected at Wenchang, Hainan Province, China. The absolute configuration of the sterogenic carbon in **69** was determined as 12*R*, based on a comparison of the experimental $({\left[\alpha \right]}_{D}^{25}$ − 56.0) and calculated optical rotation values, which was in contrast with that of (+)-2′*S*-1-hydroxyisorhodopilometrin (${\left[\alpha \right]}_{D}^{24}$ + 30) [76]. Isorhodoptilometrin-1-methyl ether (**70**) (Figure 6) was isolated from the EtOAc extract of a culture of an algicolous fungus, *A. versicolor* [34].

(+)-2′*S*-Isorhodoptilometrin (**71**) (Figure 6) was isolated, together with **1** (Figure 3), **29**, and **38–41** (Figure 4), from the fermentation extract of a soft coral-associated *Trichoderma harzianum* (XS-20090075) [55]. Nalgiovensin (**72**) (Figure 6), an anthraquinone with a 2′-hydroxypropyl substituent on C-5, was reported from a defatted EtOAc extract of the culture of the asexual morph of a marine alga-associated *A. alliaceus* (teleomorph: *Petromyces alliaceus*). The absolute configuration of the hydroxyl-bearing stereogenic carbon of the side chain was determined as 2′*S* by X ray crystallogaraphic analysis [77].

1-Methyl ether of nalgiovensin (**73**) (Figure 6) was also isolated, together with **14**, **19** (Figure 3) and **28** (Figure 4), from the MeOH fraction of the mycelial extract of a deep-sea-derived fungus, *Emericella* sp. SCSIO 05240, which was isolated from sediment samples collected from the South China Sea at a depth of 3258 m [78].

Chemical investigation of the culture extract of a marine sponge-associated fungus, *Neosartorya spinosa* KUFA 1047, led to the isolation of two alkylated anthraquinones, penipurdin A (**74**) and acetylpenipurdin A (**75**) (Figure 6). The absolute configuration of C-2′ in **75** was suggested to be the same as that of **74**, i.e., 2′*S*, on the basis of the biogenic consideration [54].

1,3,6-Trihydroxy-7-(dihydroxypropyl)-anthraquinone (**76**) (Figure 6) was isolated from a defatted culture extract of a marine sediment-derived fungus, *Thermomyces lanuginosus* Tsikl KMM 4681. The relative configurations of C-15 and C-16 of a diol side chain in **78** were determined by the observed correlations in the NOESY (Nuclear Overhauser Effect Spectroscopy) spectrum and the value of the coupling constant of the vicinal protons, as well as the presence of magnetically non-equivalent methyl groups of its acetonide (**76a**) (Figure 6). The absolute configurations of C-15 and C-16 in **76** and **76a** were established as 15*R*,16*S* by comparison of their calculated and experimental ECD spectra [66].

6,8-Dimethoxy-1-methyl-2-(3-oxobutyl)-anthrakunthone (**77**) (Figure 6) was isolated from the culture extract of a marine mangrove endophytic fungus, *Fusarium* sp. ZZF60, from the South China Sea [79], whereas norsolorinic acid (**78**) (Figure 6), a tetrahydroxyanthraquinone with a hexanoyl substituent on C-2, was purified by ethanol stress strategy from a combination of EtOAc and acetone/water extracts of the culture of *Aspergillus nidulans* MA-143, isolated from fresh leaves of a mangrove plant, *Rhizophora stylosa* [80].

The acetone/EtOAc extract of mycelia of *A. puniceus* SCSIO z021, isolated from a deep-sea sediment, which was collected from Okinawa Trough (1589 m depth), afforded the undescribed anthraquinones, 8-*O*-methyl versiconol (**79**), 2′,3′-dihydorxy versiconol (**80**), and the previously reported methyl averantin (**81**) and versiconol (**82**) (Figure 7). The stereogenic carbon (C-2′) of the 1,4-dihydroxy-butan-2-yl substituent of **79** was determined as 2′*S* based on the highly similarity of the cotton effects (CEs) at 388, 314, and 235 nm, as well as of its ECD spectrum to those of aspergilol I. However, the absolute configurations of the stereogenic carbons, C-2′ and C-3′, of the 1′,2′,3′,4′-tetrahydroxybutan-2-yl substituent in **80** remained unassigned [81].

The undescribed 6,8-di-*O*-methylaverantin (**83**) and the previously reported 6,8-di-*O*-methylversiconol (**84**) (Figure 7) were obtained from the combined MeOH and EtOAc extracts of *A. versicolor* EN-7, isolated from a brown alga, *Saragassum thunbergii*, which was collected from the Qingdao coastline of Shandong Province, China. The absolute configuration of the stereogenic carbon of the side chain of **83** (C-1′) was determined as *S* by comparison of its optical rotation (${\left[\alpha \right]}_{D}^{20}\;$= 92.2) with that of (−)-averantin (${\left[\alpha \right]}_{D}^{22}$ = −138°) [82]. Compound **84** was also isolated from the culture extract of a mangrove endophytic fungus, ZSUH-36, isolated from the Shenzhen mangrove, *Acanthus ilicifolius* Linn. [83].

Averantin (**85**) was isolated, together with **81** and **82** (Figure 7), from the culture extract of *A. versicolor*, isolated from a marine sponge, *Petrosia* sp., which was collected at the depth of 20 m at Jeju Island, Korea [84]. Compounds **82** and **85** (Figure 6) were also isolated from a culture broth of a marine-derived *Penicillium flavidorsum* SHK1-27 by bioassay-guided isolation approach [85].

6,8,1′-Tri-*O*-methylaverantin (**86**) (Figure 7) was isolated, together with **81**, from the mycelial extract of a mangrove endophytic fungal strain ZSUH-36, which was isolated from the Shenzhen mangrove, *Acanthus illicifolius* Linn. [86]. Compound **86** was also obtained from the culture extract of a marine-derived fungus, *Aspergillus* sp. SF-6796, isolated from a marine organism collected from the Ross Sea, Antarctica [87].

Averythrin (**87**) (Figure 7) was obtained, together with **80** and **85**, from the culture broth extract of *A. versicolor* INF 16–17, isolated from the inner tissue of an unidentified marine clam [88]. Compounds **81**, **85**, and **87** (Figure 7) were also obtained from the culture extract of *A. versicolor* A-21-2-7, isolated from a deep-sea sediment from the South China Sea [89]. Compound **87** was also obtained from the fermentation extract of a mangrove endophytic fungus, *Aspergillus* sp. 16-5C, which was isolated from the leaves of a mangrove tree, *Sonneratia apetala*, collected at Hainan Island, China [90].

The combined acetone and EtOAc culture extracts of *Aspergillus* sp. SCSIO F063, isolated from a deep-sea sediment from the South China Sea, furnished (1′*S*)-6,1′-*O*,*O*-dimethylaverantin (**88**), (*S*)-(−)-averantin (**89**), 6-*O*-methylaverantin (**90**), and averantin-1′-butyl ether (**91**) (Figure 7), in addition to **81** and **87** (Figure 7). The absolute configuration of the stereogenic center at C-1′ in **88** was assigned as *S* based on its negative value of rotation $({\left[\alpha \right]}_{D}^{25}$ = −140°), which was the same as that of the previously described **89** (${\left[\alpha \right]}_{D}^{25}$ = −138°) and **90** [91].

Aspergilol I (**92**), SC3-22-3 (**93**), and coccoquinone A (**94**) (Figure 7) were isolated, together with **81** and **82**, from the culture broth extract of *A. versicolor* SCSIO-41502, which was obtained from marine sediment samples collected from the South China Sea. The absolute configuration of C-16 in **92** was determined as *S* by comparison of its circular dichroism (CD) spectrum with that of the previously described (1′*S*)-7-chloroaverantin, while the absolute configuration of C-19 was established by the modified Mosher’s method. Moreover, the absolute configuration of C-16 in **93** and **94** was also determined as *S* by comparison of their CD spectra and optical rotations (${\left[\alpha \right]}_{D}^{25}$ − 30.6° for **93** and −11.1° for **94**) with those of (1′*S*)-7-chloroaverantin [92].

Versiconol B (**95**) (Figure 7) was isolated, together with **81**, from the culture extract of *Aspergillus* sp. F40, isolated from a marine sponge, *Callyspongia* sp., which was collected from the sea area near Xuwen County, Guangdong Province, China. The absolute configuration of a stereogenic carbon (C-1′) in **95** was established as *S* by comparison of its optical rotation (${\left[\alpha \right]}_{D}^{25}\;$− 38.6°) with that of **82** (${\left[\alpha \right]}_{D}^{25\;}$− 101.5°) [93].

The culture extract of a marine sponge-associated fungus, *A. europaeus* WZXY-SX-4-1, furnished (+)-1-*O*-demethylvariecolorquinone A (**96**) and (+)-variecolorquinone A (**97**) (Figure 8) [21]. The NMR data of **96** were identical to those of the previously described (2*S*)-2,3-dihydroxypropyl-1,6,8-trihydroxy-3-methyl-9,10-dioxoanthracene-2-carboxylate, a demethylated analogue of variecolorquinone A. Since the specific rotation of **96** was dextrorotatory (${\left[\alpha \right]}_{D}^{22}$ + 25° in MeOH), while that of the previously reported (2*S*)-2,3-dihydroxypropyl-1,6,8-trihydroxy-3-methyl-9,10-dioxoanthracene-2-carboxylate was levorotatory (${\left[\alpha \right]}_{D}^{22}$ − 23° in MeOH) [94], the absolute configuration 2′*R* was assigned for **96** [21]. The same authors also reported the isolation of **97** from the culture extract of *A. glaucus* HB1-19, isolated from a marine sediment collected in Fujian Province, China. Similar to **96**, the specific rotation of **97** was also dextrorotatory (${\left[\alpha \right]}_{D}^{20}$ + 16.8°), which is opposite to that of variecolorquinone A (${\left[\alpha \right]}_{D}^{20}$ = −18.0°); thus, the absolute configuration of C-2′ of **97** was assigned as *R* [22]. Compound **97** was also reported from the fermentation extract of *Eurotium cristatum* EN-220, which was isolated from the marine alga, *Sargassum thunbergii*, collected from the coast of Qingdao, China [95].

Four anthraquinone derivatives, 6-*O*-methylaverufin (**98**), 6,8-di-*O*-methylaverufin (**99**), aversin (**100**) and 8-*O*-methylversicolorin A (**101**) (Figure 9), were obtained from a defatted culture extract of *Aspergillus nidulans* MCCC 3A00050, which was isolated from a deep-sea sediment collected from the western Pacific ocean [96]. Compound **99** was also reported from the culture extract of a marine-derived fungus, *Aspergillus* sp. SF-6796 [87], while **100** was reported from the culture extract of a mangrove endophytic fungus ZSUH-36 [83] and from a fermentation extract of *A. versicolor* MF359, isolated from a marine sponge, *Hymeniacidon perleve*, which was collected from the Bohai Sea, China [97].

The ethanol-stress culture of the mangrove endophytic fungus, *A. nidulans* MA-143, furnished isoversicolorin C (**102**), versicolorin C (**103**), averufin (**104**), paeciloquinone E (**105**), and averufanin (**106**) (Figure 9). The absolute configurations of C-1′ and C-2′ in **102** were established as 1′*S*,2′*R* by comparison of calculated and experimental ECD spectra [80]. Compound **104** was also isolated from the culture extract of a deep-sea sediment-derived *A. versicolor* SCSIO-41502 [92].

Nidurufin (**107**) (Figure 9) was reported, together with **104**, from the mycelial extract of *A. niger* strain MF-16#, isolated from the sea water collected in Quanzhou Gulf, Fujian Province, China [98]. Compounds **104** and **107** were also isolated from the culture extract of a marine sponge-associated fungus, *A. versicolor* [84]. Compound **107** was also isolated from a liquid culture extract of *Penicillium flavidorsum* SHK1-27, obtained from marine sediment samples, collected from Weizhou Island, China [99].

The liquid culture extract of a mangrove endophytic fungal strain (isolate 1850), isolated from a leaf of a mangrove plant, *Kandelia candel*, collected at the estuarine mangrove in Hong Kong, also furnished **103**, **104** and **107** (Figure 9) [100], while the fermentation extract of a deep-sea sediment-derived fungus, *A. puniceus* SCSIO z021, yielded 3′-hydroxy-8-*O*-methyl verscicolorin B (**108**), versicolorin B (**109**), and 8-*O*-methylnidurufin (**110**) (Figure 9), in addition to **104**, **106** and **107 [81]**. The absolute configurations of C-1′, C-2′ and C-3′ in **108** were established as 1′*R*,2′*R*,3′*R* by comparison of its calculated and experimental ECD spectra [81]. Compounds **104** and **109** (Figure 9) were also isolated from the culture extracts of a marine sponge-associated fungus, *Aspergillus* sp. F40 [93], and of *A. versicolor* MF18051, isolated from a sediment collected from Bohai Sea, China [101].

2′-Hydroxyversicolorin B (**111**) and noraverufanin (**112**) (Figure 9) were isolated, together with **104**, **107** and **109**, from the culture extract of a marine sponge-associated fungus, *A. versicolor* SCSIO 41016 [102], whereas **98**–**100**, 6,8-di-*O*-methylnidurufin (**113**) and 6,8-di-*O*-methylversicolorin A (**114**) (Figure 9) were reported from the fermentation extract of an algicolous fungus, *A. versicolor* EN-7 [82]. The culture extract of a deep-sea sediment-derived *A. versicolor* A-21-2-7 furnished UCT1072M1 (**115**) (Figure 9), in addition to **104**, **106**, **107**, and **109** [89], whereas a mangrove endophytic fungus, *Aspergillus* sp. strain 16-5C, yielded asperquinone A (**116**) (Figure 9), along with **99**, **100**, and **113**. The absolute configurations of the stereogenic carbons, C-1′, C-4′, C-5′, in **116** were established as 1′*S*,4′*R*,5′*S* by comparison of its calculated and experimental ECD spectra [90].

The culture extract of *Aspergillus* sp., isolated from the inner part of a fresh tissue of a gorgonian, *Dichotella gemmacea*, which was collected from the South China Sea, furnished 8-*O*-methylaverufin (**117**) and 8-*O*-methylaverufanin (**118**) (Figure 9), in addition to **104**, **106**, **107**, and **110**. The relative configuration of **110** was established by ^1^H-^1^H coupling constants and analysis of NOESY correlations, whereas the absolute configurations of its stereogenic carbons were proposed as 1′*R*,2′*S*,5′*S* on the basis of the biogenic consideration as well as by comparison with those of **107**, whose stereostructure was unambiguously established [103]. Versicolorin A (**119**) (Figure 9), together with **100**, **104**, **107**, **109** and **118**, were isolated from the culture extract of a marine-derived *Penicillium flavidorsum* SHK1-27 by bioassay-guided isolation approach [85].

Insecticidal activity-guided fractionation of a solid-rice culture extract of an endophytic fungus, *Acremonium vitellinum*, isolated from a fresh inner tissue of an unidentified marine red alga, collected from Qingdao, China, led to the isolation of 6,8-di-*O*-methylbipolarin (**120**) (Figure 9), in addition to **99**, **100**, and **113**. The absolute configuration at C-1′ of **120** was established as *S* by comparison of its calculated and experimental ECD spectra [104]. 6,8-Di-*O*-methyl-averufinan (**121**) (Figure 9) was isolated, together with **99**, **103** and **104**, from the mangrove endophytic fungal strain ZSUH-36 [86].

Chemical investigation of a deep-sea sediment-derived fungus, *A. versicolor* SCSIO-41502, resulted in the isolation of four aspergilol analogs, i.e., aspergilols (±)-A (**122**), (±)-B (**123**), (±)-G (**124**), and (±)-H (**125**) (Figure 10). Since **122**–**125** displayed no optical rotation, and because the HPLC (high performance liquid chromatography) analysis with a chiral column showed the presence of two peaks with a ratio of 1:1 for each of them, it was concluded that the compounds were isolated as racemic mixtures. By using HPLC equipped with a CHIRALPAK IA column and *n*-hexane/isopropanol/trifluoroacetic acid (80:20:0.05) as eluent, (±)-**122** was further purified to give pure (+)- and (−)-optical isomers [92].

Two anthraquinone-citrinin derivatives, penicillanthranins A (**126**) and B (**127**) (Figure 10), were obtained from the mycelial extract of a gorgonian-associated fungus, *Penicillium citrinum* PSU-F51. The relative configurations of the stereogenic carbons of a dihydrofuran ring (C-1′, C-3′, C-4′) in **126** were assigned by NOEDIFF (Nuclear Overhauser Enhancement Difference) results. The absolute configurations of the stereogenic carbons in **127** were assumed to be the same as those of **126** since they showed similar optical rotations [37].

Emodacidamides A (**128**), B (**129**), D (**130**), E (**131**), and H (**132**) (Figure 10), anthraquinones with amino acid-containing amide side chains, were obtained from the culture extract of a deep-sea sediment-derived *Penicillium* sp. SCSIOsof101. The absolute configurations of the amino acid residues were determined by Marfey’s method or by a combination of Marfey’s method with chiral-phase HPLC analysis. L-Val was identified as the amino acid in the amide side chain of **128** and **129**, whereas L-Ile was the amino acid of the amide side chain of **130** and **131**, and L-Ala was identified for **132** [46].

Two anthraquinones containing a 1-hydroxy-2(2*R*)-2-(methoxycarbonyl)-5-oxopyrrolidin-1-yl substituent, anthrininones B (**133**) and C (**134**) (Figure 10), were obtained from the culture extract of a deep-sea sediment-derived fungus, *Altenaria tenuissima* DFFSCS013. In order to determine the absolute configurations of C-13 and C-18, the ECD spectra of two diastereomers (13*R*,18*S*)-**133** and (*13*S,18*S*)-**134** were calculated, which also generated the ECD spectra of their enantiomers for (13*S*,18*R*)-**133** and (13*R*,18*R*)-**134**. Comparison of the calculated and experimental ECD spectra of **133** and **134** showed that the mirror imaged-ECD spectra for (13*S*,18*R*)-**133** and (13*R*,18*R*)-**134** and the experimental ECD spectra of **133** and **134** had accordant strong positive CEs near 220 nm, thus confirming the absolute configurations of C-18 in **133** and **134** as *R*. However, because both of the experimental ECD spectra of **133** and **134** showed weak CEs around 250–460 nm, the complete absolute configurations of **133** and **134** could not be accurately determined by ECD calculations. Therefore, the absolute configurations of C-18 in **133** and **134** were determined by ^13^C NMR calculations using density functional theory (DFT) at the mPW1PW91/6-311G(d,P) level. The results strongly suggested that the absolute configurations of C-13 and C-18 in **133** and **134** were 13*S*,18*R* and 13*R*,18*R*, respectively. Compounds **133** and **134** were epimers at C-13 and that the C-18*R* was derived from a cyclization of D-glutamate to form a butylaminolate moiety as shown in Figure 10 [63].

#### 3.1.2. Halogenated Anthraquinones

Although a number of chlorinated anthraquinone derivatives have been isolated, together with non-haloginated anthraquinones, from the culture of marine-derived fungi with normal culture media, the brominated counterparts were only isolated from marine-derived fungi cultured in bromide-enriched media. Eze et al. described the isolation of 7-chloroemodin (**135**) (Figure 11) from the EtOAc extract of the culture of an underwater sea sand-derived *Penicillium ochrochloron* [17], while Luo et al. reported the isolation of 2-chloro-1,3,8-trihydroxy-6-(hydroxymethyl)anthracene-9,10-dione (**136**) (Figure 11) from the culture extract of a sea sediment-derived *Penicillium* sp. SCSIOsof101 [46].

The culture extract of a mangrove endophytic fungus, *P. citrinum* HL-5126, furnished 2′-acetoxy-7-chlorocitreorosein (**137**) (Figure 11) [41], whereas the fermentation extract of *Penicillium* sp. SCSIO sof101, isolated from sediment samples collected in the South China Sea, furnished 7-chloro-1′-hydroxyisorhodoptilometrin (**138**) (Figure 11) [105].

The halogenated derivatives of averantin, including (1′*S*)-7-chloroaverantin (**139**), (1′*S*)-6-*O*-methyl-7-chloroaverantin (**140**), (1′*S*)-1′-*O*-methyl-7-chloroaverantin (**141**), (1′*S*)-6,1′-*O*,*O*-dimethyl-7-chloroaverantin (**142**), (1′*S*)-7-chloroaverantin-1′-butyl ether (**143**), 7-chloroaverythrin (**144**), and 6-*O*-methyl-7-chloroaverythrin (**145**) (Figure 11), were isolated from the organic extract of a sea salt-containing culture of a deep-sea sediment-derived *Aspergillus* sp. SCSIO F063, while (1′*S*)-6,1′-*O*,*O*-dimethyl-7-bromoaverantin (**146**) and (1′*S*)-6-*O*-methyl-7-bromoaverantin (**147**) (Figure 11) were isolated from the fungal mycelia using a sodium bromide-containing culture medium. The absolute configurations of the stereogenic carbon (C-1′) in **139**–**143**, **146** and **147** were established as *S* by comparison of the CD spectra of **139**, **140** and **147** with that of (*S*)-(−)-averantin (**89**) (Figure 7), as well as the same sign of optical rotations of **139**–**147** [91].

Nalgiolaxin (**148**) and 7-chloro versicolorin A (**149**) (Figure 11) were isolated from the culture extract of an algicolous fungus, *A. alliaceus* [77], and the fermentation extract of a deep-sea sediment-derived *A. puniceus* SCSIO z021 [81], respectively. The absolute configurations at C-1′ and C-2′ of the furofuran ring system were established as 1′*R*,2′*S* by comparison of the calculated and experimental ECD spectra of **149** [81].

The chlorinated anthraquinones containing amide side chain, viz. emodacidamides C (**150**), F (**151**), and G (**152**) (Figure 11), were also reported from the fermentation extract of a deep-sea sediment-derived fungus, *Penicillium* sp. SCSIOsof101. The absolute configurations of the amino acids in the amide side chains were assigned by Marfey’s method and chiral-phase HPLC analysis as L-Val in **150**, L-Ile in **151**, and L-Leu in **152** [46].

#### 3.1.3. Sulphated Anthraquinones

Only three anthraquinones containing a sulfate group have been reported from cultures of marine-derived fungi. Macrosposrin-7-*O*-sulfate (**153**) (Figure 12) was reported from the solid-rice culture extract of a mangrove endophytic fungus, *Stemphylium* sp. 33231 [50], whereas emodin-3-*O*-sulphate (**154**) and citreorosein-3-*O*-sulphate (**155**) (Figure 12) were isolated from the mycelia extract of a sea mud-derived *Penicillium oxalicum* 2HL-M-6 [36].

#### 3.1.4. Glycosylated Anthraquinones

Although anthraquinones containing a sugar moiety are not common, some of them have been reported from the cultures of marine-derived fungi. The fermentation extract of *Eurotium rubrum*, obtained from the inner tissue of a stem of the mangrove plant, *Hibiscus tiliaceus*, from Hainan Island, China, furnished 6-*O*-(α-D-ribofuranosyl)-questin (**156**) (Figure 13) [106], while the culture extract of an algicolous fungus, *E. cristatum* EN-220, yielded **157** and 6-*O*-(α-D-ribofuranosyl)-questinol (**157**). The sugar moiety was identified as D-ribose by acid hydrolysis of the glycosides and by subsequent measurement of its optical rotation (${\left[\alpha \right]}_{D}^{20}$ − 17.6°) [95].

Macrosporin 2-*O*-(6′-acetyl)-α-D-glucopyranoside (**158**) (Figure 13) was isolated from the culture extract of a mangrove endophytic fungus, *Stemphylium* sp. 33231 [50], whereas macrosporin 2-*O*-α-D-glucopyranoside (**159**) (Figure 13) was isolated from the solid-rice culture extract of a gorgonian-associated fungus, *S. lycopersici* [51].

#### 3.1.5. *Seco*-Anthraquinones

*Seco*-anthraquinones are proposed to derive from an oxidative cleavage of the *p*-benzoquinone ring of the anthraquinone scaffold, followed by recyclization to form a 7-membered lactone ring. The *seco*-anthraquinones, wentiquinones A (**160**) and B (**161**), and 1,8-dihydroxy-10-methoxy-3-methyldibenzo[*b,e*]oxepin-6,11-dione (**162**) (Figure 14) were isolated from the culture extract of an algicolous fungus, *Aspergillus wentii* EN-48 [30]. The culture extract of a marine sponge-associated fungus, *A. europaeus* WZXY-SX-4-1, yielded **162** and wentiquinone C (**163**) (Figure 14) [21]. The proposed biogenetic pathways of **161** and **162** suggested that emodin is a precursor, which after oxidative cleavage and lactonization of the anthraquinone core, generates **162** and **163** [21]. Compound **163** was also isolated from the fermentation extract of an algicolous fungus, *A. wentii* EN-48 [23].

9-Dehydroxyeurotinone (**164**) and 2-*O*-methyl-9-dehydroxyeurotinone (**165**) (Figure 14) were isolated from the culture extract of a mangrove endophytic fungus, *Eurotium rubrum* [47], while a marine sediment-derived *Eurotium* sp. SCSIO F452 also furnished **165** [70]. Compound **165** was isolated, together with 2-*O*-methyleurotinone (**166**) and 2-*O*-methyl-4-*O*-(α-D-ribofuranosyl)-9-dehydroxyeurotinone (**167**) (Figure 14), from the culture extract of a mangrove endophytic fungus, *E. rubrum* [106].

### 3.2. Tetrahydroanthraquinones

The culture extract of a soft coral-associated fungus, *Aspergillus tritici* SP2-8-1, furnished aspetritone B (**168**) (Figure 15). The relative configurations of the stereogenic carbons (C-2 and C-3) in **168** were established by NOESY correlations from H-1 to H-3 and H-2 to H_ax_-4, while the absolute configurations were established as 2*R*,3*S* by comparison of the calculated and experimental ECD spectra [28].

(3*R*)-1-Deoxyaustrocortilutein (**169**) and altersolanol B (or dactylarin; **170**) (Figure 15) were obtained from the culture extract of a deep-sea sediment-derived *Altenaria tenuissima* DFFSCS013 [63]. Compound **170** was also obtained from a marine-derived fungus, *Altenaria* sp. ZJ-2008003, isolated from a soft coral, *Sarcophyton* sp., which was collected from the South China Sea [107].

The solid-rice culture extract of a mangrove endophytic fungus, *Stemphylium* sp., yielded altersolanol C (**171**) (Figure 15), together with **170**, altersolanol A (**172**), auxarthrol C (**173**), and 2-*O*-acetylaltersolanol B (**174**) (Figure 15). The absolute configurations of the stereogenic carbons in **173** were established as 1*R*,2*R*,3*R*,4*R*,1a*S*,4a*R* by X-ray analysis of the product resulting from the epoxide ring-opening reaction to obtain a suitable crystal for X-ray crystallography. The absolute configurations of the stereogenic carbons in **174** were established as 2*R*,*3**S* by X-ray analysis of a crystal obtained from a hydrolysis reaction, followed by a preparation of its 2,3-*O*-acetonide **[50]**. Compounds **170**, **172** and **173** (Figure 15) were also isolated from the solid-rice culture extract of a gorgonian-associated fungus, *S. lycopersici* [51].

Antibacterial activity-guided fractionation of the culture extract of a sea cucumber-associated fungus, *Trichoderma* sp. (H-1), resulted in the isolation of lentisone (**175**) (Figure 15) [43], whereas SZ-685C (also known as 1403C; **176**) (Figure 15) was isolated from the culture extract of a mangrove endophytic fungus, *Halorosellinia* sp. (no. 1403) [108].

Phomopsanthraquinone or (2*R*,3*S*)-7-ethyl-1,2,3,4-tetrahydro-2,3,8-trihydroxy-6-methoxy-3-methyl-9,10-anthracenedione (**177**) (Figure 15) was reported from the broth culture extract of *Phomopsis* sp. PSU-MA214, isolated from the leaves of a mangrove tree, *Rhizophora apiculata* Griff. Ex T. Anderson. The relative configurations of C-2 and C-3 in **177** were established by NOEDIFF experiment, while their absolute configurations were suggested to be the same as those of **170**, i.e., 2*R*,3*S* since the specific rotation of **177** (${\left[\alpha \right]}_{D}^{25}$ − 58°, *c* 0.05, EtOH) was almost identical to that of **170** (${\left[\alpha \right]}_{D}^{25}$ − 63°, *c* 0.05, EtOH) [52].

Ge et al. reported the isolation of auxarthrol D (**178**), a chlorine-containing auxarthrol G (**179**), and 4-dehydroxyaltersolanol A (**180**) (Figure 15), in addition to **170**, from the culture extract of *Sporendonema casei* HDN16-802, isolated from sediment samples collected from Zhangzi Island, Liaoning province, China. The relative configurations at C-2, C-3 and C-4 in **178** and **179** were established based on NOESY correlations, while their absolute configurations were established as 2*S*,3*R*,4*S*,1a*R*,4a*R* by comparison of the calculated and experimental ECD spectra [109].

### 3.3. Tetrahydro-5,8-anthraquinones

Chemical investigation of the culture extract of a soft coral-associated fungus, *Aspergillus tritici* SP2-8-1, resulted in the isolation of aspetritone A (**181**) (Figure 16). The relative configurations at C-1, C-2 and C-3 were established based on NOESY correlations, while their absolute configurations were determined as 1*S*,2*S*,3*R* by comparison of the calculated and experimental ECD spectra [28].

The culture extract of *Aspergillus* sp. strain 05F16, isolated from an unidentified alga collected in the coral reef at Manado, Indonesia, yielded bostrycin (**182**) (Figure 16) [110]. Compound **182** was also isolated from the culture extract of a mangrove endophytic fungus strain no. 1403, collected from the South China Sea [111].

Nigrosporins A (**183**), B (**184**) and a spiro dihydronaphthoquinone/tetrahydroantharquinone derivative, fusarnaphthoquinone C (**185**) (Figure 16), were isolated, together with **182**, from the extracts of the culture broth and mycelia of a gorgonian-associated fungus, *Fusarium* sp. PSU-F14 and PSU-F135. The NOEDIFF experiment was used to locate the methyl group on the tetrahydro-5,8-antharquinone moiety and the 2-oxopropyl group on the dihydronaphthoquinone portion. However, neither relative nor absolute configurations of the stereogenic carbons in **185** were determined [59].

Deoxybostrycin (**186**) (Figure 16) was obtained, together with **182**, from the culture extract of a mangrove endophytic fungus, *Nigrospora* sp. (strain no. 1403), isolated from a decayed wood of a mangrove plant, *Kandelia candel* (L.) Druce, collected from Mai Po, Hong Kong [112]. 10-Deoxybostrycin (**187**) (Figure 16) was isolated, together with **182**, **184** and **186**, from the culture extract of *Nigrospora* sp. ZJ-2010006, isolated from an unidentified sea anemone. Acetylation of **182** and **186** gave 3-acetoxybostrycin (**182a**) and 3-acetoxy-4-deoxybostrycin (**186a**) (Figure 16), respectively. The 1D NOE data of **188** showed that all asymmetric carbons had the same relative configurations as those of **182**. The absolute configurations of the stereogenic carbons in **187** were tentatively assigned as 2*S*,3*R*,4*S* on the ground that **187** shared a biogenesis with 4a-*epi*-9α-methoxydihydrodeoxybostrycin whose absolute structure had already been established [68]. Compound **187** was also reported from the same fungal strain but was isolated from the inner tissue of the zoathid, *Palythoa haddoni* [69]. A bostrycin derivative, hydroxybostrycin (**188**) (Figure 16), was isolated from the culture broth extract of a mangrove endophytic fungus, *Altenaria* sp. (SK11) [62], whereas 1403P-3 (**189**) (Figure 16) was reported from a mangrove endophytic fungus, strain no. 1403 [113].

### 3.4. Anthrones

Anthrone derivatives reported from marine-derived fungi occur as complex structures with the anthrone or modified anthrone scaffolds. These compounds can be considered to derive from a condensation of the anthraquinone scaffold, such as physcion (**6**) and catenarin (**7**) (Figure 3) with polyketides of diketopiperazine derivatives.

Du et al., in their search for antitumor compounds from marine-derived microorganisms, have isolated anthraquinone derivatives from *Aspergillus glaucus* HB1-19, isolated from a marine sediment around the mangrove roots, collected in Fujian Province, China. Fractionation of the culture extract furnished aspergiolide A (**190**) (Figure 17), an anthraquinone derivative with naphtha{1,2,3-*de*}chromene-2,7-dione skeleton [114], whereas aspergiolide B (**191**) (Figure 17) was isolated from the culture extract of *A. glaucus* HB1-19, isolated from a marine sediment, collected in Fujian Province, China [22].

Further investigation of the mycelial extract of *A. glaucus* HB1-19, isolated from the marine sediment-surrounding mangrove roots collected in Fujian Province (China), by the same authors led to the isolation of aspergiolides C (**192**) and D (**193**) (Figure 17), two spiro [5,5]undecane scaffold-containing anthrones. Although **192** and **193** possess a stereogenic center at a spiro junction of the ring system (C-19), both compounds displayed no optical rotation and CD effects. Therefore, both compounds were assumed to be a 1:1 mixture of enantiomers. By using HPLC with a Lux-Amylose-2 column, each compound gave a baseline-separated peaks in a 1:1 ratio for both compounds, confirming their racemic nature. Of these peaks, HPLC-CD spectra were recorded in the stopped-flow mode and the resulting opposite CD curves confirmed the assumption that the two peaks represent their enantiomers. Comparison of the online and calculated CD spectra and the configurations of both enantiomers of **192** and **193** were established [115].

Biosynthetically, **190** was proposed to arise from a condensation of catenarin (**7**) (Figure 3) with aromatic pentaketide, as depicted in Figure 17 [114], while **192** was proposed to derive from catenarin (**7**) with an aromatic heptaketide as shown in Figure 17 [115].

Three pairs of anthrone-based racemic spirocyclic diketopiperazine enantiomers, variecolortins A (**194**), B (**195**) and C (**196**) (Figure 18), were obtained from *Eurotium* sp. SCSIO F452, isolated from the South China Sea sediment samples. Compounds **194**–**196** represented a 6/6/6/6 tetracyclic cyclohexene–anthrone skeleton. The relative configurations of the stereogenic carbons in **194** were unambiguously determined as (12*R*,21*S*,32*R*) by X-ray analysis. However, the lack of optical rotation of **194** suggested its racemic nature. The enantiomers were subsequently separated by a chiral HPLC to give (+)-**194** and (−)-**194**. Conversely, the relative configurations of **195** and **196** were established by NOESY experiments. In each compound, the diagnostic NOESY correlations between NH-11 and H-21b, as well as between OH-22 and H-21a, resulted in the identification of α- and β-orientations, respectively. In addition, the geometry of the Δ^8^ double bond was assigned as *Z*-configuration via the deshielding effect of H-8 caused by the carbonyl group on the β-vinyl proton. The baseline ECD curves of **195** and **196** revealed that they were racemic mixtures. Therefore, **195** and **196** were separated by a chiral-phase HPLC, and the calculated ECD spectra for the individual enantiomer assigned them as 12*S*,22*R*-**195** and 12*S*,22*R*-**196**, which were in agreement with the experimental ECD spectra of (+)-**195** and (−)-**196**, respectively.

Hydroxyviocristin (Figure 18) was proposed to be a biosynthetic precursor of (±)-**194**, while physcion (**6**) (Figure 3) was proposed as a biosynthetic precursor of (±)-**195** and (±)-**196** [116]. The proposed biosynthetic pathways leading to the formation of **194**–**196** are depicted in Figure 18.

### 3.5. Tetrahydro-9-hydroxyanthrones

Terahydro-9-hydroxyanthrones are considered to derive from a reduction of the carbonyl group on C-10 of tetrahydroanthraquinones to a hydroxyl group. This group of anthraquinone derivatives are widely isolated from culture extracts of marine-derived fungi.

The culture extract of an algicolous *Aspergillus* sp. strain 05F16 furnished tetrahydrobostrycin (**197**) and 1-deoxytetrahydrobostrycin (**198**) (Figure 19). The relative configurations of the stereogenic carbons of **197** were assigned by analysis of the ^1^H-^1^H coupling constants and NOESY correlations [110].

The fermentation extract of an endophytic fungus, *Talaromyces islandicus* EN-501, isolated from the inner tissue of a marine red alga, *Laurencia okamurai*, collected in Qingdao, China, yielded 8-hydroxyconiothyrinone B (**199**), 8,11-dihydroxyconiothyrinone B (**200**), 4*R*,8-dihydroxyconiothyrinone B (**201**), 4*S*,8-dihydroxyconiothyrinone B (**202**), and 4*S*,8-dihydroxy-10-*O*-methyldendroyl E (**203**) (Figure 19). The relative stereochemistry of **199** was determined by analysis of ^1^H-^1^H coupling constants as well as by NOESY correlations. The large coupling constant value (*J* = 8.8 Hz) between H-9 and H-9a revealed a *trans* orientation. The important NOE correlations were observed between H-9 and H-4a, indicating a co-facial orientation of the two protons, while the NOE correlations between H-2 and H-9a showed that they were on the opposite sides of the molecule. The absolute configurations of the stereogenic carbons of **199** were established as 2*S*,4a*S* and 9*R*,9a*S* by X-ray analysis. The relative and absolute configurations of the stereogenic carbons of **200** and **201** were deduced to be the same as those in **199**. However, the measured ECD spectrum of **201** suggested the *R* absolute configuration at C-9 in **199–201**. The ^1^H and ^13^C NMR data revealed that **202** is a C-4 epimer of **201**. The absolute configurations of the stereogenic carbons in **202** were determined as 2*R*,4*S*,4a*R*,9*R*,9a*S* by comparison of its calculated and experimental ECD spectra. The relative configuration of **203** was established on the basis of NOESY correlations, while the absolute configurations of its stereogenic carbons were established as 2*S*,4*S*,4a*S*,10*S*,9a*S* by comparison of the calculated and experimental ECD spectra [117].

Fusaquinons A (**204**), B (**205**), and C (**206**) (Figure 19) were isolated from the fermentation extract of *Fusarium* sp. (no. ZH-210), obtained from a mangrove sediment from Zhuhai, China. The relative configurations at C-6, C-7, C-8a, C-9 and C-10a of **204** were established based on NOESY correlations. The structures of **205** and **206** were elucidated as 1,4,5,6,7,9-hexahydroxy-2-methoxy-7α-methyl-5β,6α,8αβ,8aβ,9α,10aα-hexahydroanthracen-10-one and 1,4,6,7,9-pentahydroxy-2-methoxy-7α-methyl-5αβ,6α,8αβ,8aβ,9α,10aα-hexahydroanthracen-10-one, respectively [118].

Fusaranthraquinone (**207**), 9α-hydroxydihydrodesoxybostrycin (**208**), 9α-hydroxyhalorosellinia A (**209**) (Figure 19) were isolated from the culture broth and mycelia extracts of a gorgonian sea fan-associated fungi, *Fusarium* sp. PSU-F14 and PSU-F135 [59].

4a-*Epi*-9α-methoxydihydrodeoxybostrycin (**210**) (Figure 19), together with **208**, were reported from the culture extract of a sea anemone-associated fungus, *Nigrospora* sp. ZJ-2010006. The absolute configurations of the stereogenic carbons in **210** were established as 2*S*,3*R*,9*R*,1a*S*,4a*R* by X-ray crystallographic analysis. Based on the absolute structure of **210**, the absolute configuration of the stereogenic carbons of **208**, whose relative configuration was previously determined, was unambiguously established as 2*S*,3*R*,9*R*,1a*S*,4a*S* [68].

Dihydroaltersolanol A (**211**), altersolanol L (**212**) and ampelanol (**213**) (Figure 19) were obtained from the culture extract of a deep-sea sediment-derived fungus, *Altenaria tenuissima* DFFSCS013 [63], whereas **213** was also isolated from the fermentation extract of a gorgonian soft coral-associated fungus, *Stemphylium lycopersici* [51]. Tetrahydroaltersolanol B (**214**) (Figure 19) was isolated from the mycelia extract of a mangrove endophytic fungus, *Altenaria* sp. ZJ9-6B. The absolute configurations of the stereogenic carbons in **214** were established by X-ray diffraction analysis [61].

Halorosellinia A (**215**), or 1,4,5,6,7,9-hexahydroxy-2-methoxy-7-methyl-5β,9β,8aβ,6α,10aα-hexahydroanthracen-10(10aH)-one, was isolated, together with **197** and **208** (Figure 19), from the culture broth extract of a mangrove endophytic fungus, *Altenaria* sp. (SK11) [62]. Compound **215** was also reported from the culture extract of a mangrove endophytic fungus, *Halorosellinia* sp. (no. 1403). The relative stereochemistry of **215** was established by NOE correlations and ^1^H–^1^H coupling constants [58].

Tetrahydroaltersolanols C (**216**), D (**217**), E (**218**), and F (**219**) (Figure 19) were isolated, together with **211**–**213**, from the culture extract of a soft coral-associated fungus, *Altenaria* sp. ZJ-2008003. The relative configuration of **219** was determined by observation of the correlations from the ROESY spectrum, while the absolute configurations of its stereogenic carbons were established as 2*S*,3*R*,4a*S*,9*R*,9a*S* by a modified Mosher’s method. Based on the absolute configurations of the stereogenic carbons of **219** and a shared biogenesis, the absolute configurations of the stereogenic carbons of **211** and **216**–**218** were established as 1*R*,2*R*,3*R*,9*R,*9a*S*-**211**, 2*S*,3*R*,4a*S*,9*S,*9a*S*-**216**, 2*S*,3*R*,4a*R*,9*R,*9a*R*,-**217**, and 2*S*,3*S*,4a*S*,9*R,*9a*R*,-**218**, respectively [49]. Compounds **213**, **214** and **216** (Figure 19) were also reported from the culture extract of a mangrove endophytic fungus, *Phomopsis* sp. PSU-MA214 [52].

2-*O*-Acetylaltersolanol L (**220**) (Figure 19) was isolated, together with **211–214**, from the culture extract of a mangrove endophytic fungus, *Stemphylium* sp. 33231. The absolute configurations of the stereogenic carbons of **220** were established as 1*R*,2*S*,3*R*,4a*S*,9*R,*9a*S* by X-ray analysis of the deacetylated product [50].

Harzianumnones A (**221**) and B (**222**) (Figure 19) were isolated from the culture extract of a soft coral-associated fungus, *Trichoderma harzianum* (XS-20090075). The absolute configurations of the stereogenic carbons in **221** and **222** were established as 7*R*,8*R*,8a*R*,10*S*,10a*S* and 7*R*,8*R*,8a*R*,10*R*,10a*S*, respectively, by comparison of their calculated and experimental ECD spectra. Compounds **221** and **222** are C-10 epimers [55]. The culture extract of a sea cucumber-associated fungus, *Trichoderma* sp. (H-1), furnished coniothyrinone A (**223**) (Figure 19). Compound **223** is a C-7 and C-8 diastereomer of **221** [43].

Xylanthraquinone (**224**) (Figure 19) was isolated from the culture extract of a mangrove endophytic fungus, *Xylaria* sp. 2508. The absolute configurations of the stereogenic carbons in **224** were determined as 2*S*,3*R*,4a*S*,9*R*,9a*S* by single-crystal X-ray diffraction using Cu Kα radiation [119]. Auxarthrols E (**225**), F (**226**), and H (**227**) (Figure 19), were isolated from the fermentation extract of a sediment-derived fungus, *Sporendonema casei* HDN16. The relative stereochemistry of **225**, **226**, and **227** was determined by NOESY correlations, while their absolute structures were determined as 2*R*,3*R*,4*S*,9*S*,1a*R*,4a*R*-**225**, 2*S*,3*R*,4*R*,9*R*,1a*S*,4a*R*-**226**, and 2*S*,3*R*,4*S*,9*S*,1a*S*,4a*S*-**227**, respectively, by comparison of their calculated and experimental ECD spectra [109].

### 3.6. Tetrahydroanthrols

Asperflavin (**228**) (Figure 20) was obtained, as the main pigment, from the fermentation extract of a marine sponge-associated fungus, *Eurotium repens*. Since **228** did not display an optical rotation, it was suggested to be a racemic mixture [24]. Compound **228** and isoasperflavin (**229**) (Figure 20) were isolated from the culture extract of a mangrove sediment-derived *Aspergillus glaucus* HB1-19. The relative configurations of C-3 and C-4 in **229** were determined by the value of the coupling constant between H-3 and H-4 (*J_3,4_* 7.7 Hz), while the absolute configurations at C-3 and C-4 were established as 3*R*,4*S* on the basis of maximal-negative and minimal-positive CEs at 281.0 and 223.4 nm, respectively, in the CD spectrum [22]. 3,4-Dihydro-3,9-dihydroxy-6,8-dimethoxy-3-methylanthracen-1(2*H*)-one (**230**) (Figure 20) was isolated from the fermentation extract of an algicolous fungus, *A. wentii* EN-48 [30].

Eurorubrin (**231**) (Figure 20), a bisdihydroanthracenone derivative, was obtained, together with **228**, from the culture extract of a mangrove endophytic fungus, *Eurotium rubrum*. Since both **231** (${\left[\alpha \right]}_{D}^{25}$ + 21.1°) and **228** were dextrorotatory, they were suggested to have the same stereochemistry at C-3 [106]. The culture extract of an algicolous fungus, *E. cristatum* EN-220, furnished asperflavin ribofuranoside (**232**) (Figure 20), in addition to **228** and **231** [95].

### 3.7. 9,10-Dihydroxyanthracenes

Anthrininone A (**233**) (Figure 21) was obtained from the culture extract of a deep-sea sediment-derived fungus, *Altenaria tenuissima* DFFSCS013. The absolute configurations of its stereogenic carbons were established as 4*R*,6*S*,7*R*,15*R*,17*S*,18*R* by a single-crystal X-ray diffraction analysis using Cu Kα radiation [63]. The proposed biosynthetic pathway leading to a formation of a hexacyclic spiro-fused ring system in **233** was shown in Figure 21. A condensation of the intermediate (**i**), derived from a cyclization of an octaketide, with the intermediate (**ii**), which is derived from a nucleophilic addition of D-xylose by acetoacetyl CoA, led to a formation of a spiro ketal in **233**.

### 3.8. 2-Aza-anthraquinones

2-Aza-anthraquinones consist of a naphthoquinone moiety fused with a pyridine ring. These compounds are synthesized in nature by either fungi or lichens. Van Wagoner et al. reported the isolation of scorpinone (**234**) (Figure 22) from the extract of a rare fungus, *Amorosia littoralis*, collected from an inertial sediment in the Bahamas. The biosynthetic pathway of **234** was studied using [2-^13^C]-acetate and [1,2-^13^C]-acetate, and was followed to verify if its biosynthesis was similar to that of bostrycoidin (**235**) (Figure 22). The labeling results showed that a linear heptaketide is a precursor in the biosynthesis of **234**, and consequently the incorporation of a nitrogen atom produced 2-aza-anthraquinones [120]. *A. littoralis* gen. sp. nov., also isolated from the littoral zone in the Bahamas, was capable of producing **234** (Figure 22) and caffeine [121]. Chemical investigation of the CHCl_3_-MeOH extract of cultured mycelia of *Bispora*-like tropical fungus, collected from the intertidal zone surrounding the Bahamas Island, also led to the isolation of **234** (Figure 22) [122].

The culture extract of an endophytic fungus, *Aspergillus terreus* (no. GX7-3B), isolated from a branch of a mangrove tree, *Bruguiera gymnoihiza* (Linn.) Savigny, which was collected from the salt coastline of the South China Sea in Guangxi province, yielded 8-*O*-methylbostrycoidin (**236**) (Figure 22) [123].

### 3.9. Dimeric anthraquinones

The compounds of this group include two anthraquinoid units, one anthraquinone and one tetrahydroanthraquinone, two tetrahydroanthraquinones, one anthrone and one tetrahydro-5,8-anthraquinone, or one anthraquinone and one seco-anthraquinone, linked together by C-O-C or C-C bonds.

6,6′-Oxybis(1,3,8-trihydroxy-2-((*S*)-1-methoxyhexyl)anthracene-9,10-dione) (**237**) and 6,6′-oxybis(1,3,8-trihydroxy-2-((*S*)-1-hydroxyhexyl)anthracene-9,10-dione) (**238**) (Figure 23) were reported from the culture broth extract of a marine clam-associated fungus, *Aspergillus versicolor*. The ^1^H and ^13^C NMR data of **237** resembled those of **81**, except for the signals of H-2 and H-4. Based on the sign of their optical rotations (${\left[\alpha \right]}_{D}^{23}$ = −72.4° for **237**, and −51.4° for **238**), the absolute configurations of the stereogenic centers at C-11 and C-11′ in **237** and **238** were determined as *S* [88].

2,2′-*Bis*-(7-methyl-1,4,5-trihydroxyanthracene-9,10-dione) (**239**) (Figure 24) was obtained from the fermentation extract of a marine sponge-associated fungus, *Talaromyces stipitatus* KUFA 0207 [33].

Alterporriols K (**240**), L (**241**), and M (**242**) (Figure 24) were obtained from the mycelial extract of a mangrove endophytic fungus, *Altenaria* sp. ZJ9-6B. The relative configurations at C-5 and C-8 in **240** were established as 5*S**,8*R** on the basis of NOESY correlations and the value of a coupling constant between H-5 and H-6α, whereas the relative configurations of H-6, H-7 and H-8 in **241** and **242** were established as 6*S**,7*R**,8*R** and 6*S**,7*R**,8*R**, respectively, by the NOE experiment and the value of a coupling constant between H-5α and H-6. Compound **241** is, therefore, a C-7 epimer of **242** [61].

Xia et al. reported the isolation of alterporriol S (**243**) and (+)-a*S*-alterporriol C (**244**) (Figure 24) from the culture broth extract of a mangrove endophytic fungus, *Altenaria* sp. (SK11). The relative stereochemistry of **243** was determined as 6*S**,7*R**,8*S**,8a*S**,10*R**,10a*R** and 6′*R**,7′*S** by the ^1^H-^1^H coupling constant values as well as by NOESY correlations, while the absolute configurations of the stereogenic carbons were established as 6*S*,7*R*,8*S*,8a*S*,10*R*,10a*R* and 6′*R*,7′*S* by comparison of calculated and experimental ECD spectra. The planar structure of **244**, elucidated by high-resolution mass spectrometry (HRMS) and 1D and 2D NMR analyses, was the same as that of alterporriol C. The relative configurations of the stereogenic carbons of **244** were also established by ^1^H-^1^H coupling constants and the correlations observed in the NOESY spectrum as 5′*R**,6′*S**,7′*R**,8′*S*.* Since **244** displayed the specific rotations [$\alpha {]}_{D}^{27}\;$+ 75° and +208°; *c* 0.02, in EtOH, it was suggested to be an atropisomer of alterporriol C. Comparison of the calculated and experimental ECD spectra of **244** revealed the absolute configuration of its stereogenic carbons as 5′*R*,6′*S*,7′*R*,8′*S*. Thus, the axial configuration of **244** was identified as a*S*, also called M helicity [62].

The culture extract of a soft coral-associated fungus, *Altenaria* sp. ZJ-2008003, also afforded, besides the previously reported alterporriol C (**245**), another five alterporriol-type anthranoid dimers, i.e., alterporriols N (**246**), O (**247**), P (**248**), Q (**249**), and R (**250**) (Figure 24) [49].

The liquid culture extract of a zoathid *Palythoa haddoni*-associated fungus, *Nigrospora* sp. (ZJ-20100026), afforded a hydroanthrone dimer, nigrodiquinone A (**251**) (Figure 24). The relative configurations of the stereogenic carbons of **251** were assigned as 1a*R**,2*S**,3*R**,4a*R**,9*R**,2′*S**,3′*R**,4′*S**, which were the same as those of 4a-*epi*-9α-methoxydihydrodeoxybostrycin (**210**) and 10-deoxybostrycin (**187**). The absolute configurations of the stereogenic carbons of **251** were established as 1a*R*,2*S*,3*R*,4a*R*,9*R*,2′*S*,3′*R*,4′*S* by comparison of the calculated and experimental ECD spectra as well as of the values of the calculated and experimental optical rotations [69].

The previously described cytoskyrin A (**252**) (Figure 24) was isolated from the culture broth and mycelial extracts of a marine sponge-associated fungus, *Curvularia lunata* [64].

The solid-rice culture extract of a mangrove endophytic fungus, *Stemphylium* sp. 33231, furnished alterporriols A (**253**), B (**254**), D (**255**), E (**256**), T (**257**), U (**258**), V (**259**), and W (**260**) (Figure 25), in addition to **245**, **246**, **249** and **250** (Figure 24). Compound **257** is a heterodimer consisting of **170** and **171** linked by C-5−C-7′, whereas **258** is a homodimer of **170** linked by C-5 and C-7′, and **260** is a heterodimer of **170** and **33** linked by C-1 and C-5′. The configurations of the stereogenic carbons of **257** were tentatively assigned as 2*R*,3*R*,4*R*,2′*R*,3′*S*, whereas those of **258** and **260** were also assigned as 2*R*,3*S*,2′*R*,3′*S* and 2′*R*,3′*S*,4′*R*. However, the absolute configurations for the axes of chirality in **258**–**260** were not determined. Since the CD spectrum of **260** showed the same spectral feature in the 205–340 nm range as **256**, the overall absolute configuration of **260** was tentatively assigned as a*R*,2′*R*,3′*R*,4′*S* [50].

Alterporriol Y (**261**) (Figure 25) was isolated from the EtOAc extract of a liquid culture of a gorgonian soft coral-associated fungus, *S. lycopersici*. Compound **261** is a homodimer of **171** linked by C-8−C-8′. The relative configurations of the stereogenic carbons of **261** were determined as 2*S**,3*R** on the basis of NOE experiment. The ECD spectrum of **261** presented two negative CEs at 252 and 227 nm and two positive CEs at 306 and 272 nm, which was a mirror image of the ECD spectra of **246** and **255**, with a*S* axial chirality, and close similarity to that of a*R*-**256**. Therefore, the absolute structure of **261** was assigned as a*R*,2*S*,3*R*,2′*S*,3′*R* [51].

Alterporriols F (**262**), G (**263**), Z1(**264**), Z2 (**265**), and Z3 (**266**) (Figure 25), along with **246** (Figure 24), were isolated from the MeOH extract of the solid-rice culture of *Stemphulium* sp. FJJ006, obtained from an unidentified sponge, which was collected at the coast of Jeju Island, Korea. The relative configurations of the stereogenic carbons of **264** were assigned as 1′*S**,2’*R**,3′*S**,4′*S** on the basis of ^1^H–^1^H coupling constants and NOESY correlations. Since the experimental ECD spectrum showed significant CEs at **269** (Δε 35.79) and **285** (Δε −36.06) nm, the a*R* (also defined as P helicity) configuration at C-6-C-6′ was assigned to **264**. However, the calculated ECD spectra did determine the absolute configuration of C-1-C-4. The 1D and 2D NMR analysis revealed that the planar structure and relative stereochemistry of **265** were the same as those of **264**. However, the experimental ECD spectrum of **265** was *quasi*-mirror image of **264**, indicating that **265** is an atropisomer of **264**. The relative configurations of the stereogenic carbons in **266** were determined as 1′*R**,2′*R**,3′*S**,4′*S** on the basis of the ^1^H–^1^H coupling constants and NOESY correlations, whereas the absolute configurations of the C-1/C-6′ chiral axis was assigned as a*R*, based on the similarity of its ECD spectrum to that of **264**. Moreover, the ECD spectrum of **262** also assigned the configuration of a C-5/C-5′ chiral axis as a*R* [124].

Antibacterial activity-guided fractionation of the culture extract of an unidentified marine red alga-derived fungal strain F-F-3C led to the isolation of rubellin A (**267**), 14-acetoxyrubellin A (**268**), and 14-acetoxyrubellin C (**269**) (Figure 25). The structures of **268** and **269** were elucidated by 1D and 2D NMR analysis and comparison of their NMR data with those of the previously reported **267**; however, the relative and absolute configurations of their stereogenic carbons were not described [44].

### 3.10. Bianthrones

*Trans*- and *cis*-emodin-physcion bianthrones (**271** and **272**) (Figure 26) were isolated, together with **270**, from the culture extract of a marine sediment-derived fugus, *Aspergillus glaucus* HB1-19. The *cis* and *trans* relationship between C-10/C-10′ of **271** and **272** was determined based on a comparison of their NMR data with those from the literature [22]. Two atropisomers of 8,8′-dihydroxy-1,1′,3,3′-tetramethoxy-6,6′-dimethyl-10,10’-bianthrone (**273** and **274**) (Figure 26) were obtained, together with **270**, from the culture extract of an algicolous fungus, *Aspergillus wentii* EN-48 [30].

Three chlorinated bianthrones, allianthrones A (**275**), B (**276**), and C (**277**) (Figure 26), were isolated from the EtOAc extract of the co-culture of two different developmental stages of a marine alga-derived *Aspergillus alliaceus*. The structures of **275**–**277** were elucidated by 1D and 2D NMR spectral analysis. The absolute configurations of the stereogenic carbons of **275** were established as 10*R*,10′*S*,12*S*,12′*S* by X-ray analysis, whereas those of the *pseudo*-enantiomers, **276** and **277**, were determined as 10*R*,10′*R*,12*R*,12′*S* and 10*S*,10′*S*,12*S*,12′*S*, respectively, by comparison of their calculated and experimental ECD spectra [77].

Eurotone A (**278**) (Figure 26) was isolated from the culture extract of a marine sediment-derived fungus, *Eurotium* sp. SCSIO F452. X-ray diffraction analysis not only confirmed its planar structure, elucidated by 1D and 2D NMR analysis, but also determined the relative configuration of its stereogenic carbons as 10*S**,10′*S**. However, the crystal of **278** occupied a *Pccn* space group, indicating its racemic nature, which was also supported by its lack of optical activity. Separation of (±)-**278** by chiral HPLC yielded (+)-**278** and (−)-**278**, whose absolute configurations were established as 10*S*,10′*S* and 10*R*,10′*R*, respectively, by comparison of their calculated and experimental ECD spectra [70]. The proposed biosynthetic pathway of **278** from physcion (**6**) (Figure 3) as a precursor was depicted in Figure 26.

### 3.11. Anthraquinone Analogues Fused with Xanthone and Chromone Derivatives

The previously described anthraquinone–xanthone derivatives, JBIR-97/98 (**279**) and JBIR-99 (**280**) were isolated, together with engyodontochones A (**281**), B (**282**), C (**283**), D (**284**), E (**285**), and F (**286**) (Figure 27) from the mycelia and culture broth extracts of *Engyodontium album* strain LF069, which was isolated from a tissue of a marine sponge, *Cacospinga scalaris**,* collected from the Limski Fjord, Croatia. The relative configurations of **279–286** were determined by NOESY correlations and ^1^H–^1^H coupling constants. The absolute configurations of **279**, **280** and **282** were established as 9*R*,10*S*,12*S*,24*R*,25*S*-**279**, 9*R*,10*S*,12*S*,24*R*,25*R*-**280** and 9*R*,10*S*,12*S*,24*S*,25*S*-**282**, whereas the absolute configurations of the stereogenic carbons in **281** were established, based on its common biogenesis with **279** and **280**, as 9*R*,10*S*,12*S*,24*R*,25*S.* The calculated ECD spectra of **283**–**286** only determine the absolute configurations of C-9, C-10, C-12 and C-24, but not C-25. Consequently, the absolute configurations of these compounds were established as 9*R*,10*S*,12*S*,24*R*-**283**, 9*R*,10*S*,12*S*,24*R*-**284**, 9*R*,10*S*,12*S*,24*R*-**285**, and 9*R*,10*S*,12*S*,24*S*-**286** by comparison of the calculated and experimental ECD spectra [125].

By using UPLC-ESI-QToF/MS analysis, Martins et al. identified acremonidins A (**287**), B (**288**), C (**289**), G (**290**) and acremoxanthones A (**291**), B (**292**), D (**293**), F (**294**), and G (**295**) (Figure 27) from the EtOAc extracts of a culture broth and mycelia of *Acremonium camptosporum*, isolated from a marine sponge, *Aplysina fulva*, which was collected from the mid-Atlantic Saint Peter and Saint Paul Archipelago, Brazil [126].

Ayers et al. described the isolation of the previously described anthraquinone-xanthone derivatives i.e., **287**, **289**, and **294,** together with acremoxanthone C (**296**) (Figure 27), from the solid-rice culture extract of an unidentified fungus of the order Hypocreales (MSX 17022), which was obtained from leaf litter from a beech tree community in Hillsborough, NC, USA [127].

## 4. Biological Activitiesq

### 4.1. Antibacterial and Antibiofilm Activities

Compounds **1**, **9**–**11** (Figure 3), **168** (Figure 15) and **181** (Figure 16), isolated from a culture extract of a soft coral-associated fungus, *Aspergillus tritici* SP2-8-1, were assayed for antibacterial activity against MRSA *Staphylococcus aureus* (ATCC 43300 and CGMCC 1.12409), *Vibrio vulnificus* MCCC E1758, *V. rutiferianus* MCCC E385, and *V. campbellii* MCCC E333. Compound **181** showed potent activity against all the tested strains, with minimum inhibitory concentration (MIC) values of 7.53, 7.63, 31.47, 31.17, and 15.53 μg/mL, while **168** exhibited weaker activity, with MIC values of 15.27, 15.63, 15.47, 31.33, and 15.77 μg/mL against MRSA *S. aureus* (ATCC 43300 and CGMCC 1.12409), *V. vulnificus* MCCC E1758, *V. rutiferianus* MCCC E385, and *V. campbellii* MCCC E333, respectively. Compound **1** showed similar activity to **168**, with MIC values of 15.65, 15.53, 15.73, 62.67, and 31.35 μg/mL. Compound **9** only exhibited activity against both strains of MRSA, with MIC values of 31.32 and 31.33 μg/mL. The positive control, chloramphenicol, displayed MIC values of 7.67 and 7.87 μg/mL against MRSA-ATCC 43300 and CGMCC 1.12409, respectively. Compound **10** selectively inhibited the growth of *V. rutiferianus* MCCC E385 (MIC = 31.28 μg/mL), while the positive control, erythromycin, showed MIC = 3.93 μg/mL [28].

Compounds **1**, **15** (Figure 3) and **37** (Figure 4), isolated from the culture extract of a mangrove-derived endophytic fungus, *Eurotium chevalieri* KUFA 0006, were tested for antibacterial and antibiofilm activities. Compound **1** showed antibacterial activity against *Enterococcus faecalis* ATCC29212 and *S. aureus* ATCC25923, with an MIC values = 64 and 32 μg/mL, respectively (the positive control, cefotaxime, has MIC values ranging from 0.031 to 16 μg/mL). Compounds **15** and **37**, at a concentration of 64 μg/mL, caused a significant reduction in biofilm formation in *Escherichia coli* ATCC25922 (percentage of biofilm production; 56.1% and 50.6%, respectively), while **1** and **6** (Figure 3) displayed an inhibition of a biofilm formation in *S. aureus* ATCC25923 [31].

Compounds **1** (Figure 3) and **20** (Figure 4), isolated from the fermentation extract of a marine sediment-derived, *A. versicolor*, were tested against MRSA-ATCC 43300 and MRSA-CGMCC 1.12409, *V. vulnificus*, *V. rotiferianus*, and *V. campbellii*. Compound **20** showed potent antibacterial activity against MRSA-ATCC 43300 and MRSA-CGMCC 1.12409, with MIC values = 3.9 and 7.8 μg/mL, respectively (The positive control, chloramphenicol, displayed MIC = 7.78 μg/mL against both MRSA ATCC 43300 and CGMCC 1.12409), and moderate antibacterial activity against *V. vulnificus*, *V. rotiferianus*, and *V. campbellii,* with MIC values ranging from 15.6 to 62.5 μg/mL. Conversely, **1** showed moderate activity against MRSA-ATCC 43300 and MRSA-CGMCC 1.12409 with MIC = 15.6 μg/mL for both strains, and weak activity against *V. vulnificus*, *V. rotiferianus*, and *V. campbellii,* with MIC values ranging from 15.6–62.5 μg/mL. The positive control, erythromycin, displayed MIC values of 2, 3.9 and 7.8 μg/mL against *V. vulnificus*, *V. rotiferianus*, and *V. campbellii,* respectively. Molecular docking studies showed that **20** also bound to the AmpC *β*-lactamase receptor with good least binding energy of −4.45 kcal/mol, indicating hydrogen bond interactions of OH-1 and CH(OCH_3_)_2_-O in **20** with Arg148, a π-π interaction of the fused ring system with the benzene ring of Tyr150, and hydrophobic interactions with Lys290, Ala292, Leu293, Ala294, Lys315, and Thr316 residues [35].

Wang et al., in their screening for compounds produced by marine-derived fungi that inhibit biofilm formation in *S. aureus*, have found that **1** (Figure 3) and **28** (Figure 4), isolated from *Penicillium* sp. SCSGAF 0023 (CCTCC M 2012507), exhibited antibiofilm activity. Compound **1**, at a concentration of 12.5 µg/mL, was able to inhibit a biofilm formation more than 50%, while **28** was less active, inhibiting biofilm formation less than 37% at a concentration of 25 µg/mL [128].

Compounds **1** (Figure 3), **29** (Figure 4), **175** (Figure 15) and **223** (Figure 19), isolated from a sea cucumber-associated fungus, *Trichoderma* sp. (H-1), were evaluated for their antibacterial activity against three marine pathogenic bacteria, *V. parahaemolyticus*, *V. anguillarum,* and *Pseudomonas putida*. Compound **223** showed pronounced antibacterial activity against *V. parahaemolyticus, V. anguillarum* and *P. putida*, with MIC values of 6.25, 1.56, and 3.13 μM, respectively. Compound **175** exhibited significant inhibitory activity against *V. anguillarum* and *P. putida,* with MIC values of 1.56 and 6.25 μM, respectively. Compound **1** displayed moderate activity against *P. putida* with a MIC value of 25.0 μM, whereas **29** showed activity against *V. parahaemolyticus* with a MIC value of 25.0 μM. The positive control, ciprofloxacin, showed MIC values of 2.50, 0.625, and 0.625 μM, against *V. parahaemolyticus*, *V. anguillarum,* and *P. putida*, respectively [43].

Compounds **1**, **40** (Figure 4) and **71** (Figure 6), isolated from a soft coral-associated fungus, *Trichoderma harzianum* (XS-20090075), selectively exhibited the growth of *S. aureus* with MIC values of 6.25, 25.0, and 25.0 μM, respectively [55].

Compound **6** (Figure 3), isolated from a marine sponge-associated fungus, *Eurotium chevalieri* MUT2316, showed inhibitory activity against four bacterial species including *Halomonas aquamarina* ATCC14400, *Polaribacter irgensii* ATCC700398, *Vibrio aesturianus* ATCC 35048, and *Pseudoalteromonas citrea* ATCC 29720 with low observable effect concentration (LOEC) values of 0.01, 1, 10, and 0.01 μg/mL, respectively [53].

Compounds **28** (Figure 4) and **137** (Figure 11), isolated from a mangrove endophytic fungus, *Penicillium citrinum* HL-5126, showed weak antibacterial activity against *S. aureus* ATC29213 with the same MIC values of 22.8 μM. Compound **137** also exhibited antibacterial activity against *V. parahaemolyticus* ATCC17802 with a MIC value of 10 μM. The positive control, ciprofloxacin, showed MIC values of 0.31 and 1.25 μM against *S. aureus* ATC29213 and *V. parahaemolyticus* ATCC17802, respectively [41].

Although **29** (Figure 4), isolated from the culture extract of a marine sponge-associated fungus, *Aspergillus candidus* KUFA0062, did not exhibit antibacterial activity against Gram-positive (*S. aureus* ATCC 29213, *E. faecalis* ATCC 29212, MRSA *S. aureus* 66/1, and VRE *E. faecalis* B3/101) and Gram-negative bacteria (*E. coli* ATCC 25922, *Pseudomonas aeruginosa* ATCC 27853, a colistin-resistant *E. coli* 1418/1 strain, and a clinical isolate ESBL *E. coli* SA/2), it induced a significant reduction in biofilm formation (67.7% of the control) in *E. coli* ATCC 25922 (the background absorbance was used as a control) [16].

Compounds **29**, **267**, **268**, and **269** (Figure 25), isolated from a red alga-associated fungal strain F-F-3C, showed antibacterial activity against pathogenic bacteria *E. coli* and *S. aureus* at a concentration of 50 μg/disk with inhibition zones ranging from 13 to 15.5 mm [44].

Compounds **33** (Figure 4), **171** (Figure 15), and **245** (Figure 24), isolated from the culture extract of a soft coral-associated fungus, *Alternaria* sp. ZJ-2008003, showed antibacterial activity against *E. coli*, *V. parahemolyticus*, and *Staphylococcus albus*. Compound **33** inhibited the growth of all three bacterial strains with MIC values of 2.30, 5.0, and 15 μM. The positive control, ciprofloxacin, displayed MIC values of 0.62, 0.16, and 0.31 μM. Compound **171** showed the same potency as ciprofloxacin against *E. coli* with MIC value of 0.62 μM, followed by *V. parahemolyticus* and *S. albus* (MIC values of 1.25 and 12 μM, respectively). Compound **245** inhibited antibacterial activity against *E. coli* and *V. parahaemolyticus* with the same MIC value of 2.5 μM, while no antibacterial activity against *S. albus* was recorded (IC_50_ > 20 μM) [49].

Compounds **33**, **170**, **172**, **173**, **174** (Figure 15), **214** (Figure 19), **245**, **254**–**256**, **258**, and **259** (Figure 25), isolated from a mangrove endophytic fungus, *Stemphylium* sp. 33231, were assayed for antibacterial activity against seven terrestrial pathogenic bacteria viz. *Micrococcus tetragenus* (ATCC13623), *E. coli* (ATCC 25922), *S. albus* (ATCC 8799), *Bacillus cereus* (ATCC 14579), *S. aureus* (ATCC 6538), *Kocuria rhizophila* (ATCC 9341), and *B. subtilis* (ATCC 6633). Compounds **173** and **214** selectively inhibited the growth of *E. coli* (MIC = 9.8 and 7.3 μM), while **245** selectively inhibited the growth of *S. albus* (MIC = 8.9 μM). Compounds **254**, **258**, and **259** also showed selective antibacterial activity against *B. cereus* with MIC values of 7.9, 8.3, and 8.1 μM, respectively. Compound **174** displayed better antibacterial activity against *E. coli*, *B. cereus*, *B. subtilis*, and *S. aureus* with the same MIC value of 3.9 μM, and against *M. tetragenus* with MIC value of 7.8 μM. Compound **170** had the same MIC value of 7.8 μM against *E. coli*, *S. aureus*, *K. rhizophila*, and *B. subtilis*, whereas **33**, **172**, **255**, and **256** inhibited the growth of *M. tetragenus*, *E. coli*, B. *cereus*, *S. aureus*, and *B. subtilis* with MIC values ranging from 2.07 to 10 μM. The positive control, ciprofloxacin, showed MIC = 0.3, 0.3, 0.6, 0.6, 0.16, 0.3, and 0.6 μM [50].

Compound **48** (Figure 5), isolated from the culture extract of a sea urchin-derived *Monodictys* sp., at a concentration of 2.5 μg/disk, inhibited the growth of *B. subtilis* and *E. coli*, with the inhibition zones of 7 and 8 mm, respectively [60].

Compounds **53** (Figure 5) and **252** (Figure 24), isolated from the culture extract of a marine sponge-associated fungus, *Curvularia lunata*, at a concentration of 5 μg/mL, inhibited the growth of *S. aureus* ATCC 25923, *E. coli* ATCC 25922 and *E. coli* HBI 101 in the agar plate diffusion assay with the same inhibition zones of 8.5, 9.0, and 8.0 mm, respectively. Both **53** and **252** were also active against *B. subtilis* 168, with MIC values of 7.5 and 8.0 mm, respectively [64].

Compounds **58** (Figure 5), **182**, **184**, **186**, **187** (Figure 16), **208**, **209** and **210** (Figure 19), isolated from a sea-anemone-associated fungus, *Nigrospora* sp., were tested for antibacterial activity against a panel of pathogenic bacteria including *B. subtilis*, *B. cereus*, *Micococcus luteus*, *M. tetragenus*, *S. aureus, S. albus*, *E. coli*, *V. anguillarum*, and *V. parahaemolyticus*. Although **58** did not show any antibacterial activity, its acylated derivative, **58d**, inhibited the growth of *B. subtilis*, *B. cereus*, *M. tetragenus*, *V. anguillarum*, and *V. parahemolyticus* with MIC values of 5.00, 37.5, 9.40, 4.70, and 75.0 μM, respectively. Compound **182** was active against *V. anguillarum* (MIC = 0.39 μM) (the positive control, ciprofloxacin, showed MIC = 0.0780 μM), whereas **182a** (Figure 16) displayed a pronounced inhibitory activity against *B. cereus* with MIC = 0.0488 μM (25 times more potent than ciprofloxacin whose MIC = 1.25 μM). Compound **184** exhibited potent antibacterial activity against *B. subtilis* and *B. cereus* with the same MIC values of 0.313 μM, respectively, which were comparable with the reference drug, ciprofloxacin, whose MIC values were 0.313 and 1.25 μM, respectively. Compounds **186**, **208** and **209** inhibited the growth of all the tested bacterial strains, with the exception of *M. luteus*, with MIC values ranging from 1.56–3.12 μM for **186**, and 0.780 to 25 μM for **208** and **209**. Compound **187** showed strong antibacterial activity against *B. subtilis* with an MIC value of 0.625 μM [68].

Compounds **63** and **65** (Figure 6), isolated from the culture extract of a marine plant-associated fungus, *Fusarium equiseti*, displayed antibacterial activity against *Pseudomonas syringae* pv. *lachrymans*, *Acidovorax avenae*, and *Erwinia carotovora.* While **63** showed strong inhibition against the three bacterial strains, with MIC values of 3.91, 3.91, and 7.81 μg/mL, **65** displayed weak activity with MIC values of 15.6, 15.6, and 7.81 μg/mL, respectively (the positive control, streptomycin, showed MIC values of 0.24, 0.98, and 0.98 μg/mL, respectively) [73].

Compound **70** (Figure 6), isolated from a green alga-associated fungus, *A. versicolor*, at a concentration of 50 μg/disk, inhibited the growth of *B. cereus*, *B. subtilis*, and *S. aureus* with the inhibition zones of 11, 12, and 14 mm, respectively (the reference drug, oxytetracycline, showed the inhibition zones of 17, 20, and 17 mm, at a concentration of 50 μg/disk) [34].

Compound **73** (Figure 6), isolated from a deep-sea sediment-derived *Emericella* SCISO 05240, showed moderate antibacterial activity against *E. coli* ATCC29922, *Klebsiella pneumonia* ATCC13883, *S. aureus* ATC29213, *E. faecalis* ATCC29212, *Acinetobacter baumannii* ATCC19606, and *Aeromonas hydrophila* ATCC7966 with inhibition zones ranging from 9 to 11 mm. The inhibition zone produced by the reference drug, ciprofloxacin, ranged between 35 and 40 mm [78].

Compounds **81**, **85** (Figure 7), **104** and **107** (Figure 9), isolated from a marine sponge-associated fungus, *A. versicolor*, were evaluated for antibacterial activity against clinically isolated Gram-positive strains viz. *Streptococcus pyogenes* 308A, *S. pyogenes* 77A, *S. aureus* SG511, *S. aureus* 285, and *S. aureus* 503. Compound **81** selectively inhibited the growth of *S. pyogenes* 308A with MIC = 6.25 μg/mL, while **104** displayed antibacterial activity against *S. pyogenes* 308A and *S. aureus* 503, with the same MIC value of 6.25 μg/mL. Conversely, **85** and **107** inhibited the growth of all strains, with MIC values ranging from 0.78 to 6.25 μg/mL. The positive control, meropenam, showed MIC values of 0.01, 0.01, 0.10, 0.10, and 0.05 μg/mL. Since **107** displayed stronger antibacterial activity than **104**, the OH-2′ group was suggested to play a key role in antibacterial activity in **107** [84].

Compounds **82** and **95** (Figure 7), isolated from a marine sponge-associated fungus, *Aspergillus* sp. F40, were evaluated for antibacterial activity against *S. aureus* ATCC25923 and *V. parahaemolyticus* ATCC17802. Compound **82** selectively inhibited the growth of *V. parahaemolyticus* with MIC value of 12 μg/mL, whereas **97** showed weak antibacterial activity with MIC values of 48 and 24 μg/mL, respectively. The positive control, tobramycin, displayed MIC values of 0.75 and 0.38 μg/mL, respectively [93].

Compounds **83**, **84** (Figure 7) and **113** (Figure 9), isolated from an algicolous fungus, *A. versicolor* EN-7, exhibited weak antibacterial activity against *E. coli*, at a concentration of 20 μg/disk, with inhibition zones of 7.0, 6.5, and 6.5 mm. Compound **113** also weakly inhibited the growth of *S. aureus* with an inhibition zone of 7.0 mm. The positive control, chloramphenicol, showed the inhibition zones of 25 and 22 mm, at a concentration of 20 μg/disk [82].

Compounds **102** and **103** (Figure 9), isolated from a mangrove endophytic fungus, *A. nidulans* MA-143, displayed the antibacterial activity against some human and aquatic pathogenic bacteria viz. *E. coli*, *M. luteus*, *V. vulnificus*, *V. anguillarum*, *V. alginolyticus*, *V. parahaemolyticus*, and *Edwardsiella ictaluri* with MIC values ranging from 1–64 μg/mL. Compound **102** showed potent antibacterial activity toward *V. alginolyticus* (MIC = 1 μg/mL), while **103** showed strong activity against *E. coli* and *V. parahaemolyticus,* with the same MIC value of 1 μg/mL. The positive control, chloramphenicol, showed MIC values of 1, 2, 8, 1, 0.5, 2, and 0.5 μg/mL [80].

Compounds **104** and **109** (Figure 9), isolated from a deep-sea sediment-derived *A. versicolorin* MF180151, displayed antibacterial activity against *S. aureus*, with MIC = 6.25 μg/mL. Compounds **104** and **109** also showed moderate activity against MRSA *S. aureus* with MIC = 25 and 12.5 μg/mL, respectively. The positive control, vancomycin, showed MIC = 1 μg/mL for both bacterial strains [101].

Compounds **104**, **106**, **107**, **110**, **117** and **118** (Figure 9), isolated from a gorgonian-associated fungus, *Aspergillus* sp., inhibited the growth of *S. albus* with MIC values ranging from 12.5–50 μM. Compounds **110** and **117** showed stronger antibacterial activity (MIC = 6.25 μM) than **106**, **107** and **104** (MIC values of 25, 25 and 50 μM, respectively) against *M. luteus*, suggesting that the methoxy group on C-8 might play an important role for this activity. The positive control, ciprofloxacin, showed MIC values of 3.13 and 0.780 μM, respectively [103].

Sharma et al. [129], in their search for novel potent inhibitor(s) against β-ketoacyl-ACP reductase (MabA) and polyketide synthase18 (PKS18), which are involved in mycolic acid biosynthesis in *Mycobacterium tuberculosis*, by virtual screening of anthraquinones from marine-derived fungi, have found that among 100 marine-derived anthraquinones retrieved from the PubChem database, only three fulfilled all ADMET (absorption, distribution, metabolism, excretion, and toxicity) descriptors after the filtering through Lipinski’s rule of five (for drug likeness) and in silico ADME/Tox analysis (for pharmacokinetic properties). Compound **104** showed the highest human intestinal absorption among all anthraquinones tested and the controls (isoniazid and ethambutol). Molecular docking studies using AutoDock 4.2 revealed that **104** showed the best docking conformation with binding affinities of −8.84 and −8.23 kcal/mol with MabA and PKS18, respectively, and Ki values of 1.79 and 3.12 μM, respectively. Further analysis of **104** to identify its best docking pose revealed three binding pockets and interacting residues of active sites in the respective pockets of MabA. Compound **104** showed interactions with amino acids Arg25, Gly28, Gly26, Met190, Thr191, Ile186, Pro183, Tyr185, Gly184, Gly139, Val141, Ser140, Tyr153, Asn88 in the first binding pocket and hydrogen bond formation with three amino acids, i.e., Ser140, Ile27, Thr188. In PKS18, **149** showed the interaction with Tyr188, His192, Gly136, Ser166, Gln255, Glu295, Phe253, Ser252, Ser251, Glu295, in the first binding pocket and established hydrogen bonds with five amino acid residues, i.e., Ser254, Gln255, Met296, Ser164, Asp299 [129].

Compounds **156**, **157** (Figure 13), and **231** (Figure 20), isolated from an algiclous fungus, *Eurotium cristatum* EN-220, were evaluated for their antibacterial activity. Compound **156** inhibited the growth of *E. coli* with MIC value of 32 μg/mL, while **157** was inactive, indicating that the methyl group at C-3 is essential for bioactivity in **156**. Compound **231** showed weak activity against *E. coli*, with MIC value of 64 μg/mL. The positive control, chloramphenicol, showed MIC value of 4 μg/mL [95].

Compound **126** (Figure 10), isolated from a sea fan-derived fungus, *Penicillium citrinum* PSU-F51, showed moderate antibacterial activity against *S. aureus* ATCC25923 and MRSA *S. aureus* SK1, with a MIC values of 16 μg/mL. Vincomycin was used as a positive control and showed MIC value of 1 μg/mL [37].

Compounds **170**, **178**, **179**, **180** (Figure 15) and **226** (Figure 19), isolated from the culture extract of a sea sediment-derived fungus, *Sporendonema casei* HDN16-802, displayed antibacterial activity against *Mycobacterium phlei*, *Proteus* sp., *B. subtilis*, *V. parahemolyticus*, and *P. aeruginosa*, with MIC values ranging from 12.5 to 200 μM (MIC values of the positive control, ciprofloxacin, ranged from 0.781 to 3.12 μM) [109].

Compound **177** (Figure 15), isolated from the culture extract of a mangrove endophytic fungus, *Phomopsis* sp. PSU-MA214, showed, in a colorimetric broth microdilution assay, moderate antibacterial activity against *S. aureus* ATCC25923 and methicillin-resistant *S. aureus* SK1, with MIC values of 128 and 64 μg/mL, respectively (the positive control; vancomycin, showed MIC value of 1 μg/mL) [52].

Compounds **182** and **186** (Figure 16), isolated from the mangrove endophytic fungus, *Nigrospora* sp. (strain no. 1403), showed strong antibacterial activity against *S. aureus* ATCC27154, *E. coli* ATCC25922, *P. aeroginosa* ATCC25668, *Sarcina ventriculi* ATCC29168, and *B. subtilis* ATCC6633, with the same MIC values of 3.13 μg/mL. The MIC value of the positive control, ampicillin, ranged from 3.1 to 50 μg/mL [112].

Compounds **184** and **186** (Figure 16), isolated from a mangrove endophytic fungus, *Nigrospora* sp., were tested for an in vitro anti-mycobacterial activity against various strains of *Mycobacterium*, such as *M. bovis* BCG (strain Pasteur, ATCC 35734*), M. tuberculosis* H37Rv reference strain (ATCC 27294), clinical multidrug-resistant (MDR) *M. tuberculosis* strain (K2903531, resistant to SM, INH, RFP and EMB), clinical MDR *M. tuberculosis* strain (0907961, resistant to SM and EMB), clinical drug-resistant *M. tuberculosis* strain (K0903557, resistant to INH), clinical drug-sensitive *M. tuberculosis* strain (0907762). Compound **186** displayed potent activity against two MDR *M. tuberculosis* clinical isolates, K2903531 and 0907961, and even better than that of the first line anti-tuberculosis agents. Moreover, treatment of *M. tuberculosis* H37Rv with **186** caused a significant difference of 119 genes, with 52 being significantly increased and 67 significantly decreased [130].

Compounds **184** (Figure 16), **208** and **209** (Figure 19), isolated from a sea fan-associated *Fusarium* sp. PSU-F14 and PSU-F135, exhibited a growth inhibition of *M. tuberculosis* H37Ra, with MIC values of 41, 87, and 38.57 μM, respectively (MIC values of the positive control, isoniazid, ranged from 0.17–0.34 μM) [59].

Compounds **197** and **198** (Figure 19), isolated from a coral-associated *Aspergillus* sp. strain 05F16, showed antibacterial activity against *S. aureus* IAM 12544T and *E. coli* IAM 12119T. Compound **197**, at a concentration of 100 μg/disc, weakly inhibited the growth of *S. aureus* IAM 12544T and *E. coli* IAM 12119T (inhibition zones = 15 and 9.2 mm, respectively), while **198** was only active against *S. aureus* (inhibition zone = 12 mm). It was suggested that the presence of the quinone core is necessary for the bioactivity [110].

Compounds **199**, **200**, **201**, **202** and **203** (Figure 19), isolated from the culture extract of an algicolous fungus, *Talaromyces islandicus* EN-501, showed pronounced antibacterial activity against *S. aureus* EMBLC-2 with MIC values ranging from 2 to 8 μg/mL. Compounds **200**–**203** showed weak inhibitory activity against *E. coli* EMBLC-1 and *E. tarda* QDIO-2, with MIC values ranging from 16 to 64 μg/mL. Compound **199** also exhibited weak activity against *E. coli* with MIC value of 64 μg/mL. The positive control, chloramphenicol, showed MIC = 2, 4, and 2 μM, against *S. aureus*, *E. coli*, and *E. tarda*, respectively [117].

Compounds **237** and **238** (Figure 23), isolated from a marine clam-associated fungus, *A. versicolor*, selectively inhibited *S. aureus* (inhibition zones 14 and 19 mm) at a concentration of 30 μg/well by a radial dilution assay. The positive control, tetracycline, displayed an inhibition zone of 30 mm at a concentration of 30 μg/well [88].

Compounds **243** and **244** (Figure 24), isolated from a mangrove endophytic fungus, *Altenaria* sp. (SK11), showed an inhibitory activity against *M. tuberculosis* protein tyrosine phosphatase B (MptpB), which is an essential virulence factor when *M. tuberculosis* hosts macrophages. Compound **244** (IC_50_ = 8.7 μM) was more active than **243** (IC_50_ = 64.7 μM). The IC_50_ value of the positive control, sodium orthovanadate, was 0.05 μM [62].

Compounds **279**, **283**, **285** and **286** (Figure 27), isolated from a marine sponge-associated fungus, *Engyodontium album* strain LF069, were examined against clinically relevant bacterial strains viz. *Staphylococcus epidermidis* DSM 20044, methicillin-resistant *S. aureus* (MRSA) DSM 18827, and *Propionibacterium acnes* DSM 1897. Compounds **279**–**282** (Figure 27) showed strong antibacterial activity against *S. epidermidis* and methicillin-resistant *S. aureus* (MRSA) with IC_50_ values of approximately 0.2 μM, which were 10 times more active than chloramphenicol (IC_50_ value of 1.8 and 2.9 μM, respectively), and against *P. acnes* with IC_50_ values of 11.0, 13.8, 14.1, and 11.7, respectively. Conversely, **283**, **285**, and **286** inhibited the growth of *S. epidermidis* and methicillin-resistant *S. aureus* (MRSA) with IC_50_ ranging from 1.80 to 6.77 μM [125].

### 4.2. Antifungal Activity

Compounds **33** (Figure 4), **56**, **57** (Figure 5), **212** and **214** (Figure 19), isolated from a mangrove endophytic fungus, *Phoma* sp. L28, showed an in vitro antifungal activity against *Fusarium oxysporum* Schlecht. f. sp. *lycopersici* (Sacc.) W.C. Snyder et H. N. Hansen*, F. graminearum* Schw, *Colletotrichum musae* (Berk. & M. A. Curtis) Art., *C. gloeosporioides* (Penz) Sacc., *Penicillium italicum* Wehme, and *Rhizoctonia solani* Kuhn. Compound **33** exhibited a broad-spectrum antifungal activity with MIC values of 3.75, 60, 30, 60, 100, and 60 μg/mL, respectively (the positive control, carbendazim; showed IC_50_ values of 6.25, 6.25, 6.25, 3.125, 6.25, and 12.5 μg/mL, respectively). Conversely, **56** and **57** only showed moderate to weak (MIC values ranging from 80 to 200 μg/mL) or no (MICs > 200 μg/mL) antifungal activity against all the assayed fungal strains, while **214** was inactive against all the tested pathogens with the exception of *P. italicum* (MIC = 80 μg/mL). Compound **212** moderately inhibited the growth of *P. italicum* and *R. solani* (MIC = 35 and 50 μg/mL) and weakly inhibited the growth of *F. graminearum* and *C. gloeosporioides* (MIC = 100 and 200 μg/mL) [67].

Compounds **63** and **65** (Figure 6), isolated from the culture extract of a marine plant-derived fungus, *Fusarium equiseti*, displayed a moderate inhibitory activity against *Pestallozzia theae* with MIC value of 31.3 μg/mL. The fungal inhibitor, carbendazim, showed MIC = 7.81 μg/mL [73].

Compound **182** (Figure 16), isolated from a mangrove endophytic fungus strain no. 1403, inhibited, in a yeast-based assay on *Saccharomyces cerevisiae*, cell proliferation through the cell cycle at G1 phase, leading to cell death in a time- and dose-dependent manner [111].

Compounds **182** and **186** (Figure 16), isolated from the mangrove endophytic fungus, *Nigrospora* sp. (strain no. 1403), moderately inhibited the growth of *Candida albicans* ATCC10231 with the same MIC values (12.5 μg/mL). The MIC value of the positive control, nystatin, was 1.56 μg/mL [112].

Compound **268** (Figure 25), isolated from an algicolous fungal strain F-F-3C, inhibited the growth of *Choanephora cucurbitarum* at a concentration of 50 μg/disk, with an inhibition zone of 11–12.5 mm [44].

Compounds **279**–**282** (Figure 27), isolated from a marine sponge-associated fungus, *Engyodontium album* strain LF069, exhibited weak antifungal activity against *C. albicans* and *Trichophyton rubrum* with IC_50_ values ranging from 4.1 to 13.5 μM. The IC_50_ values of positive controls, nystatin and clotrimazol, were 1.5 and 0.16 μM, respectively [125].

### 4.3. Antiviral Activity

Compounds **1** and **4** (Figure 3), isolated from a green alga-derived fungus, *Aspergillus versicolor*, showed inhibitory activity against hepatitis C virus (HCV) protease (HCV-PR) with IC_50_ values of 22.5 and 40.2 μg/mL, respectively. The positive control, HCV I2, showed an IC_50_ value of 1.5 μg/mL [34].

The fermentation extract of *Fusarium equiseti*, isolated from a brown alga, *Padina pavonica*, potently inhibited the HCV NS3-NS4A protease with an IC_50_ value of 27.0 μg/mL. Compounds **28** (Figure 4), isolated from this extract, also inhibited HCV NS3-NS4A protease with an IC_50_ value of 10.7 μg/mL, which was comparable to the positive control, HCV-I_2_ (IC_50_ = 1.5 μg/mL). Conversely, the co-isolate **29** was void of activity. It was suggested that the substituent CH_2_OH at C-3 is essential for the bioactivity of **28** [40].

Compounds **47** and **58** (Figure 5), isolated from a zoanthid-derived fungus, *Nigrospora* sp., were evaluated for antiviral activity. Compound **47** exhibited antiviral activity against respiratory syncytial virus (RSV) with an IC_50_ value of 74.0 μM, while **58** showed a moderate inhibitory activity against coxsackie virus (Cox-B3) with an IC_50_ value of 93.7 μM. The positive control, ribavirin, showed antiviral activity against RSV and Cox-B3 with IC_50_ values of 78.0 and 39.0 μM, respectively [69].

Compound **64** (Figure 6), isolated from a marine sponge-associated fungus, *Trichoderma* sp. strain SCSIO41004, showed significant antiviral activity against enterovirus 71 (EV71) on Vero cells by CCK-8 assay (IC_50_ value of 25.7 μM). The positive control, ribavirin, showed an IC_50_ value = 13.3 μM [56].

Compound **69** (Figure 6), isolated from the acidic fermentation extract of a mangrove sediment-derived fungus, *Penicillium* sp. OUCMDZ-4736, displayed anti-hepatitis B virus (HBV) activity by inhibiting the secretion of both HBeAg and HBsAg by HepG2.2.15 cells, in a dose-dependent manner. Compound **69** inhibited both HBeAg and HBsAg more efficiently than the positive control, 3TC [76].

Compounds **92**, **94** (Figure 7), and **125** (Figure 10), isolated from a deep-sea sediment-derived fungus, *Aspergillus versicolor* SCSIO 41502, showed antiviral activity toward HSV-1, in a plaque reduction assay, with half maximal effective concentration (EC_50_) values of 6.25, 3.12, and 4.68 μM, and 50% inhibitory concentration (CC_50_) values of 50.7, 65.1, and 108.6 μM, respectively. The corresponding IC_50_ and CC_50_ values of the positive control acyclovir, were 3.0 and >1000 μM, respectively [92].

Compounds **104** and **107** (Figure 9), isolated from the culture extract of a sea water-derived fungus, *Aspergillus niger* (MF-16), showed inhibitory activity against Tobacco Mosaic virus (TMV) replication at a concentration of 0.2 mg/mL (inhibition 58.1% and 64.9%, respectively), with EC_50_ values of 0.101 and 0.122 mg/mL, respectively[98].

Compounds **107** and **112** (Figure 9), isolated from the culture extract of a marine sponge-associated fungus, *A. versicolor*, reactivated the latent human immunodeficiency virus (HIV)-1 expression in an in vitro model of 2D10 cells, at a concentration of 10 μM, with reactivation of 39.1% and 43.3%, respectively. The positive control, prostratin, exhibited reactivation of 79.4% at a concentration of 2.5 μM [102].

Compound **216** (Figure 19), isolated from the culture extract of a marine alga-derived endophytic fugus, *Talaromyces islandicus* EN-501, significantly inhibited a replication of the porcine reproductive and respiratory syndrome virus (PRRSV), in a dose-dependent manner, with EC_50_ = 12.11 μM, CC_50_ = 395.31 μM, and selective index (SI) = 32.64. Further experiments revealed that **216** effectively inhibited virus entry, but did not block adsorption to the host cell surface [107].

### 4.4. Antiparasitic Activity

Compounds **182**, **184** (Figure 16), and **209** (Figure 19), isolated from a sea fan-associated fungus, *Fusarium* sp. PSU-F14 and PSU-F135, were assayed for antiparasitic activity against *Plasmodium falciparum* K1 by the microculture radioisotope technique. Compounds **182** (IC_50_ = 9.8 μM) and **184** (IC_50_ = 13 μM) showed better antimalarial activity than **209** (IC_50_ = 24.5 μM). The positive control, dihydroartemisinin, showed IC_50_ = 0.004 μM [59].

Compounds **192** and **193** (Figure 17), isolated from modified cultures of a mangrove sediment-derived fungus, *Aspergillus glaucus* HB 1-19, were examined for its activity against the pathogens of leishmaniasis and African sleeping sickness. Compound **192** showed no activity against *Leishmania major* (promastigote form) or *Trypanosoma cruzi* (IC_50s_ > 50 mm) but weak activity against *T. brucei brucei* and *L. donovani* (amastigote form) with IC_50_ values of 29 and 17 μM, respectively, while **193** had no activity against both parasites (IC_50_ > 50 μM) [115].

### 4.5. Cytotoxic Activity

Compounds **1** (Figure 3) and **164** (Figure 14), isolated from the culture extract of a mangrove endophytic fungus, *Eurotium rubrum*, were assayed for their cytotoxic activity against seven human tumor cell lines viz. breast adenocarcinoma (MCF-7), cholangiocarcinoma (SW1990), hepatoma (HepG2), non-small cell lung cancer (NCI-H460), hepatoma (SMMC7721), cervical cancer (Hela), and prostate cancer (Du145). Compound **1** showed selective cytotoxicity against DU145 (IC_50_ = 15 μg/mL), while **164** displayed selective cytotoxicity toward SW1990 (IC_50_ = 25 μg/mL) [47].

Compounds **1** (Figure 3), **28** (Figure 4), **53** (Figure 5) and **68** (Figure 6), isolated from a marine sediment-derived fungus, *Gliocladium catenulatum* T31, showed cytotoxicity against human leukemia cell line (K562) with IC_50_ values of 1.09, 1.24, 8.92, and 13.60 μmol/L, respectively [65].

Compounds **1**, **10**, **11** (Figure 3), **168** (Figure 15) and **181** (Figure 16), isolated from the culture extract of a soft coral-associated fungus, *Aspergillus tritici* SP2-8-1, were assayed against human cancer cell lines viz. HeLa, lung carcinoma (A549), and HepG2, using Cell Counting Kit-8 (CCK-8) assay. Compounds **1**, **168**, and **181** displayed cytotoxicity against HeLa, A549, and HepG2 cells with IC_50_ = 25.07, 22.17, and 30.20 μM; for **1**, IC_50_ = 10.57, 4.67, and 8.57 μM; for **168**, and IC_50_ = 2.67, 3.13, and 3.87 μM; for **181**, respectively. Compound **10** selectively inhibited the growth of A549 cells with IC_50_ value of 45.63 μM, while **11** selectively inhibited the growth of HepG2 with IC_50_ value of 42.07 μM. The positive control, doxorubicin, showed cytotoxicity with IC_50_ values of 0.5, 0.09, and 1.06 μM, respectively [28].

Compound **6** (Figure 3), isolated from the culture extract of a wild bivalve-derived fungus, *Penicillium* sp. ZZ901, displayed antiproliferative activity against glioma C6 and U78MG cells with IC_50_ values of 30.22 and 34.68 μM, respectively. The positive control, doxorubicin, showed IC_50_ values of 0.47 and 1.2 μM, respectively [27]. Compound **6**, isolated from a red alga-associated fungus, *Microsporum* sp., showed cytotoxic and anti-proliferative activities against HeLa cells through apoptosis. The Western blot analysis revealed that **6** downregulated Bcl-2 expression, upregulated Bax expression, and activated the caspase-3 enzyme [26].

Compounds **40** (Figure 4) and **71** (Figure 6), isolated from a marine coral-associated fungus, *Trichoderma harzianum*, showed cytotoxicity toward HepG2 cells, in a sulforhodamine B (SRB) assay, with IC_50_ values of 9.39 and 2.10 μM, respectively. Compound **71** also exhibited cytotoxicity against HeLa cells with an IC_50_ value of 8.59 μM [55].

Compound **46** (Figure 5), isolated from a decayed wood-derived fungus, *Halorosellinia* sp. (no. 1403), was tested against human nasopharyngeal epidermoid tumor (KB and KBv200) cell lines using a 2,5-diphenyl-2H-tetrazolium bromide (MTT) colorimetric method. Compound **46** showed remarkable cytotoxicity against both cell lines with IC_50_ values of 1.40 and 2.58 μg/mL, respectively [58]. Zhang et al., in the screening of 14 anticancer anthraquinone metabolites against KB and KBv-200 cell lines, have found that **46** was the most active anthraquinone that inhibited the growth of both cancer cell lines with IC_50_ values of 3.17 and 3.21 μM, respectively (IC_50_ values of the positive control, adriamycin, were 0.034 and 1.894 μM, respectively). The authors suggested that the mitochondrial dysfunction might be responsible for the apoptosis caused by this compound [131].

Compounds **47** (Figure 5), **182**, **184** (Figure 16), **208** and **209** (Figure 19), isolated from a sea fan-derived fungus, *Fusarium* sp. PSU-F14 and PSU-F135, were evaluated for cytotoxic activity against KB, MCF-7, and non-cancerous Vero (African green monkey kidney fibroblasts) cell lines. Compounds **182**, **184**, **208**, and **209** showed cytotoxicity against all the tested cell lines (IC_50_ = 0.9, 2.7, and 4.2 μM; for **182**, IC_50_ = 88, 5.4, and 29 μM; for **184**, IC_50_ = 19, 15, and 57 μM; for **208**, and IC_50_ = 49, 6.2, and 54 μM; for **209**, respectively), while **47** selectively inhibited MCF-7 cells (IC_50_ value of 6.3 μM). The positive control, doxorubicin, showed IC_50_ values of 0.33 and 2.18 μM against KB and KBMCF-7 cells, respectively, whereas the IC_50_ of ellipticine (the positive control for Vero cells) was 4.47 μM [59].

Compound **55** (Figure 5), isolated from a marine sediment-derived fungus, *Thermomyces lanuginosus* Tsikl. KMM 4681, displayed cytotoxic activity toward drug-resistant human prostate cancer, 22Rv1, cells. The cell viability, at a concentration of 100 μM for 48 h, was reduced by 35% following treatment with **55**. Compound **55** also did not show high cytotoxicity on human prostate non-cancer PNT-2 cells. Treatment with **55** suppressed the formation of a colony in prostate cancer 22Rv1 cells by 70% at the non-cytotoxic concentration of 50 μM [66].

Compound **77** (Figure 6), isolated from a mangrove endophytic fungus, *Fusarium* sp. ZZF60, exhibited cytotoxicity against human larynx carcinoma (Hep2) and HepG2 cancer cell lines, in a cell-based MTT assay, with IC_50_ values of 16 and 23 μmol/L, respectively [79].

Compounds **81**, **82**, **85** (Figure 7), **104** and **107** (Figure 9), isolated from a marine sponge-associated fungus, *A. versicolor*, were assayed against five human solid tumor cell lines, viz. A549 (lung), SK-OV-3 (ovarian), SK-MEL-2 (skin), XF498 (CNS), and HCT15 (colon). Compounds **81**, **85**, and **107** showed significant cytotoxic activity with IC_50_ values ranging from 0.41–3.88 μg/mL, while **82** and **104** exhibited weak cytotoxicity toward the assayed cell lines, with IC_50_ ranging from 15.29–23.73 μg/mL. The positive control, doxorubicin, showed IC_50_ values of 0.004, 0.019, 0.002, 0.01, and 0.034 μg/mL, respectively [84].

Compounds **82**, **85**, **99**, **104**, **107**, **109**, **118** and **119** (Figure 9), isolated from a marine-derived *Penicillium flavidorsum* SHK1-27, were examined for their antiproliferative activity against K562 cells by SRB method. Compound **107** was the most potent, followed by **82**, **104**, **109**, **85**, and **119** with IC_50_ values of 12.6, 27.7, 72.4, 91.0, 93.4, and 98.7 μM, respectively. Conversely, **99** and **118** weakly inhibited the proliferation of K562 cells with IC_50_ values> 100 μM [85].

Compounds **81**, **85**, **87**, **88**, **91** (Figure 7), **139–147** (Figure 11), isolated from a marine sediment-derived fungus, *Aspergillus* sp. SCSIO F063, were tested for their cytotoxic activity against three human tumor cell lines viz. SF-268 (brain), MCF-7, and NCI-H460 (non-small cell lung cancer) by SRB method. Compound **140** showed pronounced cytotoxicity against all tested cell lines, with IC_50_ values of 7.11, 6.64, and 7.42 μM, respectively, while **81**, **141**, and **147** showed moderate cytotoxicity against all the tested cell lines with IC_50_ values ranging from 18.91 to 44.22 μM. Compounds **85**, **87**, **139**, **143**, and **145** selectively inhibited the growth of MCF-7 cells, with IC_50_ values of 45.47, 29.69, 36.41, 49.53, and 24.38 μM, respectively. Compound **91** weakly inhibited the growth of SF-268 and MCF-7 cells with IC_50_ values of 47.19 and 40.47 μM, respectively. The positive control, cisplatin, showed IC_50_ values of 4.59, 10.23, and 1.56 μM, respectively [91].

Compounds **107** and **109** (Figure 9), isolated from a deep-sea sediment-derived fungus, *A. versicolor*, weakly inhibited A549 cell lines with IC_50_ values of 25.6 and 25.97 μM, respectively. Compound **107** also displayed a weak cytotoxicity against human ovary (A2780) cell line (IC_50_ value of 38.76 μmol/L) [89]. Compound **107**, isolated from a gorgonian-associated fungus, *Aspergillus* sp., showed significant growth inhibitory effects on K562 and HL-60 cell lines, in the MTT assay, with IC_50_ values of 0.87 and 1.46 μM, respectively [103]. Further in vitro antitumor activity investigation revealed that **107** caused a significant induction in cell cycle arrest at G_2_/M transition in K562 cell line in a concentration- and time-dependent manners (IC_50_ = 12.6 μM) [99].

Compound **128** (Figure 10), isolated from a sea fan-associated fungus, *Penicillium citrinum* PSU-F51, displayed mild cytotoxicity against KB cells with IC_50_ value of 30 μg/mL [37].

Compounds **170** and **172** (Figure 15), isolated from a soft coral-associated fungus, *Stemphylium lycopersici*, were assayed against HTC-116, MCF-7, and Huh7 cancer stem cell-like cells using CCK-8 assay. Compound **172** showed significant growth inhibitory activity against all tested cell lines with IC_50_ values of 1.3, 7.2, and 38.0 μM, respectively, while **170** exhibited cytotoxic affects toward HTC-116 and MCF-7 cancer cells with IC_50_ values of 3.5 and 9.0 μM, respectively. The positive control, adriamycin, showed IC_50_ values of 5.4, 6.2, and 15.4 μM, respectively, for HTC-116, MCF-7, and Huh7 cancer cells [51].

Compounds **171** (Figure 15), **245** and **248** (Figure 24), isolated from the culture extract of a soft coral-associated fungus, *Altenaria* sp., were evaluated for cytotoxic activity against human colon carcinoma (HCT-116), human breast cancer (MCF-7/ADR), human prostatic cancer (PC-3), and human hepatoma (HepG2 and Hep3B) cell lines, using MTT method. Compound **171** exhibited potent cytotoxicity against all the tested cell lines, with IC_50_ values of 2.2, 3.2, 7.6, 8.9, and 8.2 μM, respectively, while the IC_50_ values for **245** and **248** ranged from 24–98 and 6.4–23 μM, respectively. IC_50_ values of the positive control, epirubicin, were 0.82, 1.65, 0.46, 1.65, and 0.96 μM, respectively [49].

Compound **176** (Figure 15), isolated from a mangrove endophytic fungus, *Halorosellina* sp. no. 1403, showed a broad-spectrum anti-proliferative activity against six human cancer cell lines viz. breast cancer (MCF-7 and MDA-MB-435), prostate cancer (PC-3), glioma cancer (LN-444), and hepatoma cancer (Hep-3B and Huh-7), in a cell-based MTT assay, with IC_50_ values ranging from 3.0 to 9.6 μM. This compound also suppressed the growth of breast cancer xenografts in mice [108]. In order to investigate the mechanism underlying the anticancer activity, Chen et al. evaluated the effects of **176** on rat prolactinoma cell line, MMQ, using MTT assay, flow cytometry, real-time polymerase chain reaction (RT-PCR) and immunoblotting assays. Compound **176** inhibited cell growth of MMQ in a dose-dependent manner (IC_50_ = 13.2 mM) and displayed weak toxicity against rat pituitary cells (RPCs) with an IC_50_ value of 49.1 mM. The apoptotic cells in MMQ cells treated with **176** were enhanced through downregulation of miR-200c, and the expression level of prolactin (PRL) was inhibited without any changes in PRL mRNA levels [132]. Compound **176** also stimulated apoptosis in human nonfunctioning pituitary adenoma (NFPA) cells through the inhibition of the Akt pathway [133].

Compound **177** (Figure 15), isolated from the culture extract of a mangrove endophytic fungus, *Phomopsis* sp. PSU-MA214, selectively inhibited the growth of MCF-7 cells with an IC_50_ value of 27 μg/mL but was not cytotoxic to Vero cells. Doxorubicin (IC_50_ = 8.57 μg/mL) and tamoxifen (IC_50_ = 89.47 μg/mL) were used as positive controls [52].

Compounds **178** (Figure 15) and **226** (Figure 19), isolated from a sediment-derived fungus, *Sporendonema casei* HDN16-802, were assayed against ten human cancer cell lines, including HeLa, K562, HL-60 (leukemia), HCT-116 (colon), MGC-803 (gastric), HO8910 (ovarian), MDA-MB-231 (breast cancer), SH-SY5Y (neuroblastoma), PC-3, BEL-7402 (liver), and L-02 (human normal liver cell line) using MTT and SRB assays. Compounds **178** and **226** displayed moderate cytotoxicity toward all cancer cell lines tested, with IC_50_ values ranging from 4.5 to 22.9 μM. IC_50_ values of the positive control, doxorubicin, ranged from 0.1 to 1.0 μM [109].

Compounds **182** and **186** (Figure 16), isolated from a mangrove endophytic fungus, *Nigrospora* sp. (strain no. 1403), showed significant cytotoxicity against six human cancer cell lines, viz. A549, Hep-2, HepG2, KB, MCF-7, and MCF-7/Adr, with IC_50_ values ranging from 2.44 to 6.68 μg/mL [112]. In another research, **186**, isolated from the same marine-derived fungus, also exhibited the anticancer activity against MDA-MB-435, HepG2, and HCT-116 cancer cell lines, with IC_50_ values of 3.19, 9.99, and 5.69 μM, respectively. The positive control, epirubicin, showed IC_50_ values of 0.56, 0.96, and 0.48 μM, respectively [134].

Compounds **184**, **187** (Figure 16) and **209** (Figure 16), isolated from a sea anemone-associated fungus, *Nigrospora* sp. ZJ-2010006, as well as **182a** and **186a** (Figure 16), showed cytotoxic activity against A549 cells with IC_50_ values of 3.32, 4.56, 41.5, 2.72, and 5.25 μM, respectively. The positive control, mitomycin, showed IC_50_ = 3.00 μM [68].

Compound **189** (Figure 16), isolated from a mangrove endophytic fungus no. 1403, exhibited a potent cytotoxicity against KB and KBv200 cells in the MTT assay, with IC_50_ values of 19.66 and 19.27 μM, respectively. Compound **189** caused apoptosis in KB and KBv200 cells through non-related reactive oxygen species (ROS) generation in mitochondria and activation of caspase-8 in death receptor pathways [113].

Compound **190** (Figure 17), isolated from a marine sediment-derived fungus, *A. glaucus*, showed selective cytotoxicity against A-549 and BEL-7402 cell lines (by SRB method), as well as HL-60 and P388 (mouse lymphoma) cell lines (by MTT assay), with IC_50_ values of 0.13, 0.28, 7.5, and 35.0 μM, respectively [114].

Compounds **191**, **271** and **272** (Figure 26), isolated from a marine sediment-derived fungus, *A. glaucus*, were also assayed for cytotoxicity toward HL-60 and A-549 cell lines. Compound **191** displayed a potent cytotoxicity against both HL-60 and A-549 cell lines (IC_50_ values of 0.51 and 0.24 μM, respectively), while **271** (IC_50_ values against HL-60 and A-549 cell lines = 7.8 and 9.2 μM) and **272** (IC_50_ values against HL-60 and A-549 cell lines = 44.0 and 14.2 μM, respectively) were less cytotoxic. However, the *trans* congener (**271**) was more potent than the *cis* congener (**272**) [22].

Compounds **195** and **196** (Figure 18), isolated from a deep-sea sediment-derived fungus, *Eurotium* sp. SCSIO F452, showed moderate (IC_50_ = 12.5 and 15.0 μM, respectively) and weak (IC_50_ = 30.1 and 37.3 μM, respectively) cytotoxicity against SF-268 and HepG2 cancer cell lines [116].

Compound **200** (Figure 19), isolated from the culture extract of a red alga-derived fungus, *Talaromyces islandicus* EN-501, displayed weak cytotoxicity against sensitive (A2780) and cisplatin-resistant (A2780 CisR) human ovarian cancer cell lines, at a concentration of 100 μM [117].

Compounds **240** and **241** (Figure 24), isolated from a mangrove endophytic fungus, *Altenaria* sp. ZJ9-6B, displayed cytotoxicity against MDA-MB-435 (higher metastasizing cells) and MCF-7 (lower metastasizing cells) cell lines (by MTT assay), with IC_50_ values of 26.97 and 29.11 μM (for **240**), and 13.11 and 20.04 μM (for **241**), respectively [61]. Compound **241** was further investigated for its underlying mechanism for cytotoxicity in MCF-7 cells. It was found that **241** mainly induced cell necrosis, and only a portion of cells was in the state of apoptosis. Compound **241** also caused a significant increase in ROS production, a significant increase in intracellular calcium and alteration of cell morphology of the MCF-7 cells, which is characteristic of apoptosis [135].

Compounds **275**, **276**, and **277** (Figure 26), isolated from an algicolous fungus. *A. alliaceus*, were tested for cytotoxicity against HCT-116 and SK-Mel-5 (melanoma) cell lines. Compound **275** showed higher cytotoxicity toward HCT-116 and SK-Mel-5 cells (IC_50_ values = 9.0 and 11.0 μM, respectively) than **276** (IC_50_ values = 10.5 and 12.2 μM, respectively) and **277** (IC_50_ values = 13.7 and 19.7 μM, respectively) [77].

Compounds **279–283** (Figure 27), isolated from a marine sponge-associated fungus, *Engyodontium album* strain LF069, displayed weak cytotoxicity toward a mouse fibroblasts cell line (NIH-3T3) with IC_50_ values of 14.0, 11.0, 13.2, 14.4, and 34.3 μM, respectively. The IC_50_ of the positive control, tamoxifen citrate, was 16.5 µM [125].

Compounds **287**, **289**, **293** and **296** (Figure 27), isolated from the culture extract of an unidentified fungus of the order Hypocreales (MSX 17022), were assayed against MCF-7, H460, and SF268 (human astrocytoma) cancer cell lines. Compound **287** (IC_50_ values = 18.1, 13.6, and 21.4 μM, respectively) and **296** (IC_50_ values = 21.0, 10.9, and 16.1 μM, respectively) exhibited moderate cytotoxicity against MCF-7, H460, and SF268 cells. Compound **289** (IC_50_ values = 20.6 and 21.0 μM) and **293** (IC_50_ values = 14.0 and 21.4 μM, respectively) displayed moderate cytotoxicity against H460 and SF268 cells. The positive control, camptothecin, showed IC_50_ values of 0.06, 0.01, and 0.05 μM toward MCF-7, H460, and SF268 cells, respectively [127].

Compounds **104**, **106**, **109** (Figure 9) and **149** (Figure 11), isolated from a deep-sea sediment-derived fungus, *A. puniceus* SCSIO z021, showed toxicity against brine shrimps (*Artemia salina* larvae) with a lethal concentration 50% (LC_50_) values of 15, 21, 5.3, and 2.7 μM, respectively. Compound **109** also showed strong toxicity against Vero cells with a median toxic concentration (TC_50_) value of 4.3 μM [81].

Compound **158** (Figure 13), isolated from a mangrove endophytic fungus, *Stemphylium* sp. 33231, showed a moderate lethality effect in brine shrimp lethality assay, with LD_50_ = 10 μM [50].

Compound **231** (Figure 20), isolated from a brown alga-derived fungus, *Eurotium cristatum* EN-220, also displayed moderate cytotoxicity in brine shrimp lethality assay (41.4% rate) at a concentration of 10 μg/mL [95].

Compounds **6** (Figure 3) and **228** (Figure 20), isolated from a marine sponge-associated fungus, *Eurotium repens*, showed cytotoxicity against sex cells of the sea urchin (*Strongylocentrotus intermedius*) at a concentration of 25 and 10 μg/mL, respectively [24].

### 4.6. Enzyme Inhibitory Activity

#### 4.6.1. Inhibition of α-Glucosidase Activity

Compounds **1** and **2** (Figure 3), isolated from a deep-sea sediment-derived fungus, *Aspergillus flavipes* HN4-13, were assayed for α-glucosidase inhibitory activity, Compound **1** was a non-competitive α-glucosidase inhibitor, with a *Ki*/IC_50_ value of 0.79/19 μM, whereas **2** was void of activity [18].

Compound **65** (Figure 6), isolated from a mangrove endophytic fungus, *Cladosporium* sp. HNWSW-1, effectively inhibited α-glucosidase enzyme with an IC_50_ value of 49.3 μM, which was almost 5.5-fold more active than the positive control, acarbose (IC_50_ = 275.7 μM) [74].

#### 4.6.2. Inhibition of Trypsin Activity

Compounds **1** and **4** (Figure 3), isolated from a green alga-derived fungus, *A. versicolor*, displayed a non-competitive inhibitory activity against the human trypsin, with IC_50_ values of 450.5 and 50.1 μg/mL, respectively. The soybean trypsin–chymotrypsin was used as a positive control and showed trypsin inhibitory activity with an IC_50_ value of 0.01 μg/mL [34].

Compound **28** (Figure 4), isolated from the culture extract of an algicolous fungus, *Fusarium equiseti*, inhibited trypsin activity with an IC_50_ value of 48.5 μg/mL, which was comparable with the positive control (soybean trypsin-chemotrypsin inhibitor; T-I, IC_50_ = 0.01 μg/mL). Compound **29** (Figure 4), isolated from the same extract, did not show any inhibitory activity. Therefore, it was proposed that the CH_2_OH group at C-3 of the anthraquinone scaffold was essential for the bioactivity of **28** [40].

#### 4.6.3. Inhibition of Tyrosinase Activity

Compound **74** (Figure 6), isolated from a marine sponge-associated fungus, *Neosartorya spinosa* KUFA 1047, displayed weak antityrosinase activity, at the maximum concentration of 200 μM (% inhibition = 11.56%). The positive control, kojic acid, inhibited tyrosinase activity at 95.04% of the same maximum concentration [54].

#### 4.6.4. Inhibition of Indoleamine 2,3-dioxygenase (IDO1) Activity

Compounds **52** (Figure 5), **133**, **134** (Figure 10) and **233** (Figure 21), isolated from a deep-sea sediment-derived fungus, *Alternaria tenuissima* DFFSCS013, showed inhibitory activity against indoleamine 2,3-dioxygenase (IDO1). Compounds **52**, **133**, and **134** displayed a significant inhibition, with IC_50_ values of 1.7, 4.2, and 0.5 μM, respectively, while **233** showed weak inhibition with IC_50_ value of 32.3 μM. The positive control, NLG919, displayed IC_50_ = 0.08 μM [63].

#### 4.6.5. Inhibition of Protein Tyrosine Phosphatases and Protein Kinases Activity

Compounds **52** (Figure 5), **133** and **134** (Figure 10) were also assayed against five recombinant human protein tyrosine phosphatases (PTPs), viz. T cell protein tyrosine phosphatase (TCPTP), Src homology region 2 domain-containing phosphatase 1 (SHP1), Src homology region 2 domain-containing phosphatase 1 (SHP2), megakaryocyte protein tyrosine phosphatase 2 (MEG2), and protein tyrosine phosphatase 1B (PTP1B). Compound **52** inhibited all the tested PTPs, with IC_50_ values of 35.3, 34.3, 14.6, 29.6, and 2.1 μM, respectively. Compounds **133** and **134** selectively inhibited TCPTP, SHP1, and MEG2, with IC_50_ values ranging from 26.2 to 68.2 μM [63].

Compounds **60**–**62** (Figure 6), isolated from a marine sponge-associated fungus, *Microsphaeropsis* sp., displayed inhibitory activity against protein kinase C (PKC-ε), cyclin-dependent kinase 4 in complex with its activator cyclin D1 (CDK4/cyclin D1), and epidermal growth factor receptor (EGF-R), with IC_50_ values ranging from 18.5 to 54 μM [71,72].

Compounds **79**, **81**, **82** (Figure 7), **104**, **106**–**109** (Figure 9) and **149** (Figure 11), isolated from a deep-sea sediment-derived fungus, *A. puniceus* SCSIO z021, inhibited activities of seven PTPs, which are involved in cancer and type 2 diabetes, i.e., TCPTP, SHP1, SHP2, MEG2, PTP1B, CDC25B, and CD45. Compounds **81**, **104**, and **106** showed inhibitory activity against all the tested PTPs, with IC_50_ values ranging from 0.2 to 19 μM. Compounds **107** and **109** inhibited the activity of TCPTP, SHP1, MEG2, CDC25B, and CD45, with IC_50_ values ranging from 1.0 to 18 μM. Compound **79** exhibited inhibition of TCPTP, SHP1, SHP2, MEG2, CDC25B, and CD45, with IC_50_ values of 8.6, 19, 18, 1.9, 18, and 18 μM, respectively. While compound **108** inhibited the activity of SHP1, SHP2, MEG2, PTP1B, and CDC25B, with IC_50_ values of 5.4, 5.4, 2.2, 18, and 18 μM, respectively, **149** displayed inhibition of TCPTP, SHP1, SHP2, MEG2, and PTP1B with IC_50_ values of 4.8, 4.9, 13, 8.0, and 8.0 μM, respectively. Compound **82** selectively inhibited the activity of MEG2, with IC_50_ value of 6.9 μM. The positive control, Na_3_VO_4_, showed an inhibitory activity against TCPTP, SHP1, SHP2, MEG2, and PTP1B with IC_50_ values of 2.4, 4.4, 6.2, 3.2, and 1.6 μM, respectively, while the positive control, menadione AACQ, inhibited CDC25B and CD45 activities with IC_50_ values of 14 and 0.29 μM, respectively [81].

Compounds **192** and **193** (Figure 17), isolated from modified cultures of *A. glaucus*, showed an inhibitory activity of receptor tyrosine kinases (RTKs) viz. c-Met, Ron, and c-Src, with IC_50_ of 4.3, 7.5, and 7.5 μM (for **192**), and 1.8, 9.4, and 5.7 μM (for **193**), respectively [115].

#### 4.6.6. Inhibition of Acetylcholinesterase (AChE) Activity

Compounds **29**, **38**, **41** (Figure 4) and **71** (Figure 6), isolated from a soft coral-associated fungus, *Trichoderma harzianum*, exhibited weak anti-acetylcholinesterase (AChE) activity, by Ellman method, at a concentration of 100 μM [55], whereas **236** (Figure 22), isolated from the culture extract of a mangrove endophytic fungus, *A. terreus* (no. GX7-3B), showed stronger anti-AChE activity with IC_50_ = 6.71 μM. The IC_50_ value of the positive control, huperazine A, was 0.003 μM [123].

### 4.7. Anti-Inflammatory Activity

Compounds **4** and **5** (Figure 3), isolated from a marine sponge-associated fungus, *A. europaeus* WZXY-SX-4-1, were assayed for their anti-inflammatory activity. Compounds **4** and **5** were found to significantly downregulate Nuclear factor kappa B (NF-κB) in a human colon carcinoma cell line (SW480) induced by lipopolysaccharide (LPS) with the inhibitory rates of 75.9% and 73.1%, respectively, which were comparable with NF-κB inhibitor, MG132, (88.9% inhibition) [21].

Compound **86** (Figure 7), isolated from a marine-derived fungus, *Aspergillus* sp. SF6796, was assayed for its anti-neuroinflammatory activity. Compound **86** induced the expression of heme oxygenase (HO)-1 protein in BV2 microglial cells through activation of a nuclear transcription factor erythroid-2 related factor 2 (Nrf2), regulation of p38 mitogen-activated protein kinase, and phosphatidylinositol 3-kinase/protein kinase B signaling pathways. The pro-inflammatory mediators including nitric oxide (NO), prostaglandin E2, inducible nitric oxide synthase (iNOS), and cyclooxygenase-2 in LPS-stimulated BV2 microglial cells were also suppressed by treatment with **86**. [87].

Compounds **128**, **130**, **131** (Figure 10) and **150** (Figure 11), isolated from the fermentation extract of a marine sediment-derived fungus, *Penicillium* sp. SCSIO sof101, were evaluated for their abilities to inhibit interleukin 2 (IL-2) secretion by Jurkat cells. Compared with FK506 (the interleukin 2 inhibitor; IC_50_ = 5.8 μM), **128**, **131**, and **150** strongly inhibited the IL-2 secretion with IC_50_ values of 4.1, 5.4, and 5.1 μM, respectively, while **130** moderately inhibited the IL-2 secretion with IC_50_ = 12 μM [46].

Compounds **246** (Figure 24) and **262**–**265** (Figure 25), isolated from the culture extract of a marine-derived fungus, *Stemphylium* sp., were assayed for their anti-inflammatory capacity through suppression of LPS-induced NO production in RAW 264.7 mouse macrophages. Compounds **246** and **262**–**265** exhibited moderate anti-inflammatory activity with IC_50_ values of 10.7, 11.6, 16.1, 1.6, and 8.4 μM, respectively [124].

### 4.8. Anti-Obesity Activity

Compounds **1**, **15**, **17** (Figure 3), **28** (Figure 4) and **49** (Figure 5), isolated from the culture extract of a marine sponge-associated fungus, *Talaromyces stipitatus* KUFA 0207, were evaluated for their anti-obesity activity using the Zebrafish Nile red assay. However, only **15** and **28** exhibited a significant anti-obesity activity, reducing the stained lipids more than 60% and 90%, respectively, with IC_50_ values of 0.95 and 0.17 μM, respectively. The positive control, resveratrol, showed IC_50_ = 0.6 μM. Compound **1** caused death of all zebrafish larvae after 24 h of treatment [33].

### 4.9. Anticoagulant Activity

Compounds **179** and **180** (Figure 15), isolated from a marine sediment-derived fungus, *Sporendonema casei* HDN16-802, were assayed for anticoagulant activity and showed moderate inhibition of thrombin and Factor Xa, with inhibition ratios of 47.8% and 51.5%, respectively. The positive control, argatroban, showed an inhibition ratio of 65.0% [109].

### 4.10. Antiangiogenic Activity

Compounds **175** (Figure 15) and **223** (Figure 19), isolated from a sea cucumber-associated fungus, *Trichoderma* sp. (H-1), exhibited a weak antiangiogenic activity, with 23.80 and 24.60% inhibition of the growth of intersegmental vessels (ISV) of Zebrafish, respectively. The % inhibition of control (0.1% DMSO) was 25.80, and the positive control, PTK787 (0.5 µg/mL), was 0.2 [43].

### 4.11. Antifouling Activity

Compounds **1** (Figure 3), **28** (Figure 4) and **68** (Figure 6), isolated from the culture extract of a gorgonian coral-associated fungus, *Penicillium* sp. SCSGAF0023, showed significant antifouling activity against *Balanus amphitrite* larvae settlement, with EC_50_ values of 6.1, 17.9, and 13.7 μg/mL, respectively [39].

### 4.12. Algicidal Activity

Fengping et al. have investigated the algicidal activity of crude EtOAc extracts of 49 marine macroalgal endophytic fungi against red-tide phytoplanktons, i.e., *Alexandrium tamarense*, *Prorocentrum donghaiense*, *Heterosigma akashiwa*, and *Chattonella marina*, and have found that four fungal strains, including *Aspergillus wentii* (pt-1), *A. ustus* (cf-42), and *A. versicolor* (dl-29 and pt-20) potently inhibited algal growth. The secondary metabolites isolated from these fungi, including **12**, **13** and **14** (Figure 3) showed high 24 h inhibition rates against the red tide algae with EC_50(24-h)_ values ranging from 0.01–14.29 μg/mL. Compound **12** possessed the highest algicidal activity against *C. marina*, *H. akashiwa*, and *P. donghaiense* with EC_50(24-h)_ values of 0.17, 0.63, and 4.24 μg/mL, respectively. Compound **12** was also found to decrease chlorophyll *a* (Chl *a*) and superoxide dismutase (SOD) contents, while increasing soluble protein, malondialdehyde (MDA), and peroxidase contents, which decreases the photosynthesis process. Compound **13** showed the algicidal activity against *C. marina*, *A. tamarense*, and *H. akashiwa*, with EC_50(24-h)_ values of 0.44, 5.24, and 1.22 μg/mL, respectively [29].

Compound **36** (Figure 4), isolated from the culture extract of a marine sponge-associated fungus, *Eurotium chevalieri* MUT2316, inhibited the growth of two algae, including *Halamphora coffeaeformis* AC713 and *Phaeodactylum tricornutum* AC171 with low observable effect concentration (LOEC) values of 0.01 and 1 μg/mL, respectively. This compound also inhibited the adhesion of only two algae, viz. *Cylindrotheca closterrium* AC170 and *H. coffeaeformis* AC713, with LOEC values of 0.001 and 1 μg/mL, respectively [53].

### 4.13. Insecticidal Activity

Compounds **99**, **100** and **120** (Figure 9), isolated from a red alga-derived fungus, *Acremonium vitellinum*, possessed moderate inhibitory activity against third-instar larvae of Cotton bollworm (*Helicoverpa armigera*), with LC_50_ values of 0.87, 0.78, and 0.72 mg/mL, respectively. The positive control, matrine, showed LC_50_ = 0.29 mg/mL [104].

### 4.14. Antioxidant Activity

Compound **6** (Figure 3), isolated from an algicolous fungus, *Aspergillus wentii* EN-48, showed weak radical scavenging activity against 2,2-diphenyl-1-picryl-hydrazyl (DPPH^•^) radicals, with an IC_50_ value of 99.4 μg/mL. The positive control, butylated hydroxyl toluene (BHT), showed IC_50_ = 36.9 μg/mL [23].

Compound **16** (Figure 3), isolated from an algicolous fungus, *Chaetomium globosum*, showed moderate DPPH^•^ radical scavenging activity with IC_50_ value of 62 μg/mL. The positive control, BHT, showed IC_50_ value of 18 μg/mL [32].

Compounds **87** (Figure 7), **104**, **107**, **109** and **115** (Figure 9), isolated from a deep-sea sediment-derived fungus, *A. versicolor*, were assayed for antioxidant capacity by a Trolox equivalent antioxidant capacity (TEAC) assay. Compounds **87**, **104**, **107**, **109**, and **115** scavenged the 2,2-azino-bis-3-ethylbenzothiazoline-6-sulfonic acid radical cations (ABTS^•+^), which were approximately equivalent to that of trolox (1.0 mmol/L). These compounds were further evaluated for their capacity to regulate the nuclear factor E2-related factor 2 (Nrf2), a transcription factor that responds to oxidative stress by binding to the antioxidant response element (ARE) in the promoter of genes coding for antioxidant enzymes and proteins for glutathione synthesis, and its activity can be measured by ARE-driven luciferase reporters using HepG2C8 cells, stably transfected with AREluciferase reporter plasmids. Compounds **87**, **104**, **107**, **109**, and **115**, at a concentration of 10 μmol/L, caused significant induction of luciferase 1.41−1.58-folds more than that of the blank control (DMSO), and approximately half of the positive control, tBHQ (tertiary butylhydroquinone), at a concentration of 50 μmol/L [89].

Compound **97** (Figure 8), isolated from a marine sponge-associated fungus, *A. europaeus* WZXY-SX-4-1, scavenged DPPH^•^ radicals with IC_50_ value of 13.2 μg/mL. The positive control, trolox, quenched DPPH^•^ radicals with IC_50_ value of 5.4 μg/mL [21].

The antioxidant activity of (±)-**122**, (+)-**A** (**122**), (−)-**A** (**122**), (±)-(**123**), (±)-(**124**), and (±)-(**125**) (Figure 10), isolated from a deep-sea sediment-derived fungus, *A. versicolor* SCSIO 41502, were assayed for their antioxidant activity against ABTS^•+^ radical cations. Compounds **122**–**125** showed TEAC values of 2.11, 2.07, 2.00, 2.27, 2.18, and 2.03 mmol/g, respectively. These results indicated that the configuration of the stereogenic carbon in **122** (Figure 10) did not influence its antioxidant activity [92].

Compounds **2** (Figure 3), **155** (Figure 12), **165**–**167** (Figure 14), **228** and **231** (Figure 20), isolated from a mangrove endophytic fungus, *Eurotium rubrum*, were examined for their DPPH^•^ radical scavenging capacity. Compounds **166** and **231** displayed moderate to potent scavenging activity, with IC_50_ values of 74.0 to 44.0 μM, while **2**, **155**, **165**, **167**, and **228** exhibited weak activity. The positive control, BHT, showed IC_50_ = 82.6 μM [106].

Compound (±)-**194** (Figure 18), isolated from a deep-sea sediment-derived fungus, *Eurotium* sp. SCSIO F452, showed DPPH^•^ radicals scavenging activity, with an IC_50_ value of 58.4 μM, while the pure (−)-**194** scavenged DPPH^•^ radicals with IC_50_ = 159.2 μM. Ascorbic acid was used as a positive control and showed IC_50_ = 45.8 μM [116].

Compounds **199**–**203** (Figure 19), isolated from the culture extract of an algicolous fungus, *Talaromyces islandicus* EN-501, scavenged DPPH^•^ radicals with IC_50_ values ranging from 12 to 52 μg/mL, which were better t=han the reference compound, BHT, whose IC_50_ = 61 μg/mL. Compounds **199**–**203** also showed moderate scavenging activity toward ABTS^•+^ radical cations, with IC_50_ values ranging from 8.3 to 34 μM, which were comparable to ascorbic acid (positive control) whose IC_50_ = 16 μM [117].

### 4.15. Other Biological Activities

Using calcium imaging assay, **233** (Figure 21), isolated from a deep-sea sediment-derived fungus, *Altenaria tenuissima* DFFSCS013, effectively stimulated intracellular levels of calcium flux in HEK293 (human embryonic kidney) cells, at a concentration of 10 μM, in the calcium imaging assay. However, **233** did not show any effect at a concentration less than 10 μM [63].

Compounds **287**, **289**, **293** and **296** (Figure 27), isolated from an unidentified fungus of the order Hypocreales (MSX 17022), displayed the 20S proteasome inhibitory activity at a concentration of 20 μg/mL (% inhibition ranging from 13% to 67%) [127].

In order to enhance a readability of this review, we have summarized the anthraquinoid metabolites and their derivatives, obtained from the marine environment in Table 1. This includes the names and numbers of the isolated compounds, the names of fungal producers, the sources from which the fungi were obtained, the reported biological/pharmacological activities and the references.

## 5. Concluding Remarks and Future Perspectives

This review shows that polyketides are the predominant metabolites reported from marine-derived fungi. Altogether, we have reported 296 specialized metabolites belonging to the anthraquinone class and their derivatives, which were isolated from 28 marine fungal strains, and less-studied fungal species highlighting the chemical diversity and their myriad biological/pharmacological properties. In general, these compounds exhibited a wide range of biological activities, including antibacterial and antibiofilm formation, antifungal, antiviral, antiparasitic, anti-inflammatory, enzyme inhibitory, antioxidant, anticoagulant, anti-angiogenesis, anti-obesity, anti-fouling, algicidal, insecticide and cytotoxic activities. More specifically, members of the genera *Aspergillus*, *Penicillium*, *Eurotium*, and *Fusarium* are the most prolific sources of anthraquinones and their derivatives. Among the isolated anthraquinones, 112 were from *Aspergillus*, 37 from *Penicillium*, 36 from *Altenaria*, 26 from *Stemphylium*, 23 from *Eurotium*, 19 from *Fusarium*, 14 from *Trichoderma*, 13 from *Acremonium*, 11 from *Talaromyces*, 10 from *Nigrospora*, and the rest of anthraquinones are from other fungal resources (Figure 28). Members of the genera *Aspergillus* and *Penicillium* are found to be more versatile in terms of secondary metabolite biosynthesis, producing various types of anthraquinones viz. hydro-, alkylated, halogenated, *seco*-, furano and pyrano derivatives. Sulphated anthraquinoids and anthraquinones fused with xanthones and chromones have been also reported in species of *Penicillium*, while the glycosylated anthraquinones were reported from algicolous and mangrove endophytic fungi of the genera *Fusarium* and *Stemphylium*, which have a close symbiotic relationship with the hosts, indicating that they can adjust the biosynthetic pathways to each other. Bianthraquinones are found predominantly in *Altenaria* and *Stemphylium* species, while the anthraquinone–xanthones are more preponderant in *Acremonium* and *Engyodontium* species, suggesting the species-specific metabolites. Another interesting observation is the elasticity of the biosynthetic capacity of fungi, for instance, cytoskyrin anthraquinone has been reported from the fungus *Curvularia* sp., which is associated with sponges. The influence of the fungal habitats, the organisms with which they are associated, the type of culture media and biotic and abiotic stressors can influence their capacity to biosynthesize a myriad of specialized metabolites with unique structural features, which ultimately can manifest different biological/pharmacological activities. The advantage of fungi in terms of secondary metabolite production over other organisms is their capacity to produce a large quantity of interesting compounds by fermentation. These compounds can be used as a scaffold for medicinal chemistry study. Given a versatility of the anthraquinoid scaffolds for their biological activities, it is legitimate to think that varying the side chains of the anthraquinoid scaffolds could render compounds with unique structures and efficient biological/pharmacological activities. Therefore, searching for marine-derived fungi from different niches, with different pressure, temperature and light intensity such as from thermal vent, deep-sea, polar habitats, and different animal hosts, can be promising to find structurally unique and biologically relevant compounds. Another perspective is the development of new culture media, which can allow for unculturable marine-derived fungi, which do not grow in normal media to thrive. In addition, taking advantage of the plasticity of the enzymology of the biosynthetic pathways of fungi, the addition of natural or synthetic amino acids to the culture media should be another challenging avenue to obtain compounds of unknown values.

## Figures and Tables

**Figure 1 marinedrugs-20-00474-f001:**
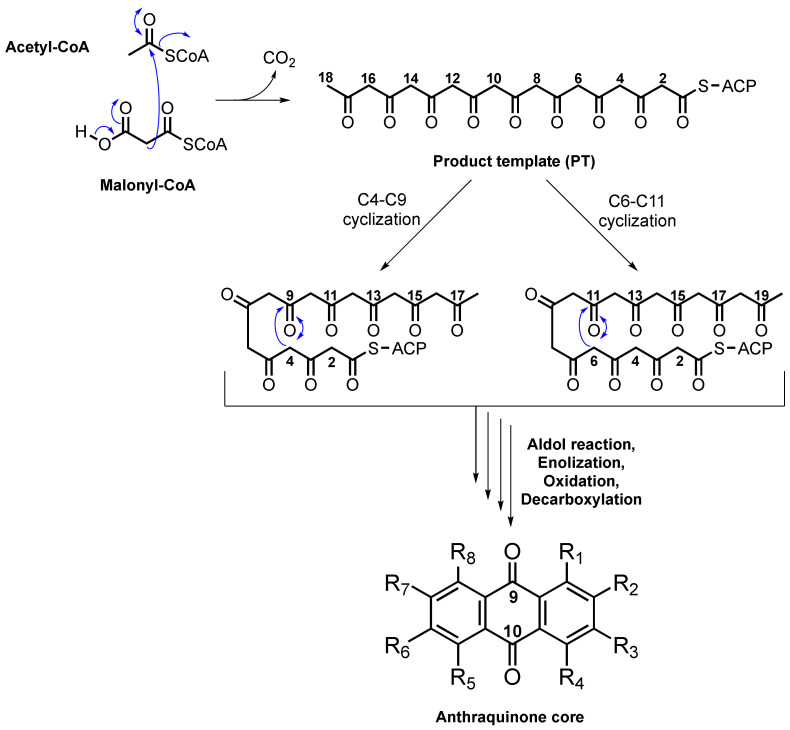
Plausible biosynthetic pathways of fungal anthraquinones.

**Figure 2 marinedrugs-20-00474-f002:**
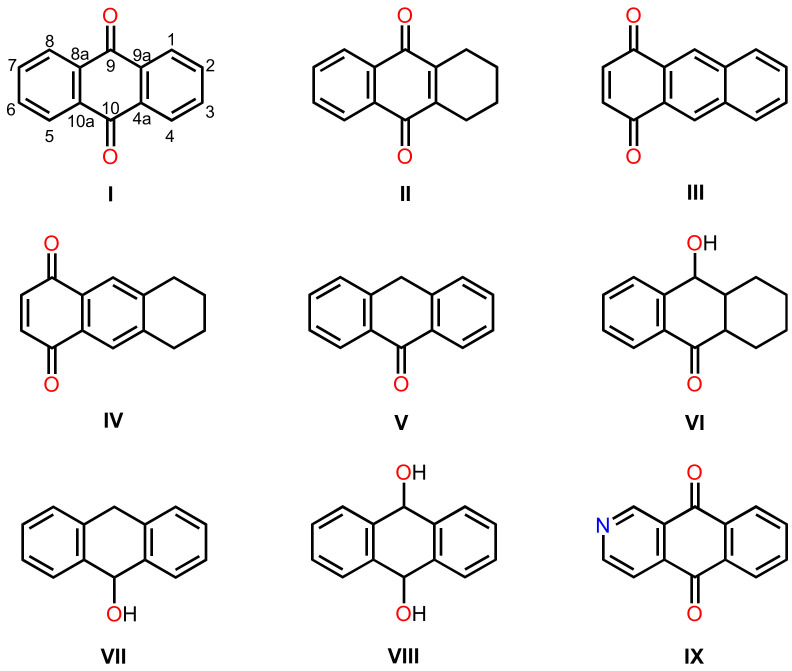
Anthraquinone scaffolds reported from marine-derived fungi.

**Figure 3 marinedrugs-20-00474-f003:**
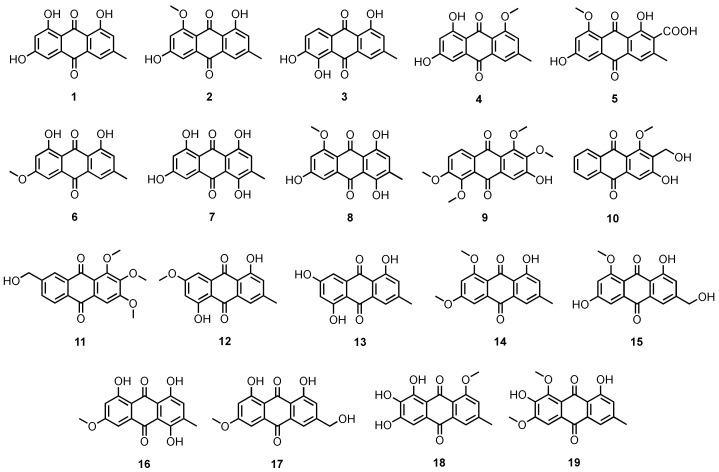
Structures of **1**–**19**.

**Figure 4 marinedrugs-20-00474-f004:**
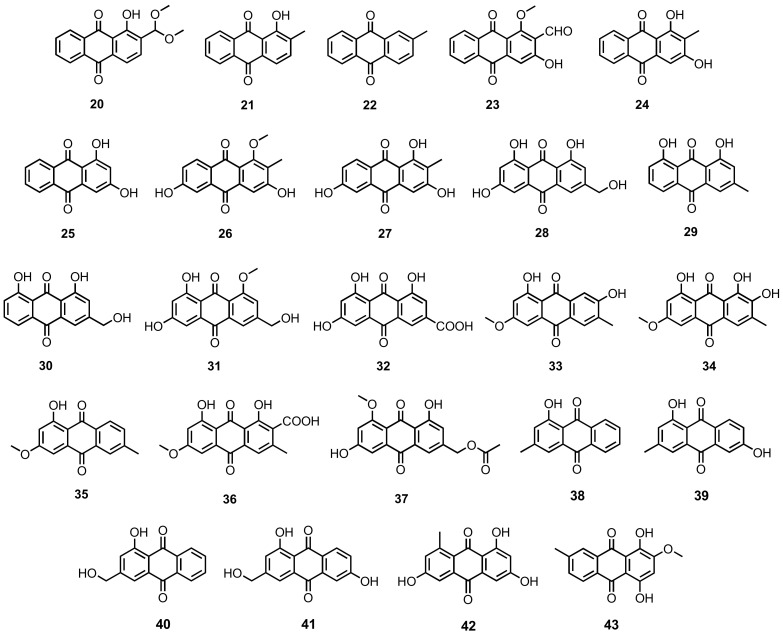
Structures of **20**–**43**.

**Figure 5 marinedrugs-20-00474-f005:**
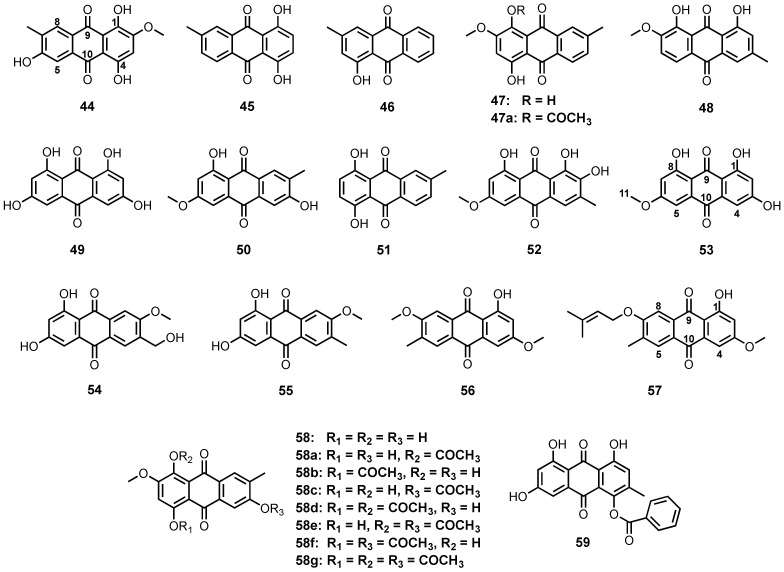
Structures of **44**–**59**.

**Figure 6 marinedrugs-20-00474-f006:**
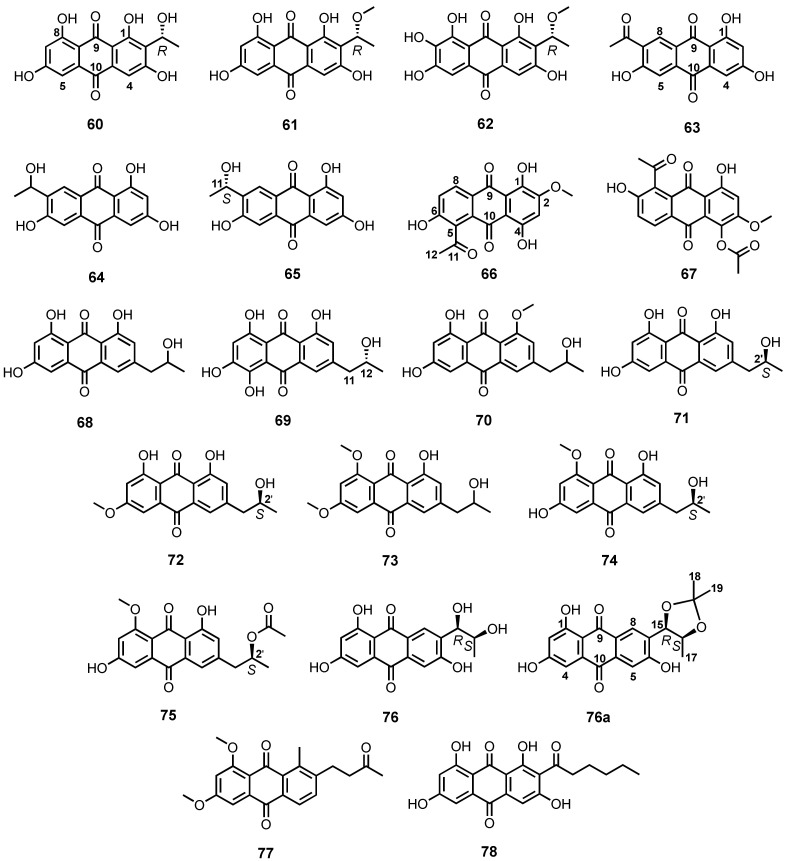
Structures of **60**–**78**.

**Figure 7 marinedrugs-20-00474-f007:**
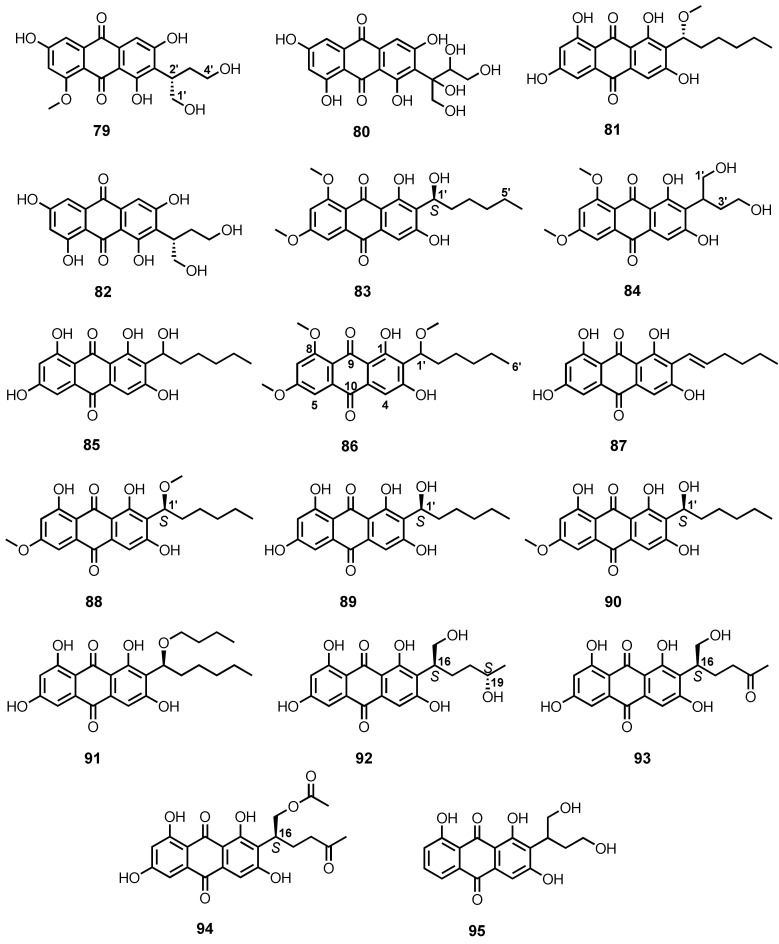
Structures of **79**–**95**.

**Figure 8 marinedrugs-20-00474-f008:**
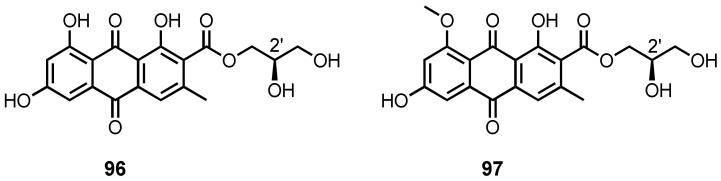
Structures of **96** and **97**.

**Figure 9 marinedrugs-20-00474-f009:**
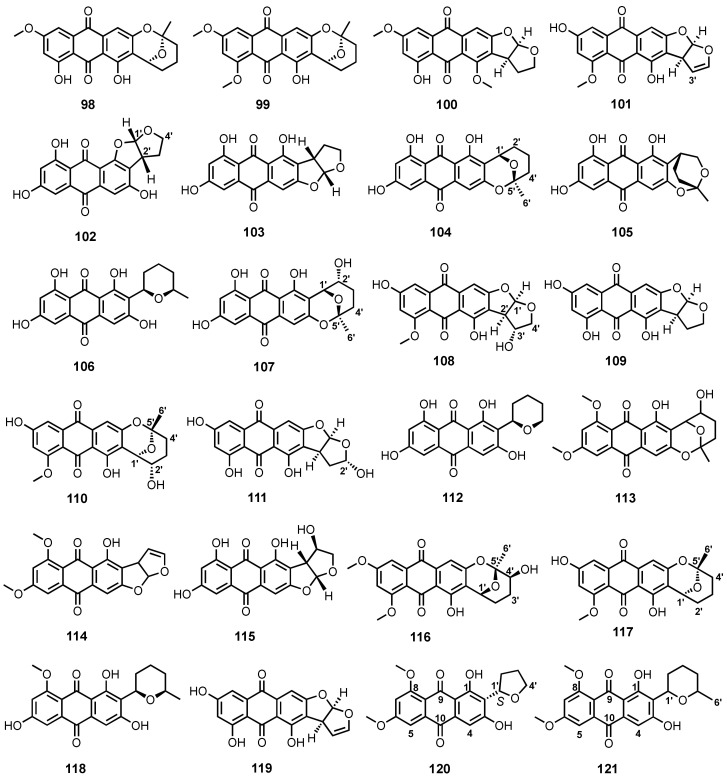
Structures of **98**–**121**.

**Figure 10 marinedrugs-20-00474-f010:**
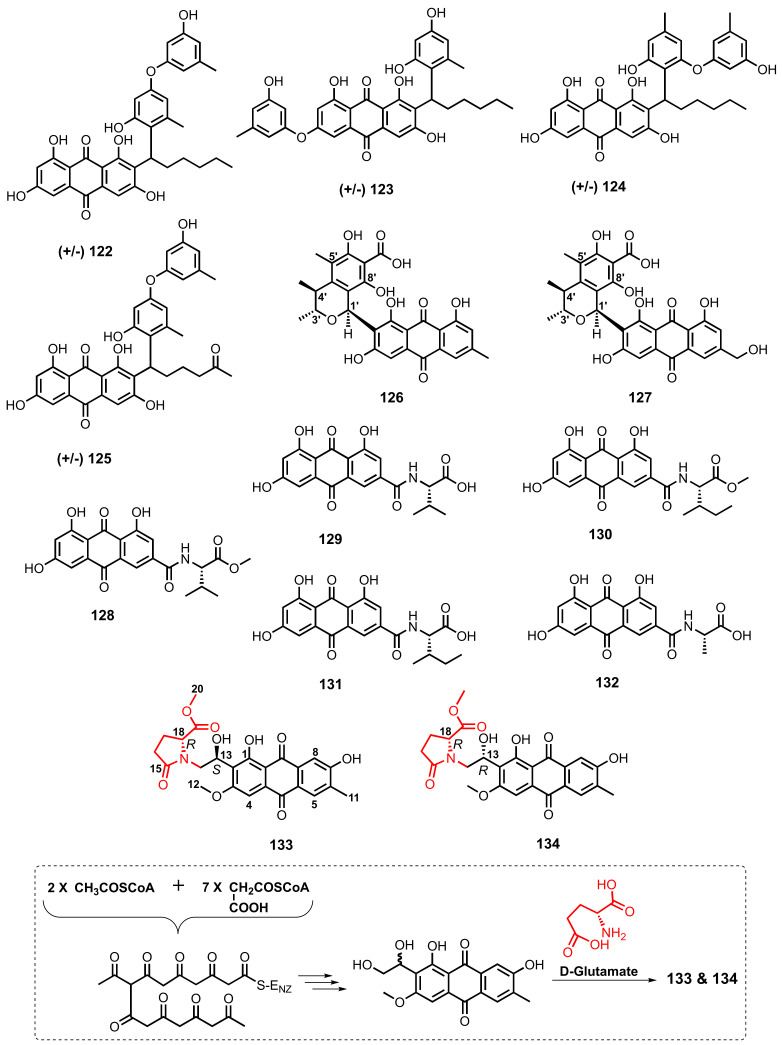
Structures of **122**–**134** and a plausible biosynthesis of **133** and **134**.

**Figure 11 marinedrugs-20-00474-f011:**
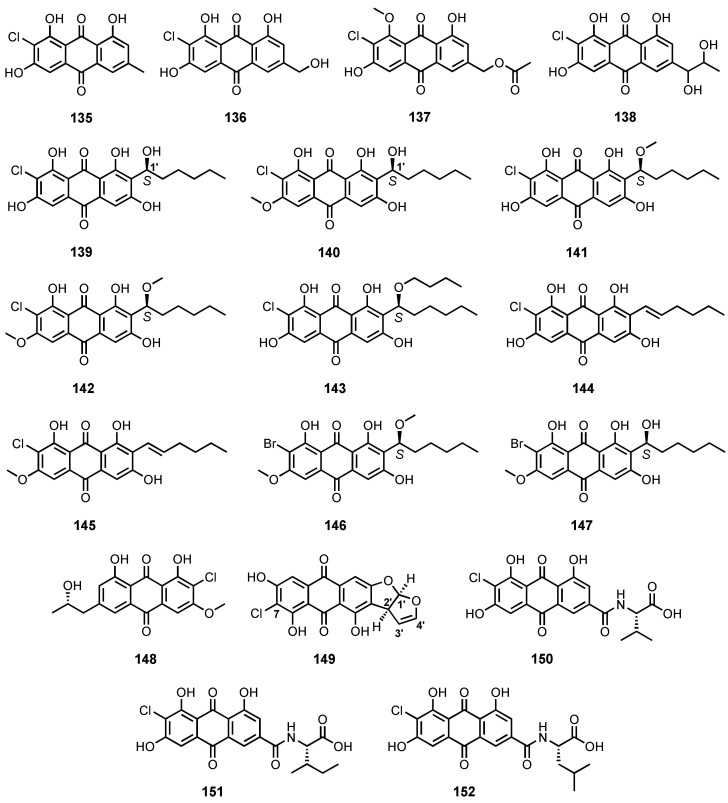
Structures of **135**–**152**.

**Figure 12 marinedrugs-20-00474-f012:**
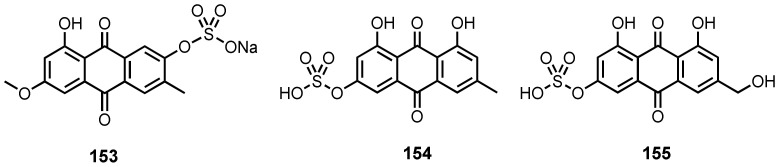
Structures of **153**–**155**.

**Figure 13 marinedrugs-20-00474-f013:**
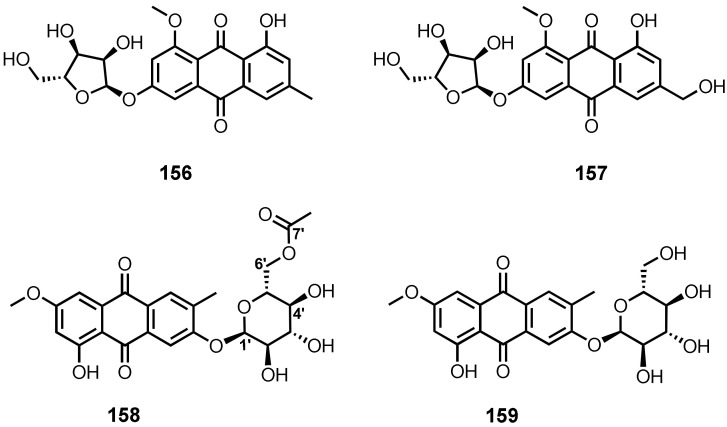
Structures of **156**–**159**.

**Figure 14 marinedrugs-20-00474-f014:**
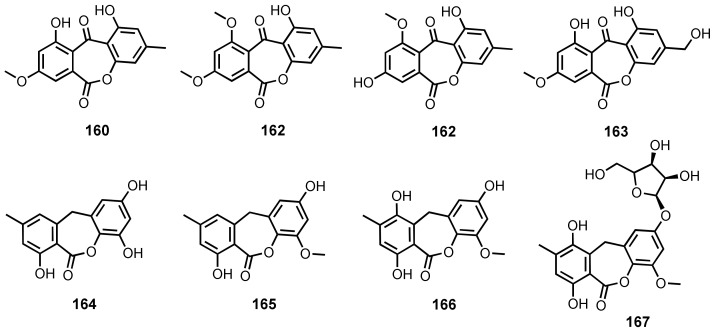
Structures of **160**–**167**.

**Figure 15 marinedrugs-20-00474-f015:**
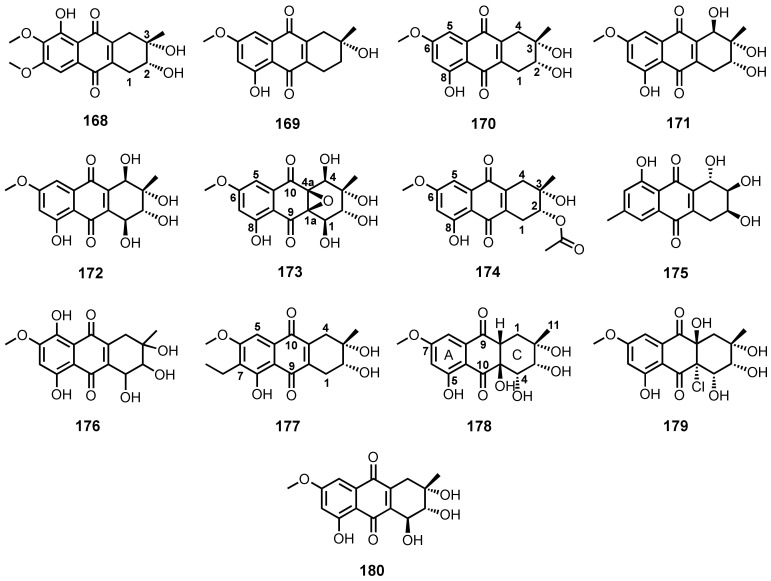
Structures of **168**–**180**.

**Figure 16 marinedrugs-20-00474-f016:**
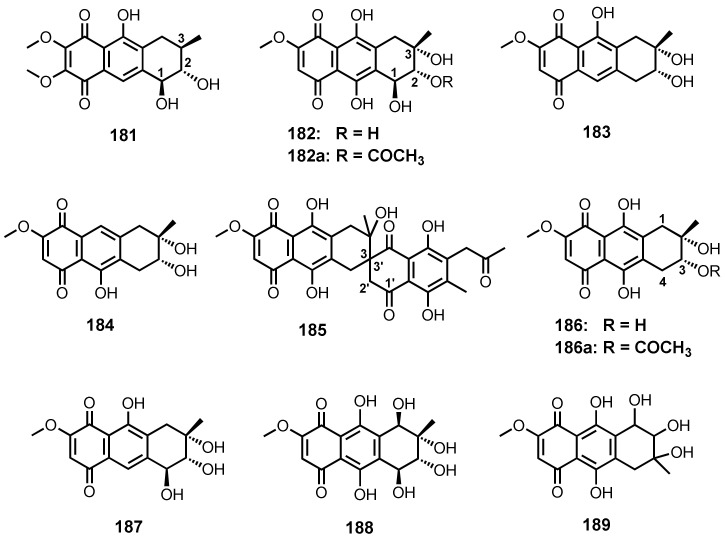
Structures of **181**–**189**.

**Figure 17 marinedrugs-20-00474-f017:**
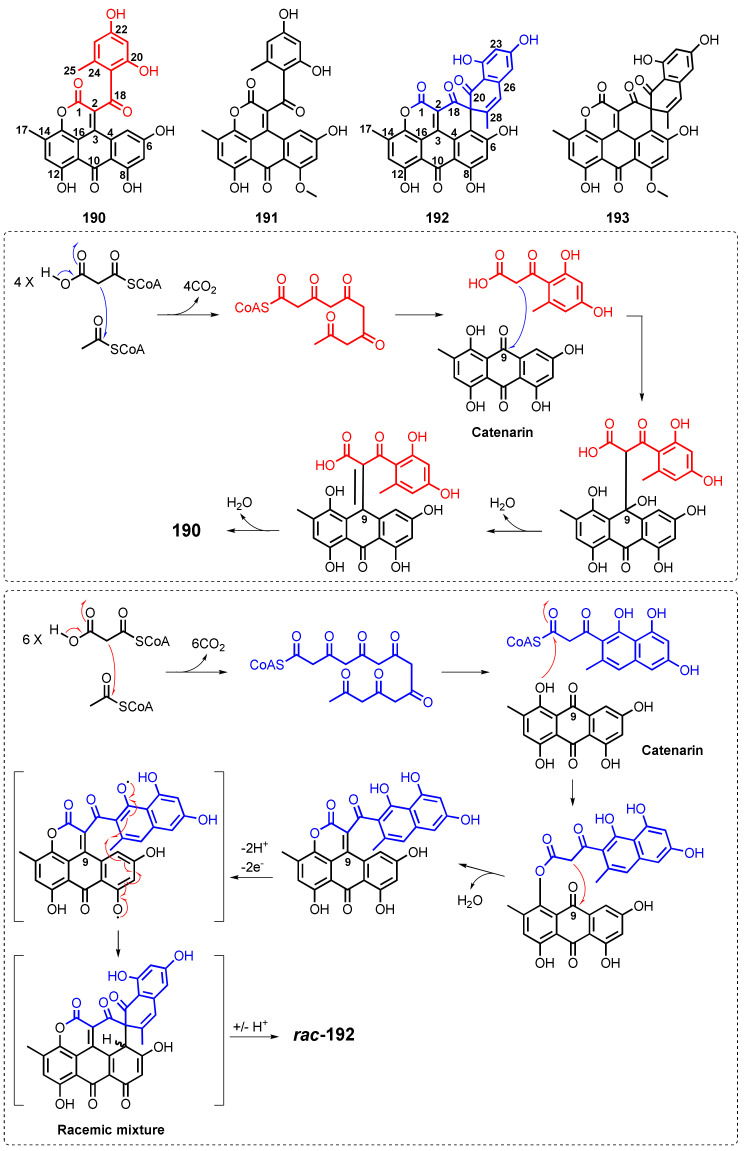
Structures of **190**–**193** and plausible biosynthetic pathways of **190** and **192**.

**Figure 18 marinedrugs-20-00474-f018:**
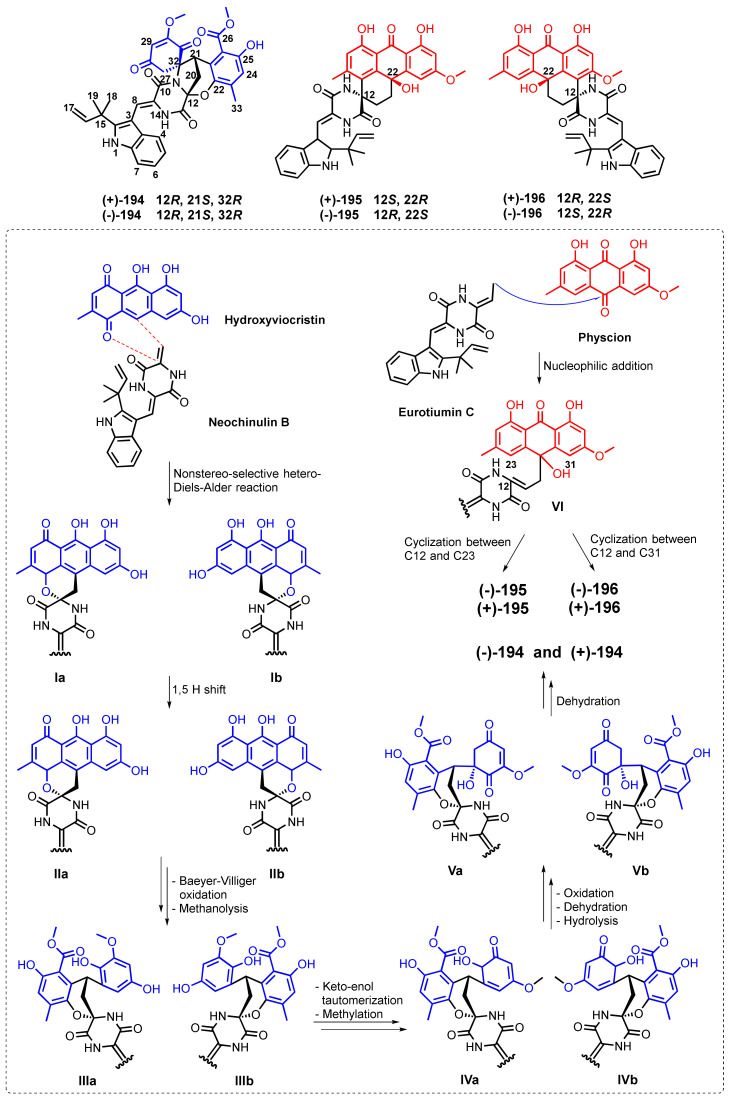
Structures of **194**–**196** and plausible biosynthetic pathways of **194**–**196**.

**Figure 19 marinedrugs-20-00474-f019:**
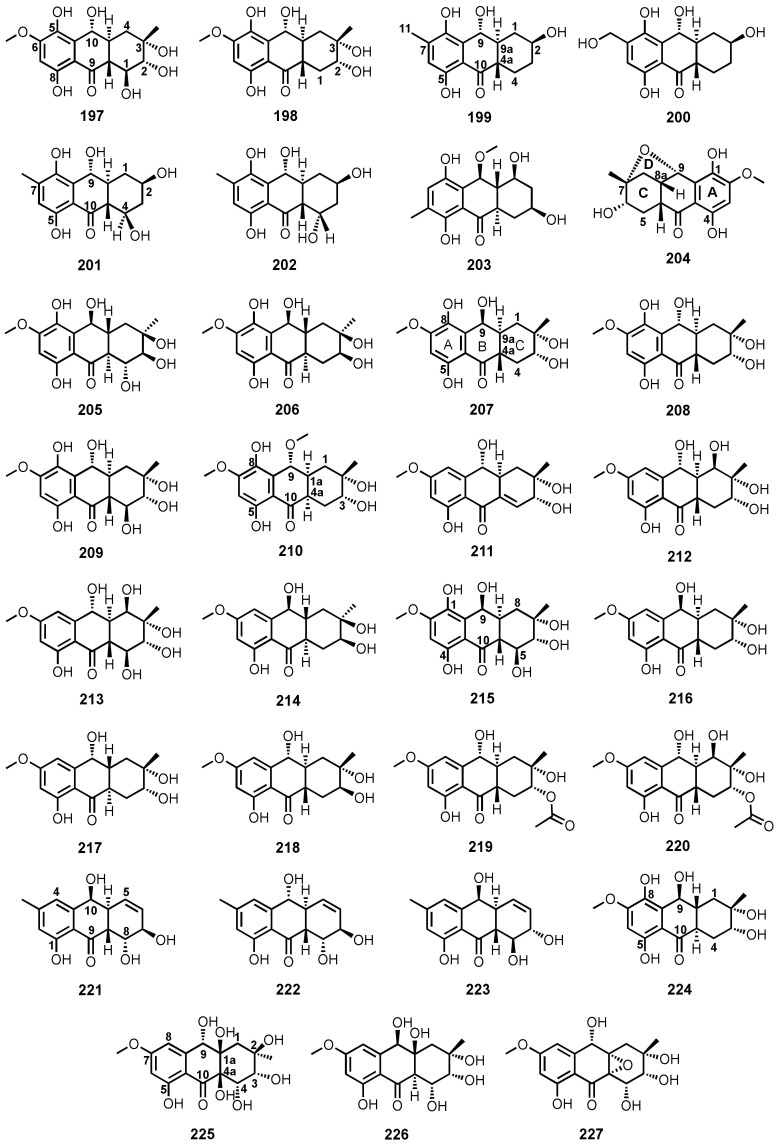
Structures of **197**–**227**.

**Figure 20 marinedrugs-20-00474-f020:**
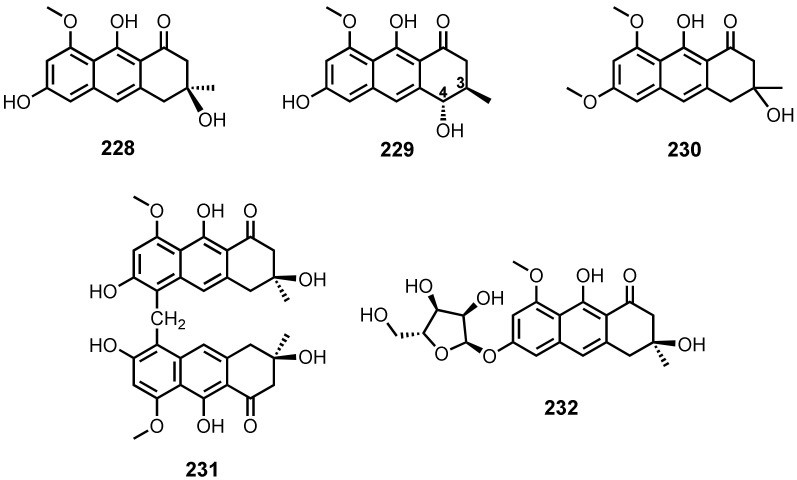
Structures of **228**–**232**.

**Figure 21 marinedrugs-20-00474-f021:**
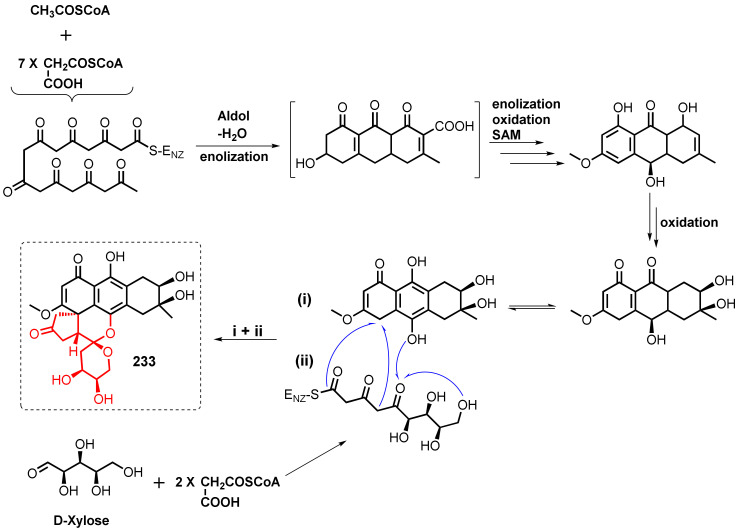
Plausible biosynthetic pathway for **233**.

**Figure 22 marinedrugs-20-00474-f022:**
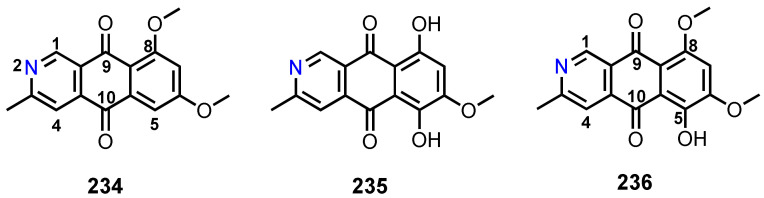
Structures of **234**–**236**.

**Figure 23 marinedrugs-20-00474-f023:**
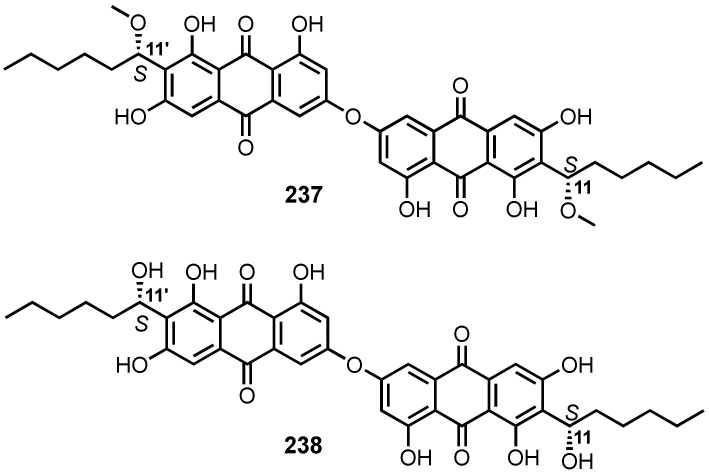
Structures of **237** and **238**.

**Figure 24 marinedrugs-20-00474-f024:**
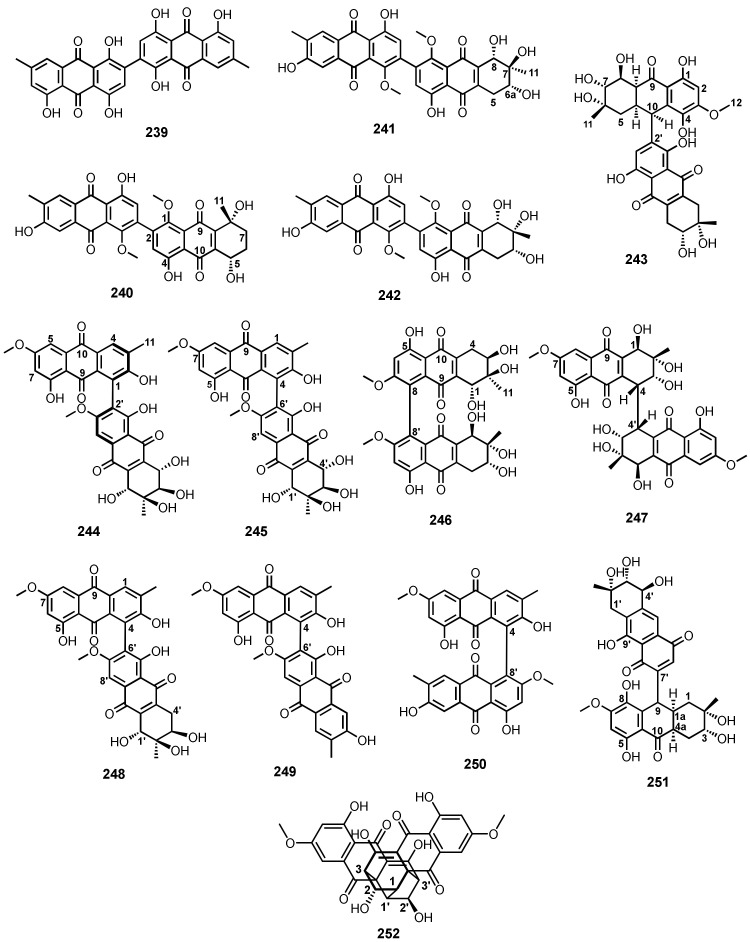
Structures of **239**–**252**.

**Figure 25 marinedrugs-20-00474-f025:**
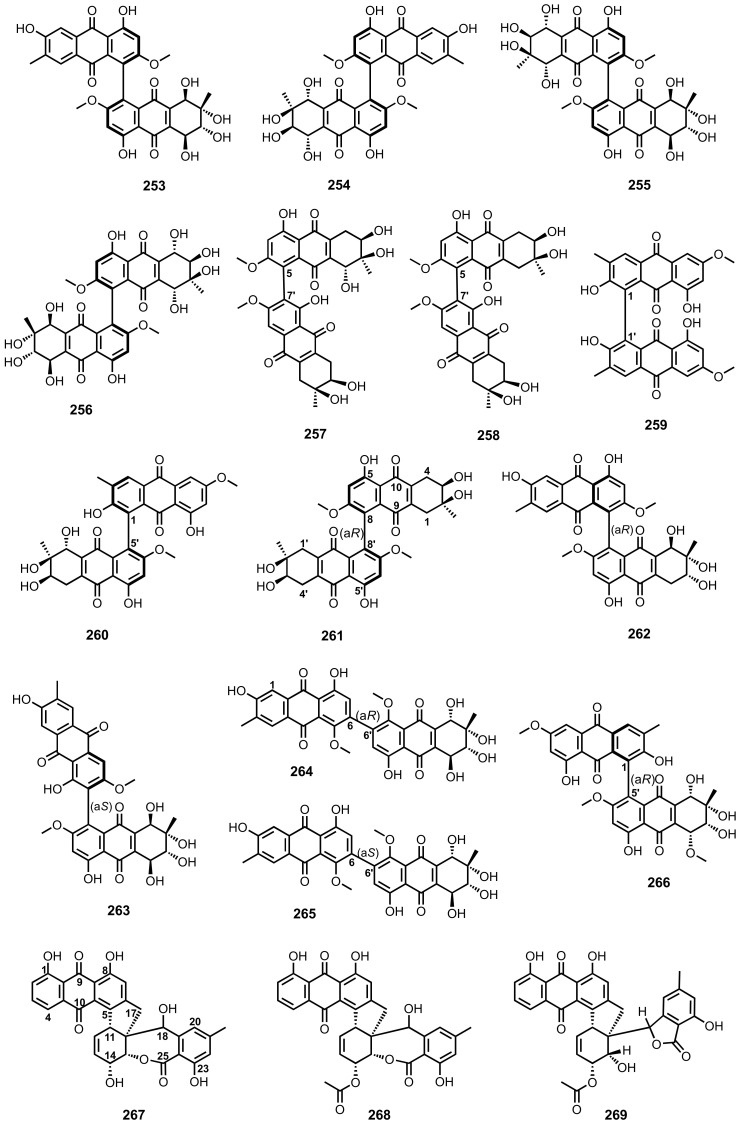
Structures of **253**–**269**.

**Figure 26 marinedrugs-20-00474-f026:**
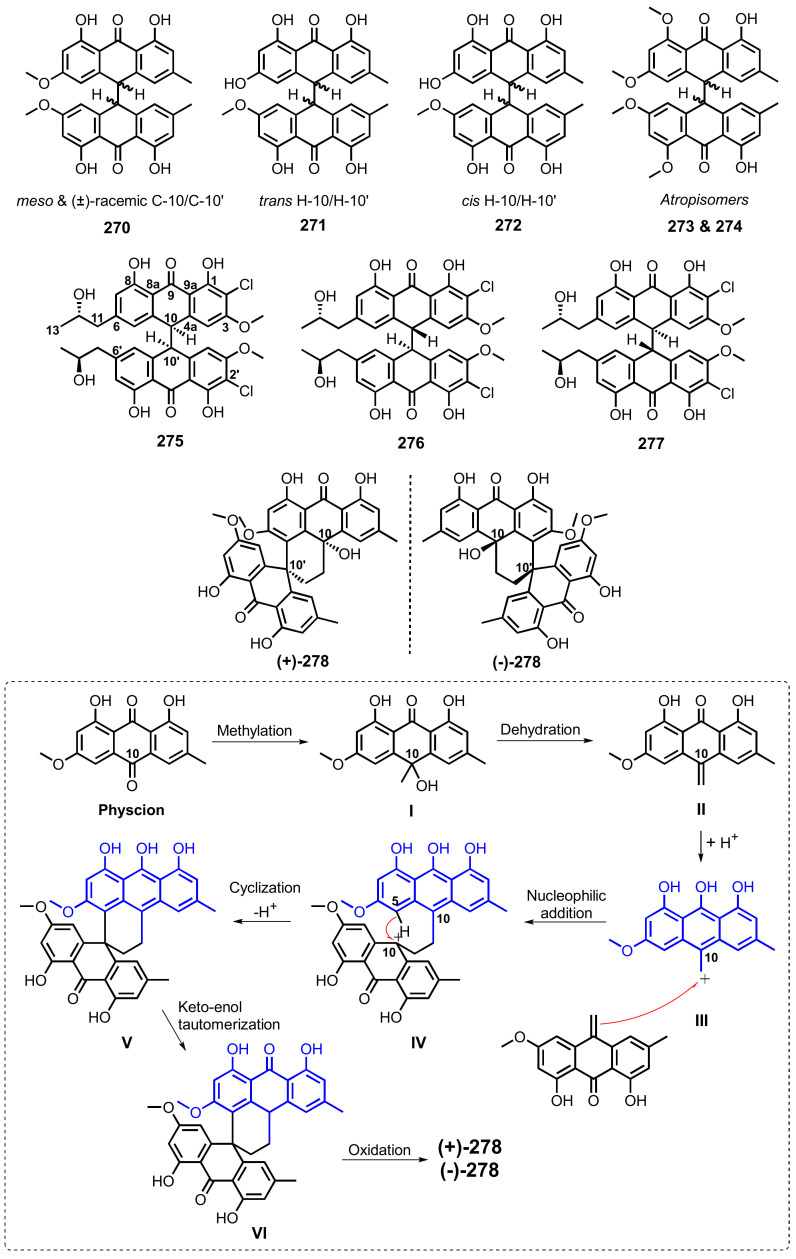
Structures of **270**–**278** and a plausible biosynthetic pathway of **278**.

**Figure 27 marinedrugs-20-00474-f027:**
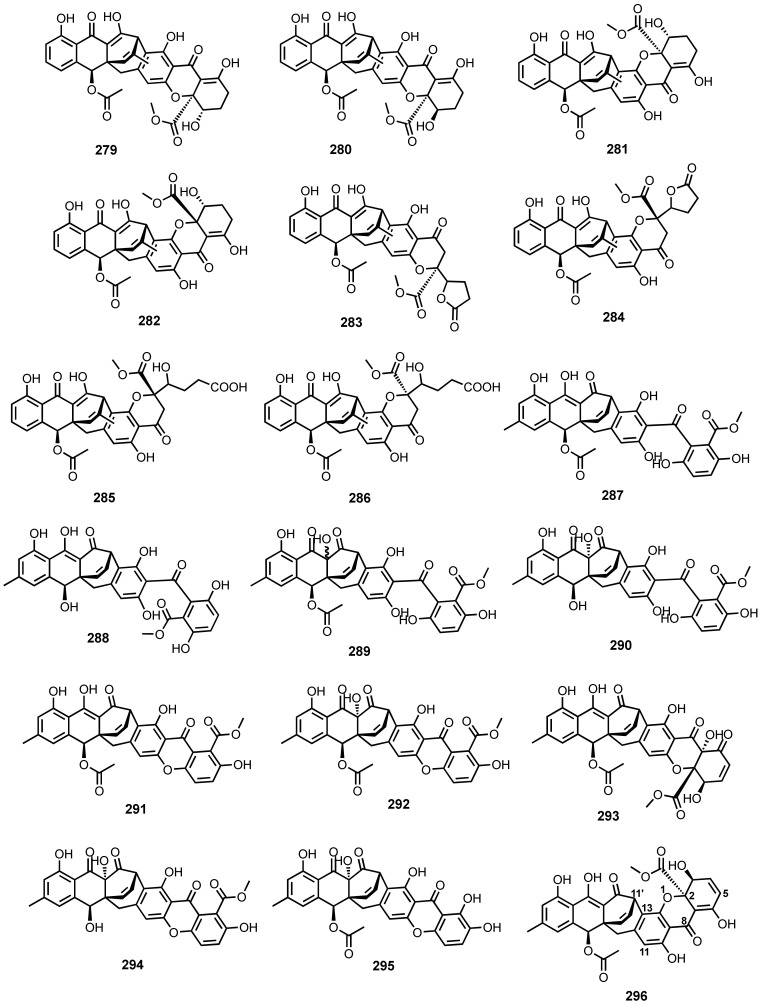
Structures of **279**–**296**.

**Figure 28 marinedrugs-20-00474-f028:**
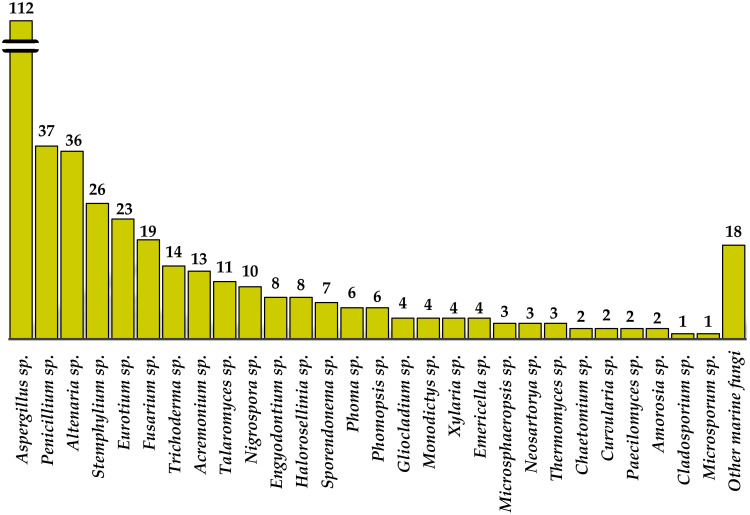
The number of isolated anthraquinone metabolites and their derivatives from the marine-derived fungal resources.

**Table 1 marinedrugs-20-00474-t001:** Anthraquinone metabolites and their analogues reported from marine-derived fungi.

Compound	Fungus Species/Strain No.	Source of Marine-Derived Fungi	Bioactivity	Ref.
Emodin (**1**)	*Aspergillus candidus* KUFA0062	-Marine sponge *Epipolasis* sp.	-	[16]
	*A. flavipes* HN4-13	-Coastal sediment.	-Non-competitive α-glucosidase inhibitor.	[18]
	*Aspergillus* sp. LS57	-Marine sponge *Haliclona* sp.	-	[19]
	*A. glaucus* HB1-19	-Marine sediment.	-Antibacterial and cytotoxic activities.	[22]
	*A. tritici* SP2-8-1	-Coral *Galaxea fascicularis.*	-Antibacterial and cytotoxic activities.	[28]
	*A. versicolor*	-Green alga *Halimeda opuntia.*	-Antiviral activity; inhibition of human trypsin activity.	[34]
	*A. versicolor*	-Deep-sea sediment.	-Antibacterial activity.	[35]
	*Penicillium oxalicum* 2HL-M-6	-Sea mud.	-	[36]
	*P. ochrochloron*	-Sea mud.	-	[17]
	*P. citrinum* PSU-F51	-Gorgonian Sea fan (*Annella* sp.)	-Antifouling activity.	[37]
	*Penicillium* sp. SCSGAF0023	-Gorgonian coral.	-Antifouling activity.	[39]
	*Eurotium rubrum*	-Inner tissue of semi-mangrove plant *Hibiscus tiliaceus.*	-Antibacterial, reduction of biofilm formation and cytotoxic activities.	[47]
	*E. chevalieri* KUFA0006	-Inner twig of mangrove plant *Rhizophora mucronata* Poir.	-Antibacterial activity.	[31]
	*Talaromyces stipitatus* KUFA0207	-Marine sponge *Stylissa flabelliformis.*	-Anti-obesity activity.	[33]
	*Paecilomyces* sp. (Tree1-7)	-Mangrove saprophytic bark.	-Antibacterial activity.	[42]
	*Trichoderma harzianum* (XS-20090075)	-Inner tissue of soft coral.	-Antibacterial activity.	[55]
	*Trichoderma* sp. (H-1)	-Sea cucumber.	-Antibacterial activity.	[43]
	*Gliocladium* sp. T31	-Marine lichen.	-	[38]
	*G. catenulatum* T31	-Marine sediment.	-Cytotoxic activity.	[65]
	*Monodictys* sp.	-Sea urchin *Anthocidaris crassispina.*	-Antitumor activity.	[60]
	-	-	-Antibiofilm formation.	[128]
Questin (MT-1; **2**)	*Aspergillus flavipes* HN4-13	-Coastal sediment.	-	[18]
	*A. terreus* DTO 403-C9	-Leaves of an unidentified mangrove tree.	-	[20]
	*A. glaucus* HB1-19	-Marine sediment.	-	[22]
	*Penicillium citrinum* HL-5126	-Mangrove *Bruguiera sexangula* var. *rhynchopetala.*	-	[41]
	*Eurotium chevalieri* KUFA0006	-Inner twig of mangrove plant *Rhizophora mucronata* Poir.	-	[31]
	*Eurotium* sp. SCSIO F452	-Marine sediment.	-	[70]
	*E. rubrum*	-Inner tissue of mangrove plant *Hibiscus tiliaceus.*	-DPPH^•^ radicals scavenging activity.	[106]
1,2,5-Trihydroxy-7-methyl-9,10-anthraquinone (**3**)	*Aspergillus terreus* DTO 403-C9	-Leaves of an unidentified mangrove tree.	-	[20]
1-Methyl emodin (**4**)	*A. europaeus* WZXY-SX-4-1	-Marine sponge *Xestospongia testudinaria.*	-Down-regulation of NF-κB.	[21]
	*A. vesicolor*	-Green alga *Halimeda opuntia.*	-Antiviral activity; inhibition of human trypsin activity.	[34]
Dermolutein (**5**)	*A. europaeus* WZXY-SX-4-1	-Marine sponge *X. testudinaria.*	-Down-regulation of NF-κB.	[21]
Physcion (or parietin; **6**)	*A. glaucus* HB1-19	-Marine sediment.	-	[22]
	*A. wentii* EN-48	-Brown alga *Sargassum* sp.	-DPPH^•^ radicals scavenging activity.	[23]
	*Penicillium* sp. ZZ901	-Wild bivalve of *Scapharca broughtonii* (Schrenck).	-Anti-proliferative activity.	[27]
	*Eurotium chevalieri* MUT2316	-Marine sponge *Grantia compressa.*	-Antifouling and antibacterial activities.	[53]
	*E. chevalieri* KUFA0006	-Inner twig of mangrove plant *Rhizophora mucronata* Poir.	-Reduction of biofilm formation.	[31]
	*Eurotium* sp. SCSIO F452	-Marine sediment.	-	[70]
	*E. repens*	-Marine sponge *Suberites domuncula.*	-Cytotoxicity against sex cells.	[24]
	*E. cristatum*	-Marine sponge *Mycale* sp.	-	[25]
	*Altenaria* sp. ZJ9-6B	-Mangrove tree *Aegiceras corniculatum* fruits.	-	[61]
	*Chaetomium globosum*	-Inner tissue of the marine red alga *Polysiphonia urceolata.*	-	[32]
	*Microsporum* sp. MFS-YL	-Marine red alga *Lomentaria catenata.*	-Anti-proliferative and cytotoxic activities.	[26]
Catenarin (**7**)	*Aspergillus glaucus* HB1-19	-Marine sediment.	-	[22]
	*Eurotium* sp. SCSIO F452	-Marine sediment.	-	[70]
Rubrocristin (**8**)	*A. glaucus* HB1-19	-Marine sediment.	-	[22]
3-Hydroxy-1,2,5,6-tetramethoxy anthracene-9,10-dione (**9**)	*A. tritici* SP2-8-1	-Soft coral *Galaxea fascicularis.*	-Antibacterial activity.	[28]
3-Hydroxy-2-hydroxymethyl-1-methoxy anthracene-9,10-dione (**10**)	*A. tritici* SP2-8-1	-Soft coral *G. fascicularis.*	-Antibacterial and cytotoxic activities.	[28]
1,2,3-Trimethoxy-7-hydroxy methylanthracene-9,10-dione (**11**)	*A. tritici* SP2-8-1	-Soft coral *G. fascicularis.*	-Antibacterial and cytotoxic activities.	[28]
1,5-Dihydroxy-3-methoxy-7-methylanthraquinone (**12**)	*A. wentii* (pt-1), *A. ustus* (cf-42), *A. versicolor* (dl-29 and pt-20)	-	-Algicidal activity.	[29]
1,3,5-Trihydroxy-7-methylanthraquinone (**13**)	*A. wentii* (pt-1), *A. ustus* (cf-42), *A. versicolor* (dl-29 and pt-20)	-	-Algicidal activity.	[29]
5-Hydroxy-2,4-dimethoxy-7-methylanthraquinone (or emodin-6,8-dimethyl ether; **14**)	*A. wentii* (pt-1), *A. ustus* (cf-42), *A. versicolor* (dl-29 and pt-20)	-	-Algicidal activity.	[29]
	*A. wentii* EN-48	-Brown alga *Sargassum* sp.	-	[30]
	*Emericella* sp. SCSIO 05240	-Marine sediment.	-	[78]
Questinol (**15**)	*Eurotium chevalieri* KUFA0006	-Inner twig of mangrove plant *Rhizophora mucronata* Poir.	-Reduction of biofilm formation.	[31]
	*Talaromyces stipitatus* KUFA0207	-Marine sponge *Stylissa flabelliformis.*	-Anti-obesity activity.	[33]
Erythroglaucin (**16**)	*Chaetomium globosum*	-Inner tissue of the marine red alga *Polysiphonia urceolata.*	-DPPH^•^ radicals scavenging activity.	[32]
Fallacinol (**17**)	*Talaromyces stipitatus* KUFA0207	-Marine sponge *Stylissa flabelliformis.*	-Anti-obesity activity.	[33]
Evariquinone (**18**)	*A. vesicolor*	-Green alga *Halimeda opuntia.*	-	[34]
7-Hydroxyemodin-6,8-dimethyl ether (**19**)	*A. vesicolor*	-Green alga *H. opuntia*	-	[34]
	*Emericella* sp. SCSIO 05240	-Marine sediment.	-	[78]
2-(dimethoxymethyl)-1-hydroxy anthracene-9,10-dione (**20**)	*A. versicolor*	-Deep-sea sediment.	-Antibacterial activity.	[35]
1-Hydroxy-2-methylanthracene-9,10-dione (**21**)	*A. versicolor*	-Deep-sea sediment.	-	[35]
2-Methylanthracene-9,10-dione (**22**)	*A. versicolor*	-Deep-sea sediment.	-	[35]
Damnacanthal (**23**)	*A. versicolor*	-Deep-sea sediment.	-	[35]
Rubiadin (**24**)	*A. versicolor*	-Deep-sea sediment.	-	[35]
Xanthopurpurin (**25**)	*A. versicolor*	-Deep-sea sediment.	-	[35]
Rubianthraquinone (**26**)	*A. versicolor*	-Deep-sea sediment.	-	[35]
6-Hydroxyrubiadin (**27**)	*A. versicolor*	-Deep-sea sediment.	-	[35]
Citreorosein (or ω-hydroxyemodin; **28**)	*Penicillium oxalicum* 2HL-M-6	-Sea mud.	-	[36]
	*P. citrinum* PSU-F51	-Gorgonian Sea fan (*Annella* sp.)	-	[37]
	*Penicillium* sp. SCSGAF0023	-Gorgonian coral.	-Antifouling activity.	[39]
	*P. citrinum* HL-5126	-Mangrove *Bruguiera sexangula* var. *rhynchopetala.*	-Antibacterial activity.	[41]
	*Talaromyces stipitatus* KUFA0207	-Marine sponge *Stylissa flabelliformis.*	-Anti-obesity activity.	[33]
	*Emericella* sp. SCSIO 05240	-Marine sediment.	-Antibacterial activity.	[78]
	*Fusarium equiseti*	-Marine brown alga *Padina pavonica.*	-Antiviral activity; inhibition of human trypsin activity.	[40]
	*Gliocladium* sp. T31	-Marine lichen.	-	[38]
	*G. catenulatum* T31	-Marine sediment.	-Anti-tumor activity.	[65]
	-	-	-Anti-biofilm formation.	[135]
Chrysophanol (or chrysophanic acid; **29**)	*Aspergillus candidus* KUFA0062	-Marine sponge *Epipolasis* sp.	-Anti-biofilm formation.	[16]
	*Penicillium oxalicum* 2HL-M-6	-Sea mud.	-	[36]
	*P. citrinum* PSU-F51	-Gorgonian Sea fan (*Annella* sp.)	-	[37]
	*Paecilomyces* sp. (Tree1-7)	-Mangrove saprophytic bark.	-	[42]
	*Fusarium equiseti*	-Marine brown alga *Padina pavonica.*	-Antiviral activity; inhibition of human trypsin activity.	[40]
	*Trichoderma harzianum* (XS-20090075)	-Inner tissue of soft coral.	-Anti-acetylcholinesterase activity.	[55]
	*Trichoderma* sp. (H-1)	-Sea cucumber.	-Antibacterial activity.	[43]
	*Monodictys* sp.	-Sea urchin *Anthocidaris crassispina.*	-	[60]
	Strain F-F-3C	-Unidentified marine red alga.	-Antibacterial activity.	[44]
Aloe-emodin (**30**)	*Penicillium oxalicum* 2HL-M-6	-Sea mud.	-	[36]
Carviolin (**31**)	*Penicillium* sp. strain F01V25	-Marine alga *Dictyosphaeria versluyii.*	-	[45]
Emodic acid (**32**)	*Penicillium* sp. SCSIOsof101	-Deep-sea sediment.	-	[46]
	*Eurotium rubrum*	-Inner tissue of semi-mangrove plant *Hibiscus tiliaceus.*	-	[47]
Macrosporin (**33**)	*Penicillium* sp.	-Soft coral *Sarcophyton tortuosum.*	-	[48]
	*Altenaria* sp. ZJ-2008003	-Soft coral *Sarcophyton* sp.	-Antibacterial activity.	[49]
	*Stemphylium* sp. 33231	-Mangrove tree *Bruguiera sexangula* var. *rhynchopetala*.	-Antibacterial activity.	[50]
	*S. lycopersici*	-Inner tissue of gorgonian soft coral *Dichotella gammacea.*	-	[51]
	*Phoma* sp. L28	-Mangrove plant *Myoporum bontioides* A. Gray.	-Anti-fungal activity.	[67]
	*Phomopsis* sp. PSU-MA214	-Leaves of mangrove plant *Rhizophora apiculata* Griff. Ex T. Anderson.	-	[52]
1,7,8-Tri-hydroxy-3-methoxy-6-methyl anthraquinone (**34**)	*Penicillium* sp.	-Soft coral *Sarcophyton tortuosum.*	-	[48]
1-Hydroxy-3-methoxy-6-methylanthraquinone (**35**)	*Penicillium* sp.	-Soft coral *Sarcophyton tortuosum.*	-	[48]
	*Phomopsis* sp. PSU-MA214	-Leaves of mangrove plant *Rhizophora apiculata* Griff. Ex T. Anderson.	-	[52]
Cinnalutein (**36**)	*Eurotium chevalieri* MUT2316	-Marine sponge *Grantia compressa.*	-Antifouling and algicidal activities.	[53]
Acetylquestinol (**37**)	*E. chevalieri* KUFA0006	-Inner twig of mangrove plant *Rhizophora mucronata* Poir.	-Reduction of biofilm formation.	[31]
	*Neosartorya spinosa* KUFA1047	-Marine sponge *Mycale* sp.	-	[54]
Pachybasin (**38**)	*Trichoderma harzianum* (XS20090075)	-Inner tissue of a soft coral.	-Anti-acetylcholinesterase activity.	[55]
	*Monodictys* sp.	-Sea urchin *Anthocidaris crassispina.*	-	[60]
Phomarin (**39**)	*T. harzianum* (XS-20090075)	-Inner tissue of a soft coral.	-	[55]
1-Hydroxy-3-hydroxy methylanthraquinone (**40**)	*T. harzianum* (XS-20090075)	-Inner tissue of a soft coral.	-Antibacterial and cytotoxic activities.	[55]
ω-Hydroxydigitoemodin (**41**)	*T. harzianum* (XS-20090075)	- Inner tissue of a soft coral.	-Anti-acetylcholinesterase activity.	[55]
1,3,6-Trihydroxy-8-methylanthraquinone (**42**)	*Trichoderma* sp. strain SCSIO41004	-Marine sponge *Callyspongia* sp.	-	[56]
1,4-Dihydroxy-2-methoxy-7-methylanthracene-9,10-dione (**43**)	*Halorosellinia* sp. (no. 1403)	-Estuarine.	-	[57]
1,4,6-Trihydroxy-2-methoxy-7-methylanthracene-9,10-dione (**44**)	*Halorosellinia* sp. (no. 1403)	-Decayed *Kandelia candel* (L.) Druce.	-	[58]
Demethoxyaustrocortirubin (**45**)	*Halorosellinia* sp. (no. 1403)	-Decayed *K. candel* (L.) Druce.	-	[58]
Hydroxy-9,10-anthraquinone (**46**)	*Halorosellinia* sp. (no. 1403)	-Decayed *Kandelia candel* (L.) Druce.	-Cytotoxic activity.	[58,131]
Austrocortinin (**47**)	*Altenaria* sp. (SK11)	-Root of mangrove tree *Excoecaria agallocha.*	-	[62]
	*Fusarium* sp. PSU-F14	-Gorgonian sea fan.	-Antibacterial and cytotoxic activities.	[59]
	*Nigrospora* sp. ZJ-2010006	-Unidentified sea anemone.	-	[68]
	*Nigrospora* sp. ZJ-2010006	-Inner tissue of the zoathid *Palythoa haddoni* (GX-WZ-20100026).	-Antiviral activity.	[69]
	*Halorosellinia* sp. (no. 1403)	-Decayed *Kandelia candel* (L.) Druce.	-	[58]
Monodictyquinone A (**48**)	*Monodictys* sp.	-Sea urchin *Anthocidaris crassispina.*	-Antibacterial activity.	[60]
Rheoemodin (**49**)	*Talaromyces stipitatus* KUFA0207	-Marine sponge *Stylissa flabelliformis.*	-Antiobesity activity.	[33]
Marcrospin (**50**)	*Altenaria* sp. ZJ9-6B	-Mangrove tree *Aegiceras corniculatum* fruits.	-	[61]
6-Methylquinizarin (**51**)	*Altenaria* sp. (SK11)	-Root of mangrove tree *Excoecaria agallocha.*	-	[62]
6-*O*-Methylalaternin (**52**)	*Altenaria tenuissima* DFFSCS013	-Marine sediment.	-Inhibition of human protein tyrosine phosphatases and inhibition of indoleamine 2,3-dioxygenase activity.	[63]
Lunatin (**53**)	*Curvularia lunata*	-Marine sponge *Niphates olemda.*	-Antibacterial activity.	[64]
	*Gliocladium catenulatum* T31	-Marine sediment.	-Anti-tumor activity.	[65]
1,3-Dihydroxy-6-hydroxymethyl-7-methoxyanthraquinone (**54**)	*Thermomyces lanuginosus* Tsikl KMM 4681	-Marine sediment.	-	[66]
1,3-Dihydroxy-6-methyl-7-methoxyanthraquinone (**55**)	*T. lanuginosus* Tsikl KMM 4681	-Marine sediment.	-Cytotoxic activity.	[66]
7-Methoxymacrosporin (**56**)	*Phoma* sp. L28	-Mangrove plant *Myoporum bontioides* A. Gray.	-Antifungal activity.	[67]
7-(γ,γ)-Dimethylallyloxymacrosporin (**57**)	*Phoma* sp. L28	-Mangrove plant *M. bontioides* A. Gray.	-Antifungal activity.	[67]
3,5,8-Tri-hydroxy-7-methoxy-2-methylanthracene-9,10-dione (**58**)	*Nigrospora* sp. ZJ-2010006	-Unidentified sea anemone.	-Antibacterial activity.	[68]
	*Nigrospora* sp. ZJ-2010006	-Inner tissue of the zoathid *Palythoa haddoni* (GX-WZ-20100026).	-Antiviral activity.	[69]
1,6,8-Trihydroxy-4-benzoyloxy-3-methylanthraquinone (**59**)	*Eurotium* sp. SCSIO F452	-Marine sediment.	-	[70]
1,3,6,8-Tetrahydroxyanthraquinone analogues (**60**-**62**)	*Microsphaeropsis* sp.	-Marine sponge *Aplysina aerophoba.*	-Inhibition of protein kinases.	[71,72]
7-Acetyl-1,3,6-trihydroxyanthracene-9,10-dione (**63**)	*Trichoderma* sp. strain SCSIO41004	-Marine sponge *Callyspongia* sp.	-	[56]
	*Fusarium equiseti*	-Intertidal marine plants.	-Antibacterial activity.	[73]
ZSU-H85 (**64**)	*Trichoderma* sp. strain SCSIO41004	-Marine sponge *Callyspongia* sp.	-Antiviral activity.	[56]
(11*S*)-1,3,6-Trihydroxy-7-(1-hydroxyethyl)anthracene-9,10-dione (**65**)	*F. equiseti*	-Intertidal marine plants.	-Antibacterial activity.	[73]
	*Cladosporium* sp. HNWSW-1	-Fresh roots of *Ceriops tagal.*	-Inhibition of α-glucosidase activity.	[74]
5-Acetyl-2-methoxy-1,4,6-trihydroxy-anthraquinone (**66**)	*Fusarium* sp. (no. b77)	-Costal environment.	-	[75]
1-Acetoxy-5-acetyl-2-methoxy-4,6- trihydroxy-anthraquinone (**67**)	*Fusarium* sp. (no. b77)	-Costal environment.	-	[75]
Isorhodoptilometrin (**68**)	*Penicillium oxalicum* 2HL-M-6	-Sea mud.	-	[36]
	*Penicillium* sp. SCSGAF0023	-Gorgonian coral.	-Antifouling activity.	[39]
	*Gliocladium* sp. T31	-Marine lichen.	-	[38]
	*G. catenulatum* T31	-Marine sediment.	-Antitumor activity.	[65]
(−)-2′*R*-1-Hydroxyisorhodopilometrin (**69**)	*Penicillium* sp. OUCMDZ-4736	-Mangrove roots of *Acanthus ilicifolius.*	-Antiviral activity.	[76]
Isorodoptilometrin-1-methyl ether (**70**)	*Aspergillus vesicolor*	-Green alga *Halimeda opuntia.*	-Antibacterial activity.	[34]
(+)-2′*S*-isorhodoptilometrin (**71**)	*Trichoderma harzianum* (XS-20090075)	-Inner tissue of a soft coral.	-Antibacterial, cytotoxic and anti-acetylcholinesterase activities.	[55]
Nalgiovensin (**72**)	*A. alliaceus*	-	-	[77]
1-Methylether nalgiovensin (**73**)	*Emericella* sp. SCSIO 05240	-Marine sediment.	-Antibacterial activity.	[78]
Penipurdin A (**74**)	*Neosartorya spinosa* KUFA 1047	-Marine sponge *Mycale* sp.	-Anti-tyrosinase activity.	[54]
Acetylpenipurdin A (**75**)	*N. spinosa* KUFA 1047	-Marine sponge *Mycale* sp.	-	[54]
1,3,6-trihydroxy-7-(dihydroxypropyl)-anthraquinone (**76**)	*Thermomyces lanuginosus* Tsikl KMM 4681	-Marine sediment.	-	[66]
6,8-Dimethoxy-1-methyl-2-(3-oxobutyl)-anthrakunthone (**77**)	*Fusarium* sp. ZZF60	-Marine mangrove plant.	-Cytotoxic activity.	[79]
Norsolorinic acid (**78**)	*A. nidulans* MA-143	-Leaves of mangrove plant *Rhizophora stylosa.*	-	[80]
8-*O*-Methyl versiconol (**79**)	*A. puniceus* SCSIO z021	-Deep-sea sediment.	-Inhibition of human protein tyrosine phosphatases.	[81]
2′,3′-Dihydorxy versiconol (**80**)	*A. puniceus* SCSIO z021	-Deep-sea sediment.	-	[81]
Methyl averantin (**81**)	*A. puniceus* SCSIO z021	-Deep-sea sediment.	-Inhibition of human protein tyrosine phosphatases.	[81]
	*A. versicolor*	-Marine sponge *Petrosia* sp.	-Antibacterial and cytotoxic activities.	[84]
	*A. versicolor* INF 16-17	-Inner tissue of an unidentified marine clam.	-	[88]
	*A. versicolor* A-21-2-7	-Deep-sea sediment.	-	[89]
	*Aspergillus* sp. SCSIO F063	-Deep-sea sediment.	-Cytotoxic activity.	[91]
	*A. versicolor* SCSIO-41502	-Deep-sea sediment.	-	[92]
Versiconol (**82**)	*A. puniceus* SCSIO z021	-Deep-sea sediment.	-Inhibition of human protein tyrosine phosphatases.	[81]
	*A. versicolor*	-Marine sponge *Petrosia* sp.	-Cytotoxic activity.	[84]
	*A. versicolor* SCSIO-41502	-Deep-sea sediment.	-	[92]
	*Aspergillus* sp. F40	-Marine sponge *Callyspongia* sp.	-Antibacterial activity.	[93]
	*Penicillium flavidorsum* SHK1-27	-	-Anti-proliferative activity.	[85]
	Strain ZSUH-36	-Mangrove *Acanthus ilicifolius* Linn.	-	[83]
6,8-Di-*O*-methylaverantin (**83**)	*A. versicolor* EN-7	-Brown algae *Saragassum thunbergii.*	-Antimicrobial activity.	[82]
6,8-Di-*O*-methylversiconol (**84**)	*A. versicolor* EN-7	-Brown algae *Saragassum thunbergii.*	-Antimicrobial activity.	[82]
	Strain ZSUH-36	-Mangrove *Acanthus ilicifolius* Linn.	-	[83]
Averantin (**85**)	*A. versicolor*	-Marine sponge *Petrosia* sp.	-Antibacterial and cytotoxic activities.	[84]
	*A. versicolor* INF 16-17	-Inner tissue of an unidentified marine clam.	-	[88]
	*A. versicolor* A-21-2-7	-Deep-sea sediment.	-	[89]
	*Aspergillus* sp. SCSIO F063	-	-Cytotoxic activity.	[91]
	*P. flavidorsum* SHK1-27	-	-Anti-proliferative activity.	[85]
6,8,1′-Tri-*O*-methylaverantin (**86**)	*Aspergillus* sp. SF-6796	-	-Anti-neuroinflammatory activity.	[87]
	Strain ZSUH-36	-Mangrove *Acanthus ilicifolius* Linn.	-	[86]
Averythrin (**87**)	*A. versicolor* INF 16-17	-Inner tissue of an unidentified marine clam.	-	[88]
	*A. versicolor* A-21-2-7	-Deep-sea sediment.	-Antioxidant activity.	[89]
	*Aspergillus* sp. 16-5C	-Leaves of *Sonneratia apetala.*	-Anti-*Mycobacterium tuberculosis* activity.	[90]
	*Aspergillus* sp. SCSIO F063	-Deep-sea sediment.	-Cytotoxic activity.	[91]
(1′*S*)-6,1′-*O*,*O*-Dimethylaverantin (**88**)	*Aspergillus* sp. SCSIO F063	-Deep-sea sediment.	-Cytotoxic activity.	[91]
(*S*)-(−)-Averantin (**89**)	*Aspergillus* sp. SCSIO F063	-Deep-sea sediment.	-	[91]
6,*O*-Methylaverantin (**90**)	*Aspergillus* sp. SCSIO F063	-Deep-sea sediment.	-	[91]
Averantin-1′-butyl ether (**91**)	*Aspergillus* sp. SCSIO F063	-Deep-sea sediment.	-Cytotoxic activity.	[91]
Aspergilol I (**92**)	*A. versicolor* SCSIO-41502	-Deep-sea sediment.	-Antiviral activity.	[92]
SC3-22-3 (**93**)	*A. versicolor* SCSIO-41502	-Deep-sea sediment.	-	[92]
Coccoquinone A (**94**)	*A. versicolor* SCSIO-41502	-Deep-sea sediment.	-Antiviral activity.	[92]
Versiconol B (**95**)	*Aspergillus* sp. F40	-Marine sponge *Callyspongia* sp.	-Antibacterial activity.	[93]
(+)-1-*O*-Demethylvariecolorquinone A (**96**)	*A. europaeus* WZXY-SX-4-1	-Marine sponge *Xestospongia testudinaria.*	-	[21]
(+)-Variecolorquinone A (**97**)	*A. europaeus* WZXY-SX-4-1	-Marine sponge *X. testudinaria.*	-DPPH^•^ radical scavenging activity.	[21]
	*A. glaucus* HB1-19	-Deep-sea sediment.	-	[22]
	*Eurotium cristatum* EN-220	-Marine brown alga *Sargassum thunbergii.*	-	[95]
6-*O*-Methylaverufin (**98**)	*A. nidulans* MCCC 3A00050	-Deep-sea sediment.	-	[96]
	*A. versicolor* EN-7	-Brown alga *Sargassum thunbergii.*	-	[82]
6,8-Di-*O*-methylaverufin (**99**)	*A. nidulans* MCCC 3A00050	-Deep-sea sediment.	-	[96]
	*Aspergillus* sp. SF-6796	-	-	[87]
	*A. versicolor* EN-7	-Brown alga *Sargassum thunbergii.*	-	[82]
	*Aspergillus* sp. 16-5C	-Leaves of *Sonneratia apetala.*	-	[90]
	*P. flavidorsum* SHK1-27	-	-Anti-proliferative activity.	[85]
	*Acremonium vitellinum*	-Inner tissue of an unidentified marine red alga.	-Insecticidal activity.	[104]
	Strain ZSUH-36	-Mangrove *Acanthus ilicifolius* Linn.	-	[86]
Aversin (**100**)	*A. nidulans* MCCC 3A00050	-Deep-sea sediment.	-	[96]
	*A. versicolor* MF359	-Marine sponge *Hymeniacidon perleve.*	-	[97]
	*A. versicolor* EN-7	-Brown alga *Sargassum thunbergii.*	-	[82]
	*Aspergillus* sp. 16-5C	-Leaves of *Sonneratia apetala*	-	[90]
	*Acremonium vitellinum*	-Inner tissue of an unidentified marine red alga.	-Insecticidal activity.	[104]
	*P. flavidorsum* SHK1-27	-	-Anti-proliferative activity.	[85]
	Strain ZSUH-36	-Mangrove *Acanthus ilicifolius* Linn.	-	[83]
8-*O*-Methylversicolorin A (**101**)	*A. nidulans* MCCC 3A00050	-Deep-sea sediment.	-	[96]
Isoversicolorin C (**102**)	*A. nidulans* MA-143	-Leaves of mangrove plant *Rhizophora stylosa.*	-Antibacterial activity.	[80]
Versicolorin C (**103**)	*A. nidulans* MA-143	-Leaves of mangrove plant *Rhizophora stylosa.*	-Antibacterial activity.	[80]
	Strain ZSUH-36	-Mangrove *Acanthus ilicifolius* Linn.	-	[86]
	Isolate 1850	-Leaves of a mangrove tree *Kandelia candel.*	-	[100]
Averufin (**104**)	*A. nidulans* MA-143	-Leaves of mangrove plant *Rhizophora stylosa.*	-	[80]
	*A. versicolor* SCSIO-41502	-Deep-sea sediment.	-	[92]
	-	-	-Anti-*Mycobacterium tuberculosis* activity.	[129]
	*A. niger* (MF-16)	-Sea water.	-Antiviral activity.	[98]
	*A. versicolor*	-Marine sponge *Petrosia* sp.	-Antibacterial and cytotoxic activities.	[84]
	*A. puniceus* SCSIO z021	-Deep-sea sediment.	-Inhibition of protein tyrosine phosphatases and toxicity against Brine shrimps.	[81]
	*A. versicolor* MF18051	-Marine sediment.	-Antibacterial activity.	[101]
	*Aspergillus* sp. F40	-Marine sponge *Callyspongia* sp.	-	[93]
	*A. versicolor* SCSIO 41016	-Marine sponge.	-	[102]
	*A. versicolor* A-21-2-7	-Deep-sea sediment.	-Antioxidant activity.	[89]
	*Aspergillus* sp.	-Gorgonian *Dichotella gemmacea.*	-Antibacterial activity.	[103]
	*P. flavidorsum* SHK1-27	-	-Anti-proliferative activity.	[85]
	Strain ZSUH-36	-Mangrove *Acanthus ilicifolius* Linn.	-	[86]
	Isolate 1850	-Leaves of a mangrove tree *Kandelia candel.*	-	[100]
Paeciloquinone E (**105**)	*A. nidulans* MA-143	-Leaves of mangrove plant *Rhizophora stylosa.*	-	[80]
Averufanin (**106**)	*A. nidulans* MA-143	-Leaves of mangrove plant *Rhizophora stylosa.*	-	[80]
	*A. puniceus* SCSIO z021	-Deep-sea sediment.	-Inhibition of protein tyrosine phosphatases and toxicity against Brine shrimps.	[81]
	*A. versicolor* A-21-2-7	-Deep-sea sediment.	-	[89]
	*Aspergillus* sp.	-Gorgonian *Dichotella gemmacea.*	-Antibacterial activity.	[103]
Nidurufin (**107**)	*A. niger* (MF-16)	-Sea water.	-Antiviral activity.	[98]
	*A. versicolor*	-Marine sponge *Petrosia* sp.	-Antibacterial and cytotoxic activities.	[84]
	*A. puniceus* SCSIO z021	-Deep-sea sediment.	-Inhibition of protein tyrosine phosphatases.	[81]
	*A. versicolor* SCSIO 41016	-Marine sponge.	-Antiviral activity.	[102]
	*A. versicolor* A-21-2-7	-Deep-sea sediment.	-Antioxidant and cytotoxic activities.	[89]
	*Aspergillus* sp.	-Gorgonian *Dichotella gemmacea.*	-Antibacterial and cytotoxic activities.	[103]
	*P. flavidorsum* SHK1-27	-	-Anti-proliferative activity and cell cycle inhibitor.	[85,99]
	Isolate 1850	-Leaves of a mangrove tree *Kandelia candel.*	-	[100]
3′-Hydroxy-8-*O*-methyl verscicolorin B (**108**)	*A. puniceus* SCSIO z021	-Deep-sea sediment.	-Inhibition of protein tyrosine phosphatases.	[81]
Versicolorin B (**109**)	*A. puniceus* SCSIO z021	-Deep-sea sediment.	-Inhibition of protein tyrosine phosphatases and toxicity against Brine shrimps.	[81]
	*A. versicolor* MF18051	-Marine sediment.	-Antibacterial activity.	[101]
	*Aspergillus* sp. F40	-Marine sponge *Callyspongia* sp.	-	[93]
	*A. versicolor* SCSIO 41016	-Marine sponge.	-	[102]
	*A. versicolor* A-21-2-7	-Deep-sea sediment.	-Antioxidant and cytotoxic activities.	[89]
	*P. flavidorsum* SHK1-27	-	-Anti-proliferative activity.	[85]
8-*O*-Methylnidurufin (**110**)	*A. puniceus* SCSIO z021	-Deep-sea sediment.	-Inhibition of protein tyrosine phosphatases.	[81]
	*Aspergillus* sp.	-Gorgonian *Dichotella gemmacea.*	-Antibacterial activity.	[103]
2′-Hydroxyversicolorin B (**111**)	*A. versicolor* SCSIO 41016	-Marine sponge.	-	[102]
Noraverufanin (**112**)	*Aspergillus versicolor* SCSIO 41016	-Marine sponge.	-Antiviral activity.	[102]
6,8-Di-*O*-methylnidurufin (**113**)	*A. versicolor* EN-7	-Brown alga *Sargassum thunbergii.*	-Antibacterial activity.	[82]
	*Aspergillus* sp. 16-5C	-Leaves of *Sonneratia apetala.*	-	[90]
	*Acremonium vitellinum*	-Inner tissue of an unidentified marine red alga.	-Insecticidal activity.	[104]
6,8-Di-*O*-methylversicolorin A (**114**)	*A. versicolor* EN-7	-Brown alga *Sargassum thunbergii.*	-	[82]
UCT1072M1 (**115**)	*A. versicolor* A-21-2-7	-Deep-sea sediment.	-Antioxidant activity.	[89]
Asperquinone A (**116**)	*Aspergillus* sp. 16-5C	-Leaves of *Sonneratia apetala.*	-	[90]
8-O-Methylaverufin (**117**)	*Aspergillus* sp.	-Gorgonian *Dichotella gemmacea.*	-Antibacterial activity	[103]
8-*O*-Methylaverufanin (**118**)	*Aspergillus* sp.	-Gorgonian *D. gemmacea.*	-	[103]
	*P. flavidorsum* SHK1-27	-	-Anti-proliferative activity.	[85]
Versicolorin A (**119**)	*P. flavidorsum* SHK1-27	-	-Anti-proliferative activity.	[85]
6,8-Di-*O*-methylbipolarin (**120**)	*A. vitellinum*	-Inner tissue of an unidentified marine red alga.	-Insecticidal activity.	[104]
6,8-Di-*O*-methyl-averufinan (**121**)	Strain ZSUH-36	-Mangrove *Acanthus ilicifolius* Linn.	-	[86]
Aspergilol (±)-A (**122**)	*A. versicolor* SCSIO-41502	-Deep-sea sediment.	-Antioxidant activity.	[92]
Aspergilol (±)-B (**123**)	*A. versicolor* SCSIO-41502	-Deep-sea sediment.	-Antioxidant activity.	[92]
Aspergilol (±)-G (**124**)	*A. versicolor* SCSIO-41502	-Deep-sea sediment.	-Antioxidant activity.	[92]
Aspergilol (±)-H (**125**)	*A. versicolor* SCSIO-41502	-Deep-sea sediment.	-Antioxidant and antiviral activities.	[92]
Penicillanthranin A (**126**)	*Penicillium citrinum* PSU-F51	-Gorgonian Sea fan (*Annella* sp.)	-Antibacterial and cytotoxic activities.	[37]
Penicillanthranin B (**127**)	*P. citrinum* PSU-F51	-Gorgonian Sea fan (*Annella* sp.)	-	[37]
Emodacidamide A (**128**)	*Penicillium* sp. SCSIOsof101	-Marine sediment.	-Anti-inflammatory activity.	[46]
Emodacidamide B (**129**)	*Penicillium* sp. SCSIOsof101	-Marine sediment.	-	[46]
Emodacidamide D (**130**)	*Penicillium* sp. SCSIOsof101	-Marine sediment.	-Anti-inflammatory activity.	[46]
Emodacidamide E (**131**)	*Penicillium* sp. SCSIOsof101	-Marine sediment.	-Anti-inflammatory activity.	[46]
Emodacidamide H (**132**)	*Penicillium* sp. SCSIOsof101	-Marine sediment.	-	[46]
Anthrininone B (**133**)	*Altenaria tenuissima* DFFSCS013	-Marine sediment.	-Inhibition of human protein tyrosine phosphatases and inhibition of indoleamine 2,3-dioxygenase activity.	[63]
Anthrininone C (**134**)	*A. tenuissima* DFFSCS013	-Marine sediment.	-Inhibition of human protein tyrosine phosphatases and inhibition of indoleamine 2,3-dioxygenase activity.	[63]
7-Chloroemodin (**135**)	*Penicillium ochrochloron*	-Sea sand.	-	[17]
2-Chloro-1,3,8-trihydroxy-6-(hydroxy methyl)anthracene-9,10-dione (**136**)	*Penicillium* sp. SCSIOsof101	-Marine sediment.	-	[46]
2′-Acetoxy-7-chlorocitreorosein (**137**)	*P. citrinum* HL-5126	-Mangrove *Bruguiera sexangula* var. *rhynchopetala.*	-Antibacterial activity.	[41]
7-Chloro-1′-hydroxyisorhodoptilometrin (**138**)	*Penicillium* sp. SCSIO sof101	-Marine sediment.	-	[105]
(1′*S*)-7-Chloroaverantin (**139**)	*Aspergillus* sp. SCSIO F063	-Marine sediment.	-Cytotoxic activity.	[91]
(1′*S*)-6-*O*-Methyl-7-chloroaverantin (**140**)	*Aspergillus* sp. SCSIO F063	-Marine sediment.	-Cytotoxic activity.	[91]
(1′*S*)-1′-*O*-Methyl-7-chloroaverantin (**141**)	*Aspergillus* sp. SCSIO F063	-Marine sediment.	-Cytotoxic activity.	[91]
(1′*S*)-6,1′-*O*,*O*-Dimethyl-7-chloroaverantin (**142**)	*Aspergillus* sp. SCSIO F063	-Marine sediment.	-Cytotoxic activity.	[91]
(1′*S*)-7-Chloroaverantin-1′-butyl ether (**143**)	*Aspergillus* sp. SCSIO F063	-Marine sediment.	-Cytotoxic activity.	[91]
7-Chloroaverythrin (**144**)	*Aspergillus* sp. SCSIO F063	-Marine sediment.	-Cytotoxic activity.	[91]
6-*O*-Methyl-7-chloroaverythrin (**145**)	*Aspergillus* sp. SCSIO F063	-Marine sediment.	-Cytotoxic activity.	[91]
(1′*S*)-6,1′-*O*,*O*-Dimethyl-7-bromoaverantin (**146**)	*Aspergillus* sp. SCSIO F063	-Marine sediment.	-Cytotoxic activity.	[91]
(1′*S*)-6-*O*-Dimethyl-7-bromoaverantin (**147**)	*Aspergillus* sp. SCSIO F063	-Marine sediment.	-Cytotoxic activity.	[91]
Nalgiolaxin (**148**)	*A. alliaceus*	-Marine algae.	-	[77]
7-Chloro-versicolorin A (**149**)	*A. puniceus* SCSIO z021	-Deep-sea sediment.	-Inhibition of protein tyrosine phosphatases and toxicity against Brine shrimps.	[81]
Emodacidamide C (**150**)	*Penicillium* sp. SCSIOsof101	-Marine sediment.	-Anti-inflammatory activity.	[46]
Emodacidamide F (**151**)	*Penicillium* sp. SCSIOsof101	-Marine sediment.	-	[46]
Emodacidamide G (**152**)	*Penicillium* sp. SCSIOsof101	-Marine sediment.	-	[46]
Macrosposrin-7-*O*-sulfate (**153**)	*Stemphylium* sp. 33231	-Mangrove tree *Bruguiera sexangula* var. *rhynchopetala.*	-	[50]
Emodin-3-*O*-sulphate (**154**)	*Penicillium P. oxalicum* 2HL-M-6	-Sea mud.	-	[36]
Citreorosein-3-*O*-sulphate (**155**)	*P. oxalicum* 2HL-M-6	-Sea mud.	-	[36]
	*Eurotium rubrum*	-Inner tissue of semi-mangrove plant *Hibiscus tiliaceus.*	-DPPH^•^ radicals scavenging activity.	[106]
6-*O*-(α-D-ribofuranosyl)-questin (**156**)	*Eurotium rubrum*	-Inner tissue of mangrove plant *H. tiliaceus.*	-DPPH^•^ radicals scavenging activity.	[106]
	*E. cristatum* EN-220	-Marine brown alga *Sargassum thunbergii.*	-Antibacterial activity.	[95]
6-*O*-(α-D-ribofuranosyl)-questinol (**157**)	*E. cristatum* EN-220	-Marine brown alga *S. thunbergii.*	-Antibacterial activity.	[95]
2-*O*-(6′-Acetyl)-α-D-glucopyranoside (**158**)	*Stemphylium* sp. 33231	-Mangrove tree *Bruguiera sexangula* var. *rhynchopetala.*	-Cytotoxic activity.	[50]
Macrosporin 2-*O*-α-D-glucopyranoside (**159**)	*S. lycopersici*	-Inner tissue of gorgonian soft coral *Dichotella gammacea.*	-	[51]
Wentiquinone A (**160**)	*Aspergillus wentii* EN-48	-Marine alga *Sargassum* sp.	-	[30]
Wentiquinone B (**161**)	*A. wentii* EN-48	-Marine alga *Sargassum* sp.	-	[30]
1,8-Dihydroxy-10-methoxy-3-methyldibenzo[*b,e*]oxepin-6,11-dione (**162**)	*A. wentii* EN-48	-Marine alga *Sargassum* sp.	-	[30]
	*A. europaeus* WZXY-SX-4-1	-Marine sponge *Xestospongia testudinaria.*	-	[20]
Wentiquinone C (**163**)	*A. europaeus* WZXY-SX-4-1	-Marine sponge *X. testudinaria.*	-	[20]
	*A. wentii* EN-48	-Marine brown alga *Sargassum* sp.	-	[23]
9-Dehydroxyeurotinone (**164**)	*E. rubrum*	-Inner tissue of semi-mangrove plant *Hibiscus tiliaceus.*	-Cytotoxic activity.	[47]
2-*O*-Methyl-9-dehydroxyeurotinone (**165**)	*E. rubrum*	-Inner tissue of semi-mangrove plant *H. tiliaceus.*	-	[47]
	*Eurotium* sp. SCSIO F452	-Marine sediment.	-	[70]
	*E. rubrum*	-Inner tissue of semi-mangrove plant *H. tiliaceus.*	-DPPH^•^ radicals scavenging activity.	[106]
2-*O*-Methyleurotinone (**166**)	*E. rubrum*	-Inner tissue of semi-mangrove plant *H. tiliaceus.*	-DPPH^•^ radicals scavenging activity.	[106]
2-*O*-Methyl-4-*O*-(α-D-ribofuranosyl)-9-dehydroxyeurotinone (**167**)	*Eurotium rubrum*	-Inner tissue of semi-mangrove plant *H. tiliaceus.*	-DPPH^•^ radicals scavenging activity.	[106]
Aspetritone B (**168**)	*A. tritici* SP2-8-1	-Soft coral *Galaxea fascicularis.*	-Antibacterial and cytotoxic activities.	[28]
(3*R*)-1-Deoxyaustrocortilutein (**169**)	*Altenaria tenuissima* DFFSCS013	-Marine sediment.	-	[63]
Altersolanol B (or dactylarin; **170**)	*Altenaria tenuissima* DFFSCS013	-Marine sediment.	-	[63]
	*Altenaria* sp. ZJ9-6B	-Fruits of a mangrove tree *Aegiceras corniculatum.*	-	[61]
	*Altenaria* sp. ZJ-2008003	-Soft coral *Sarcophyton* sp.	-	[49]
	*Stemphylium* sp. 33231	-Mangrove tree *Bruguiera sexangula* var. *rhynchopetala.*	-Antibacterial activity.	[50]
	*S. lycopersici*	-Inner tissue of gorgonian soft coral *Dichotella gammacea.*	-Cytotoxic activity.	[51]
	*Sporendonema casei* HDN16-802	-Marine sediment.	-Anti-*Mycobacterium tuberculosis* activity.	[109]
Altersolanol C (**171**)	*Altenaria* sp. ZJ-2008003	-Soft coral *Sarcophyton* sp.	-Antibacterial and cytotoxic activities.	[49]
	*Stemphylium* sp. 33231	-Mangrove tree *B. sexangula* var. *rhynchopetala.*	-	[50]
Altersolanol A (**172**)	*Stemphylium* sp. 33231	-Mangrove tree *B. sexangula* var. *rhynchopetala.*	-Antibacterial activity.	[50]
	*S. lycopersici*	-Inner tissue of gorgonian soft coral *Dichotella gammacea.*	-Cytotoxic activity.	[51]
	*Xylaria* sp. 2508	-Mangrove plant.	-	[119]
Auxarthrol C (**173**)	*Stemphylium* sp. 33231	-Mangrove tree *Bruguiera sexangula* var. *rhynchopetala*	-Antibacterial activity.	[50]
	*S. lycopersici*	-Inner tissue of gorgonian soft coral *Dichotella gammacea.*	-	[51]
2-*O*-Acetylaltersolanol B (**174**)	*Stemphylium* sp. 33231	-Mangrove tree *B. sexangula* var. *rhynchopetala.*	-Antibacterial activity.	[50]
Lentisone (**175**)	*Trichoderma* sp. (H-1)	-Sea cucumber.	-Antibacterial and antiangiogenic activities.	[43]
SZ-685C (known as 1403C; **176**)	*Halorosellinia* sp. (no. 1403)	-Mangrove plant.	-Anti-proliferative activity.	[108]
	-	-	-Cytotoxic activity and induction of cell apoptosis.	[132,133]
(2*R*,3*S*)-7-Ethyl-1,2,3,4-tetrahydro-2,3,8-trihydroxy-6-methoxy-3-methyl-9,10-anthracenedione (**177**)	*Phomopsis* sp. PSU-MA214	-Leaves of a mangrove tree *Rhizophora apiculata* Griff. Ex T. Anderson.	-Antibacterial and cytotoxic activities.	[52]
Auxarthrol D (**178**)	*Sporendonema casei* HDN16-802	-Marine sediment.	-Antibacterial and cytotoxic activities.	[109]
Auxarthrol G (**179**)	*S. casei* HDN16-802	-Marine sediment.	-Antibacterial and anticoagulant activities.	[109]
4-Dehydroxyaltersolanol A (**180**)	*S. casei* HDN16-802	-Marine sediment.	-Antibacterial and anticoagulant activities.	[109]
Aspetritone A (**181**)	*Aspergillus tritici* SP2-8-1	-Soft coral *Galaxea fascicularis.*	-Antibacterial and cytotoxic activities.	[28]
Bostrycin (**182**)	*Aspergillus* sp. strain 05F16	-Unidentified marine alga.	-	[110]
	*Fusarium* sp. PSU-F14 and PSU-F135	-Gorgonian sea fan (*Annella* sp.)	-Antimalarial and cytotoxic activities.	[59]
	*Nigrospora* sp. (strain no. 1403)	-Decayed wood of *Kandelia candel* (L.) Druce.	-Antibacterial, antifungal, and cytotoxic activities.	[112]
	*Nigrospora* sp. ZJ-2010006	-Unidentified sea anemone.	-Antibacterial and cytotoxic activities.	[68]
	*Xylaria* sp. 2508	-Mangrove plant.	-	[119]
	Strain no. 1403	-Mangrove plant.	-Anti-yeast activity and induction of cell apoptosis.	[111]
	-	-	-Cytotoxic activity.	[134]
Nigrosporin A (**183**)	*Fusarium* sp. PSU-F14 and PSU-F135	Gorgonian sea fan (*Annella* sp.).	-	[59]
Nigrosporin B (**184**)	*Fusarium* sp. PSU-F14 and PSU-F135	-Gorgonian sea fan (*Annella* sp.)	-Antimalarial and anti-*Mycobacterium tuberculosis*, and cytotoxic activities.	[59,130]
	*Nigrospora* sp. ZJ-2010006	-Unidentified sea anemone.	-Antibacterial and cytotoxic activities.	[68]
Fusarnaphthoquinone C (**185**)	*Fusarium* sp. PSU-F14 and PSU-F135	-Gorgonian sea fan (*Annella* sp.)	-	[59]
4-Deoxybostrycin (**186**)	*Nigrospora* sp. (strain no. 1403)	-Decayed wood of *Kandelia candel* (L.) Druce.	-Antibacterial, antifungal, and cytotoxic activities.	[112]
	-	-	-Anti-tumor activity.	[134]
	-	-	-Anti-*Mycobacterium tuberculosis* activity.	[130]
	*Nigrospora* sp. ZJ-2010006	-Unidentified sea anemone.	-Antibacterial activity.	[68]
	*Xylaria* sp. 2508	-Mangrove plant.	-	[119]
10-Deoxybostrycin (**187**)	*Nigrospora* sp. ZJ-2010006	-Unidentified sea anemone.	-Antibacterial and cytotoxic activities.	[68]
	*Nigrospora* sp. ZJ-2010006	-Inner tissue of the zoathid *Palythoa haddoni* (GX-WZ-20100026).	-	[69]
Hydroxybostrycin (**188**)	*Altenaria* sp. (SK11)	-Root of mangrove tree *Excoecaria agallocha.*	-	[62]
1403P-3 (**189**)	*Halorosellinia* sp. (no. 1403)	-Mangrove plant.	-Apoptosis in cancer cells.	[113]
Aspergiolide A (**190**)	*A. glaucus* HB1-19	-Marine sediment.	-Cytotoxic activity.	[114]
Aspergiolide B (**191**)	*A. glaucus* HB1-19	-Marine sediment.	-Cytotoxic activity.	[22]
Aspergiolide C (**192**)	*A. glaucus* HB1-19	-Marine sediment.	-Inhibition of receptor tyrosine kinases and anti-parasite activities.	[115]
Aspergiolide D (**193**)	*A. glaucus* HB1-19	-Marine sediment.	-Inhibition of receptor tyrosine kinases and anti-parasite activity.	[115]
Variecolortin A (**194**)	*Eurotium* sp. SCSIO F452	-Marine sediment.	-DPPH^•^ radicals scavenging activity.	[116]
Variecolortin B (**195**)	*Eurotium* sp. SCSIO F452	-Marine sediment.	-Cytotoxic activity.	[116]
Variecolortin C (**196**)	*Eurotium* sp. SCSIO F452	-Marine sediment.	-Cytotoxic activity.	[116]
Tetrahydrobostrycin (**197**)	*Aspergillus* sp. strain 05F16	-Unidentified marine alga.	-Antibacterial activity.	[110]
	*Altenaria* sp. (SK11)	-Root of mangrove tree *Excoecaria agallocha.*	-	[62]
1-Deoxytetrahydrobostrycin (**198**)	*Aspergillus* sp. strain 05F16	-Unidentified marine alga.	-Antibacterial activity.	[110]
8-Hydroxyconiothyrinone B (**199**)	*Talaromyces islandicus* EN-501	-Inner tissue of marine red alga *Laurencia okamurai.*	-Antibacterial, DPPH^•^, and ABTS^+•^ radicals scavenging activities.	[117]
8,11-Dihydroxyconiothyrinone B (**200**)	*T. islandicus* EN-501	-Inner tissue of marine red alga *L. okamurai.*	-Antibacterial, cytotoxic, DPPH^•^, and ABTS^+•^ radicals scavenging activities.	[117]
4*R*,8-Dihydroxyconiothyrinone B (**201**)	*T. islandicus* EN-501	-Inner tissue of marine red alga *L. okamurai.*	-Antibacterial, DPPH^•^, and ABTS^+•^ radicals scavenging activities.	[117]
4*S*,8-Dihydroxyconiothyrinone B (**202**)	*T. islandicus* EN-501	-Inner tissue of marine red alga *L. okamurai.*	-Antibacterial, DPPH^•^, and ABTS^+•^ radicals scavenging activities.	[117]
4*S*,8-Dihydroxy-10-*O*-methyldendroyl E (**203**)	*T. islandicus* EN-501	-Inner tissue of marine red alga *L. okamurai.*	-Antibacterial, DPPH^•^, and ABTS^+•^ radicals scavenging activities.	[117]
Fusaquinon A (**204**)	*Fusarium* sp. (no. ZH-210)	-Mangrove sediment.	-Cytotoxic activity.	[118]
Fusaquinon B (**205**)	*Fusarium* sp. (no. ZH-210)	-Mangrove sediment.	-Cytotoxic activity.	[118]
Fusaquinon C (**206**)	*Fusarium* sp. (no. ZH-210)	-Mangrove sediment.	-Cytotoxic activity.	[118]
Fusaranthraquinone (**207**)	*Fusarium* sp. PSU-F14 and PSU-F135	-Gorgonian sea fan (*Annella* sp.)	-	[59]
9α-Hydroxydihydrodesoxybostrycin (**208**)	*Altenaria* sp. (SK11)	-Root of mangrove tree *Excoecaria agallocha.*	-	[62]
	*Fusarium* sp. PSU-F14 and PSU-F135	-Gorgonian sea fan (*Annella* sp.)	-Antibacterial and cytotoxic activities.	[59]
	*Nigrospora* sp. ZJ-2010006	-Unidentified sea anemone.	-Antibacterial activity.	[68]
9α-Hydroxyhalorosellinia A (**209**)	*Fusarium* sp. PSU-F14 and PSU-F135	-Gorgonian sea fan (*Annella* sp.)	-Anti-leismanial, antimalarial, anti-*Mycobacterium tuberculosis*, and cytotoxic activities.	[59]
	*Nigrospora* sp. ZJ-2010006	-Unidentified sea anemone.	-Antibacterial and cytotoxic activities.	[68]
4a-*epi*-9α-Methoxydihydrodeoxybostrycin (**210**)	*Nigrospora* sp. ZJ-2010006	-Unidentified sea anemone.	-	[68]
	*Nigrospora* sp. ZJ-2010006	-Inner tissue of the zoathid *Palythoa haddoni* (GX-WZ-20100026).	-	[69]
Dihydroaltersolanol A (**211**)	*Altenaria tenuissima* DFFSCS013	-Marine sediment.	-	[63]
	*Altenaria* sp. ZJ-2008003	-Soft coral *Sarcophyton* sp.	-	[49]
	*Stemphylium* sp. 33231	-Mangrove tree *Bruguiera sexangula* var. *rhynchopetala.*	-	[50]
Altersolanol L (**212**)	*A. tenuissima* DFFSCS013	-Marine sediment.	-	[63]
	*Altenaria* sp. ZJ-2008003	-Soft coral *Sarcophyton* sp.	-	[49]
	*Stemphylium* sp. 33231	-Mangrove tree *B. sexangula* var. *rhynchopetala.*	-	[50]
	*Phoma* sp. L28	-Mangrove plant *Myoporum bontioides* A. Gray.	-Anti-fungal activity.	[67]
Ampelanol (**213**)	*Altenaria tenuissima* DFFSCS013	-Marine sediment.	-	[63]
	*Altenaria* sp. ZJ-2008003	-Soft coral *Sarcophyton* sp.	-	[49]
	*Stemphylium* sp. 33231	-Mangrove tree *Bruguiera sexangula* var. *rhynchopetala.*	-	[50]
	*S. lycopersici*	-Inner tissue of gorgonian soft coral *Dichotella gammacea.*	-	[51]
	*Phoma* sp. L28	-Mangrove plant *Myoporum bontioides* A. Gray.	-	[67]
	*Phomopsis* sp. PSU-MA214	-Leaves of a mangrove tree *Rhizophora apiculata* Griff. Ex T. Anderson.	-	[52]
Tetrahydroaltersolanol B (**214**)	*Altenaria* sp. ZJ9-6B	-Fruits of a mangrove tree *Aegiceras corniculatum.*	-	[61]
	*Altenaria* sp. ZJ-2008003	-Soft coral *Sarcophyton* sp.	-	[49]
	*Stemphylium* sp. 33231	-Mangrove tree *Bruguiera sexangula* var. *rhynchopetala.*	-Antibacterial activity.	[50]
	*Phoma* sp. L28	-Mangrove plant *Myoporum bontioides* A. Gray.	-Antifungal activity.	[67]
	*Phomopsis* sp. PSU-MA214	-Leaves of a mangrove tree *R. apiculata* Griff. Ex T. Anderson.	-	[52]
Halorosellinia A (**215**)	*Altenaria* sp. (SK11)	-Root of mangrove tree *Excoecaria agallocha.*	-	[62]
	*Halorosellinia* sp. (no. 1403)	-Mangrove plant.	-	[58]
Tetrahydroaltersolanol C (**216**)	*Altenaria* sp. ZJ-2008003	-Soft coral *Sarcophyton* sp.	-	[49]
	-	-	-Antiviral activity.	[107]
	*Phomopsis* sp. PSU-MA214	-Leaves of a mangrove tree *Rhizophora apiculata* Griff. Ex T. Anderson.	-	[52]
Tetrahydroaltersolanol D (**217**)	*Altenaria* sp. ZJ-2008003	-Soft coral *Sarcophyton* sp.	-	[49]
Tetrahydroaltersolanol E (**218**)	*Altenaria* sp. ZJ-2008003	-Soft coral *Sarcophyton* sp.	-	[49]
Tetrahydroaltersolanol F (**219**)	*Altenaria* sp. ZJ-2008003	-Soft coral *Sarcophyton* sp.	-	[49]
2-*O*-acetylaltersolanol L (**220**)	*Stemphylium* sp. 33231	-Mangrove tree *Bruguiera sexangula* var. *rhynchopetala.*	-	[50]
Harzianumnone A (**221**)	*Trichoderma harzianum* (XS-20090075)	-Soft coral	-	[55]
Harzianumnone B (**222**)	*T. harzianum* (XS-20090075)	-Soft coral.	-	[55]
Coniothyrinone A (**223**)	*Trichoderma* sp. (H-1)	-Sea cucumber.	-Antibacterial and antiangiogenic activities.	[43]
Xlyanthraquinone (**224**)	*Xylaria* sp. 2508	-Mangrove plant.	-	[119]
Auxarthrol E (**225**)	*Sporendonema casei* HDN16-802	-Marine sediment.	-Antibacterial activity.	[109]
Auxarthrol F (**226**)	*S. casei* HDN16-802	-Marine sediment.	-Antibacterial and cytotoxic activities.	[109]
Auxarthrol H (**227**)	*Sporendonema casei* HDN16-802	-Marine sediment.	-Antibacterial activity.	[109]
Asperflavin (**228**)	*A. glaucus* HB1-19	-Marine sediment.	-	[22]
	*Eurotium repens*	-Marine sponge *Suberites domuncula.*	-Cytotoxicity against sex cells.	[24]
	*E. rubrum*	-Inner tissue of mangrove plant *Hibiscus tiliaceus.*	-DPPH^•^ radicals scavenging activity.	[106]
	*Eurotium cristatum* EN-220	-Marine brown alga *Sargassum thunbergii.*	-Antibacterial activity.	[95]
Isoasperflavin (**229**)	*A. glaucus* HB1-19	-Marine sediment.	-	[22]
3,4-Dihydro-3,9-dihydroxy-6,8-dimethoxy-3-methylanthracen-1(2*H*)-one (**230**)	*A. wentii* EN-48	-Brown alga *Sargassum* sp.	-	[30]
Eurorubrin (**231**)	*Eurotium rubrum*	-Inner tissue of mangrove plant *H. tiliaceus.*	-DPPH^•^ radicals scavenging activity.	[106]
	*E. cristatum* EN-220	-Marine brown alga *Sargassum thunbergii.*	-Antibacterial activity and cytotoxicity against Brine Shrimp.	[95]
Asperflavin ribofuranoside (**232**)	*E. cristatum* EN-220	-Marine brown alga *Sargassum thunbergii.*	-	[95]
Anthrininone A (**233**)	*Altenaria tenuissima* DFFSCS013	-Marine sediment.	-Induction of intracellular calcium flux in HEK293 cells and inhibition of indoleamine 2,3-dioxygenase activity.	[63]
Scorpinone (**234**)	*Amorosia littoralis*	-Inertial sediment.	-	[120]
	*Bispora*-like tropical fungus	-Inertial sediment.	-	[121]
Bostrycoidin (**235**)	-	-	-	[120]
8-*O*-Methylbostrycoidin (**236**)	*A. terreus* (no. GX7-3B)	-Mangrove *Bruguiera gymnoihiza* (Linn.) Savigny.	-Anti-acetylcholinesterase activity.	[123]
6,6′-*Oxybis*(1,3,8-trihydroxy-2-((*S*)-1-methoxyhexyl)anthracene-9,10-dione) (**237**)	*A. versicolor*	-The inner tissue of an unidentified marine clam.	-Antibacterial activity.	[88]
6,6′-*Oxybis*(1,3,8-trihydroxy-2-((*S*)-1-hydroxyhexyl)anthracene-9,10-dione) (**238**)	*A. versicolor*	-The inner tissue of an unidentified marine clam.	-Antibacterial activity.	[88]
2,2′-*bis*-(7-methyl-1,4,5-trihydroxy-anthracene-9,10-dione) (**239**)	*Talaromyces stipitatus* KUFA0207	-Marine sponge *Stylissa flabelliformis.*	-	[33]
Alterporriol K (**240**)	*Altenaria* sp. ZJ9-6B	-Mangrove tree *Aegiceras corniculatum* fruits.	-Cytotoxic activity.	[61]
Alterporriol L (**241**)	*Altenaria* sp. ZJ9-6B	-Mangrove tree *A. corniculatum* fruits.	-Cytotoxic activity.	[61,135]
Alterporriol M (**242**)	*Altenaria* sp. ZJ9-6B	-Mangrove tree *A. corniculatum* fruits.	-	[61]
Alterporriol S (**243**)	*Altenaria* sp. (SK11)	-Root of mangrove tree *Excoecaria agallocha.*	-Anti-*Mycobacterium tuberculosis* activity.	[62]
(+)-a*S*-Alterporriol C (**244**)	*Altenaria* sp. (SK11)	-Root of mangrove tree *E. agallocha.*	-Anti-*Mycobacterium tuberculosis* activity.	[62]
Alterporriol C (**245**)	*Altenaria* sp. ZJ-2008003	-Soft coral reef *Sarcophyton* sp.	-Antibacterial and cytotoxic activities.	[49]
	*Stemphylium* sp. 33231	-Mangrove tree *Bruguiera sexangula* var. *rhynchopetala.*	-Antibacterial activity.	[50]
Alterporriol N (**246**)	*Altenaria* sp. ZJ-2008003	-Soft coral *Sarcophyton* sp.	-	[49]
	*Stemphylium* sp. 33231	-Mangrove tree *B. sexangula* var. *rhynchopetala.*	-	[50]
	*Stemphulium* sp. FJJ006	-Unidentified sponge.	-Anti-inflammatory activity.	[124]
Alterporriol O (**247**)	*Altenaria* sp. ZJ-2008003	-Soft coral *Sarcophyton* sp.	-	[49]
Alterporriol P (**248**)	*Altenaria* sp. ZJ-2008003	-Soft coral *Sarcophyton* sp.	-Cytotoxic activity.	[49]
Alterporriol Q (**249**)	*Altenaria* sp. ZJ-2008003	-Soft coral *Sarcophyton* sp.	-	[49]
	*Stemphylium* sp. 33231	-Mangrove tree *B. sexangula* var. *rhynchopetala.*	-	[50]
Alterporriol R (**250**)	*Altenaria* sp. ZJ-2008003	-Soft coral *Sarcophyton* sp.	-	[49]
	*Stemphylium* sp. 33231	-Mangrove tree *B. sexangula* var. *rhynchopetala.*	-	[50]
Nigrodiquinone A (**251**)	*Nigrospora* sp. ZJ-2010006	-Inner tissue of the zoathid *Palythoa haddoni* (GX-WZ-20100026).	-	[69]
Cytoskyrin A (**252**)	*Curvularia lunata*	-Marine sponge *Niphates olemda.*	-Antibacterial activity.	[64]
Alterporriol A (**253**)	*Stemphylium* sp. 33231	-Mangrove tree *B. sexangula* var. *rhynchopetala.*	-	[50]
Alterporriol B (**254**)	*Stemphylium* sp. 33231	-Mangrove tree *B. sexangula* var. *rhynchopetala.*	-Antibacterial activity.	[50]
Alterporriol D (**255**)	*Stemphylium* sp. 33231	-Mangrove tree *B. sexangula* var. *rhynchopetala.*	-Antibacterial activity.	[50]
Alterporriol E (**256**)	*Stemphylium* sp. 33231	-Mangrove tree *B. sexangula* var. *rhynchopetala.*	-Antibacterial activity.	[50]
Alterporriol T (**257**)	*Stemphylium* sp. 33231	-Mangrove tree *B. sexangula* var. *rhynchopetala.*	-	[50]
Alterporriol U (**258**)	*Stemphylium* sp. 33231	-Mangrove tree *B. sexangula* var. *rhynchopetala.*	-Antibacterial activity.	[50]
Alterporriol V (**259**)	*Stemphylium* sp. 33231	-Mangrove tree *B. sexangula* var. *rhynchopetala.*	-Antibacterial activity.	[50]
Alterporriol W (**260**)	*Stemphylium* sp. 33231	-Mangrove tree *B. sexangula* var. *rhynchopetala.*	-	[50]
Alterporriol Y (**261**)	*S. lycopersici*	-Inner tissue of gorgonian soft coral *Dichotella gammacea.*	-	[51]
Alterporriol F (**262**)	*Stemphulium* sp. FJJ006	-Unidentified sponge.	-Anti-inflammatory activity.	[124]
Alterporriol G (**263**)	*Stemphulium* sp. FJJ006	-Unidentified sponge.	-Anti-inflammatory activity.	[124]
Alterporriol Z_1_ (**264**)	*Stemphulium* sp. FJJ006	-Unidentified sponge.	-Anti-inflammatory activity.	[124]
Alterporriol Z_2_ (**265**)	*Stemphulium* sp. FJJ006	-Unidentified sponge.	-Anti-inflammatory activity.	[124]
Alterporriol Z_3_ (**266**)	*Stemphulium* sp. FJJ006	-Unidentified sponge.	-	[124]
Rubellin A (**267**)	Strain F-F-3C	-Unidentified marine red alga.	-Antibacterial activity.	[44]
14-Acetoxyrubellin A (**268**)	Strain F-F-3C	-Unidentified marine red alga.	-Antibacterial and antifungal activities.	[44]
14-Acetoxyrubellin C (**269**)	Strain F-F-3C	-Unidentified marine red alga.	-Antibacterial activity.	[44]
Physcion-10,10′-bianthrone (**270**)	*A. glaucus* HB1-19	-Deep-sea sediment.	-	[22]
	*A. wentii* EN-48	-Brown alga *Sargassum* sp.	-	[30]
*trans*-Emodin-physcion bianthrone (**271**)	*A. glaucus* HB1-19	-Deep-sea sediment.	-Cytotoxic activity.	[22]
*cis*-Emodin-physcion bianthrone (**272**)	*A. glaucus* HB1-19	-Deep-sea sediment.	-Cytotoxic activity.	[22]
Atropisomer of 8,8′-dihydroxy-1,1′,3,3′-tetramethoxy-6,6′-dimethyl-10,10-bianthrone (**273**)	*A. wentii* EN-48	-Brown alga *Sargassum* sp.	-	[30]
Atropisomer of 8,8′-dihydroxy-1,1′,3,3′-tetramethoxy-6,6′-dimethyl-10,10-bianthrone (**274**)	*A. wentii* EN-48	-Brown alga *Sargassum* sp.	-	[30]
Allianthrone A (**275**)	*A. alliaceus*	-Marine algae.	-Cytotoxic activity.	[77]
Allianthrone B (**276**)	*A. alliaceus*	-Marine algae.	-Cytotoxic activity.	[77]
Allianthrone C (**277**)	*A. alliaceus*	-Marine algae.	-Cytotoxic activity.	[77]
(±)-Eurotone A (**278**)	*Eurotium* sp. SCSIO F452	-Marine sediment.	-	[70]
JBIR-97/98 (**279**)	*Engyodontium album*	-Marine sponge *Cacospinga scalaris.*	-Antibacterial, antifungal, and cytotoxic activities.	[125]
JBIR-99 (**280**)	*E. album*	-Marine sponge *C. scalaris.*	-Antibacterial, antifungal, and cytotoxic activities.	[125]
Engyodontochone A (**281**)	*E. album*	-Marine sponge *C. scalaris.*	-Antibacterial, antifungal, and cytotoxic activity.	[125]
Engyodontochone B (**282**)	*E. album*	-Marine sponge *C. scalaris.*	-Antibacterial, antifungal, and cytotoxic activities.	[125]
Engyodontochone C (**283**)	*E. album*	-Marine sponge *C. scalaris.*	-Antibacterial and cytotoxic activities.	[125]
Engyodontochone D (**284**)	*E. album*	-Marine sponge *C. scalaris.*	-	[125]
Engyodontochone E (**285**)	*E. album*	-Marine sponge *C. scalaris.*	-Antibacterial activity.	[125]
Engyodontochone F (**286**)	*Engyodontium album*	-Marine sponge *C. scalaris.*	-Antibacterial activity.	[125]
Acremonidin A (**287**)	*Acremonium camptosporum*	-Marine sponge *Aplysina fulva.*	-Antibacterial and cytotoxic activities.	[126]
	Unidentified fungus of the order Hypocreales (MSX 17022)	-	-20S proteasome inhibitory activity.	[127]
Acremonidin B (**288**)	*A. camptosporum*	-Marine sponge *A. fulva.*	-Antibacterial and cytotoxic activities.	[126]
Acremonidin C (**289**)	*A. camptosporum*	-Marine sponge *A. fulva.*	-Antibacterial and cytotoxic activities.	[126]
	Unidentified fungus of the order Hypocreales (MSX 17022)	-	-20S proteasome inhibitory activity.	[127]
Acremonidin G (**290**)	*A.camptosporum*	-Marine sponge *A. fulva.*	-Antibacterial and cytotoxic activities.	[126]
Acremoxanthone A (**291**)	*A. camptosporum*	-Marine sponge *A. fulva.*	-Antibacterial and cytotoxic activities.	[126]
Acremoxanthone B (**292**)	*A. camptosporum*	-Marine sponge *A. fulva.*	-Antibacterial and cytotoxic activities.	[126]
Acremoxanthone D (**293**)	*A. camptosporum*	-Marine sponge *A. fulva.*	-Antibacterial and cytotoxic activities.	[126]
	Unidentified fungus of the order Hypocreales (MSX 17022)	-	-20S proteasome inhibitory activity.	[127]
Acremoxanthone F (**294**)	*A. camptosporum*	-Marine sponge *A. fulva.*	-Antibacterial and cytotoxic activities.	[126]
Acremoxanthone G (**295**)	*A. camptosporum*	-Marine sponge *A. fulva.*	-Antibacterial and cytotoxic activities.	[126]
Acremoxanthone C (**296**)	Unidentified fungus of the order Hypocreales (MSX 17022)	-	-20S proteasome inhibitory activity.	[127]

## Data Availability

Data sharing is not applicable.

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
