# Peer review of "Anthraquinones and Their Analogues from Marine-Derived Fungi: Chemistry and Biological Activities"

_marinedrugs, 2022, doi:10.3390/md20080474_

Round 1

Reviewer 1 Report

This is a very large and generally well documented and presented review. For a reviewer it is not possible to validate every structure and claim, so my review focuses on assessing the over-all quality and clarity, value to readership, and also any obvious typos or errors. Firstly, I would like to congratulate the authors on such a substantive review, which I believe will be well received by researchers across multiple dimensions of chemistry. I recommend publication subject to some minor corrections (see below).

Line 110               “anthrarquinoid”

Line 199               “Sout Pole”

Fig 6                     the inclusion of the structure for the acetonide derivative 77a seems unnecessary, as it could be misinterpreted by readers as being a natural product. For example, at line 656 the acetonide of 175 bdid not warrant a structure diagram.

Line 507               “Insecticidal activity guided… yielded 121” Does this mean 121 was the insecticide? Having introduced the concept of activity guided it would be good to give some closure.

Line 528               “streogenic”

Line 532               “acid-containg” and “chainsHPL”

Line 534               “amino acids of the amide side chains” is a redundant statement “amino acid residues” would be better.

Line 536               “aminde”

Line 539               “butylaminolate” doesn't make much sense chemically? Wouldn't “(R)-5-methylcarboxylate-2-pyrrolidine” or even more simply “a substituted pyrrolidone” be better?

Fig 10                   my understanding is that NRPS BGCs are more likely to use L-Glu as the actual substrate, which is converted to D-Glu in situ via an epimerase module, prior to cyclization and release from the PKS-NRPS. As 134 and 135 are diastereomers of known absolute configuration they should include appropriate stereo descriptors at C-13 (ie not a wavey line which implies a mixture of epimers).

Fig 11                   The structure representation of 152 does not reveal the configuration of the Ile secondary methyl – ie L-Ile vs allo-L-Ile. Was this even determined. In my experience all to often natural products chemists ignore the allo option.

Fig 12                   154 is represented as the Na salt, while 155 and 156 are presented as sulfate anions, lacking counter ions. Is this deliberate? While it's a relatively minor issue, by presenting them differently it does suggest a distinction that may not really apply. As an aside Ref 50 does not provide any structure proof for 154, certainly not a preferred counter ion, while ref 36 notes 155 and 156 as the protonated forms (SO3H) – not salts, and without counterions?

Line 617               Structure 160 needs correcting (OH floating)

Lines 620-2          Not entirely sure the header “seco-anthraquinones”, of the opening sentence is accurate. These so named “seco-anthraquinones” are also known as “depsidones” and the biosynthesis can proceed via a Baeyer-Villiger-like enzymatic oxidation direct from the anthraquinone to the ring-expanded lactone – without proceed via a ring opened (ie seco) intermediate. In such a biogenesis the seco form arise via hydrolysis of the depsidone lactone. At the very least 161–164 would be better characterised as depsidones, and as 165–168 are not ring opened it seems odd to characterise them as seco, rather than reduced depsidones.

                            As an interesting aside, it has recently become clear that depsidones exist as atropisomers, a hidden layer of chirality that has been overlooked for decades – and which clearly impacts on biological activity. For example, see RSC Advances, 2021, 11, 29661-29667 (spiromastimelleins) and J. Nat. Prod. 2014, 77 (4), 1021-1030 (spiromastixones). These seco/depsidones seem to have been overlooked in this current review.

Line 702               The structure diagrams for the semi synthetic acetates 183a and 187a are superfluous.

Fig 20                   3-H in the structure diagram for 230 is redundant

Line 861               “mangrave”

Line 910               “anthraquines”

Fig 24                   Structure 253 is indecipherable.

Line 2048             formatting – blue coloured title and journal details?

Line 2086             ref 66 title is incorrect.

Author Response

Reply to Reviewers

Reviewer#1: This is a very large and generally well documented and presented review. For a reviewer it is not possible to validate every structure and claim, so my review focuses on assessing the over-all quality and clarity, value to readership, and also any obvious typos or errors. Firstly, I would like to congratulate the authors on such a substantive review, which I believe will be well received by researchers across multiple dimensions of chemistry. I recommend publication subject to some minor corrections (see below).

Reply:

We wish to sincerely thank reviewer #1 for his/her appreciation of the manuscript as well as for a thorough reading of the manuscript and for giving  very constructive comments. Reviewer#1’s comments will no doubt improve a readability of this manuscript.

 Before we start to reply reviewer #1, we wish to clarify that the numbers of the compounds have been altered as we have just discovered that compounds 1 (chrysophanic acid) and 30 (chrysophanol) are the same compound but was reported with different names by different authors. Therefore, we had to rearrange the whole numbers of the compounds starting from compound 30 forward. So, we will use the former numbers to reply to reviewer#1 but will put the new number in a parenthesis.

Reviewer#1: Line 110: “anthrarquinoid”

Reply: The typo was corrected

Reviewer#1: Line 199: “Sout Pole”

Reply: The typo was corrected

Reviewer#1: Fig 6: the inclusion of the structure for the acetonide derivative 77a seems unnecessary, as it could be misinterpreted by readers as being a natural product. For example, at line 656 the acetonide of 175 bdid not warrant a structure diagram.

Reply: As reviewer#1 can see, we used the letter “a” after the number of the compounds to denote their derivatized products, such as acetonide or acetate. In the case of “77a (76a)” it was used to discuss how to determine the absolute configurations of the stereogenic carbons of the vic- diol. Therefore, we do not think the structure of 77a (now 76a), would lead to any misunderstanding as the explanation is very clear in the body text. On the contrary, the structure of 77a (76a) gives an added value to the readers who might have not known about this simple method of determination of the absolute configurations of vic-diol by preparation of the acetonide.

Reviewer#1: Line 507: Insecticidal activity guided… yielded 121” Does this mean 121 was the insecticide? Having introduced the concept of activity guided it would be good to give some closure.

Reply: In this section we only report the compounds (including 121) isolated from the extract by using insecticidal activity-guided fractionation. However, the insecticidal activity of these compounds (100, 101 and 121) are discussed in Section 4: Biological Activities, in the subheading 4.13. Insecticidal activity.

Reviewer#1: Line 528: “streogenic”

Reply: The typo was corrected.

Reviewer#1: Line 532 “acid-containg” and “chainsHPL”

Reply: The typo was corrected.

Reviewer#1: Line 534 “amino acids of the amide side chains” is a redundant statement

Reply: We thank reviewer #1 for a nice suggestion. However, we prefer to maintain the term “amino acid-containing amide side chains” to stress that the amide side chain is formed by amino acid and not any other biogenic amines which can derive from amino acids

Reviewer#1: Line 536: “aminde”

Reply: The typo is corrected.

Reviewer#1: Line 539 “butylaminolate” doesn't make much sense chemically? Wouldn't “(R)-5-methylcarboxylate-2-pyrrolidine” or even more simply “a substituted pyrrolidone” be better?

Reply: We thank reviewer #1 for a good suggestion. In fact, the term “4,5-disubstituted butylaminolate side chains” is used in the original article that we used as a reference. In order to be precise, we changed to “1-hydroxy-2[(2R)-2-(methoxycarbonyl)-5-oxopyrrolidin-1-yl substituent”, which is a correct IUPAC name of the substituent.

Reviewer#1: Fig 10: my understanding is that NRPS BGCs are more likely to use L-Glu as the actual substrate, which is converted to D-Glu in situ via an epimerase module, prior to cyclization and release from the PKS-NRPS. As 134 and 135 are diastereomers of known absolute configuration they should include appropriate stereo descriptors at C-13 (ie not a wavey line which implies a mixture of epimers).

Reply: We have put a correct stereochemistry of C-13 for compounds 134 (133) and 135 (134).

Reviewer#1: Fig 11-The structure representation of 152 does not reveal the configuration of the Ile secondary methyl – ie L-Ile vs allo-L-Ile. Was this even determined. In my experience all to often natural products chemists ignore the allo option.

Reply: We understand Reviewer#1’s comment and we also share his rightful concern. However, this is a review article and we have to limit ourselves to the information available from the original article. According to Luo et al. (Ref. 46), they identified the amino acid residue as L-Val by Marfey’s method, which is a recognized method to identify amino acid residues in Natural Products Chemistry.

Reviewer#1: Fig 12: 154 is represented as the Na salt, while 155 and 156 are presented as sulfate anions, lacking counter ions. Is this deliberate? While it's a relatively minor issue, by presenting them differently it does suggest a distinction that may not really apply. As an aside Ref 50 does not provide any structure proof for 154, certainly not a preferred counter ion, while ref 36 notes 155 and 156 as the protonated forms (SO3H) – not salts, and without counterions?

Reply: We understand reviewer#1’s point of view. One of the reasons we present the structure of 154 (now 153) as a sodium salt was because we followed the report by Zhou et al (Ref. 50). Although, as reviewer#1 mentioned “As an aside Ref 50 does not provide any structure proof for 154”, we have cross-checked with the original article to which Zhou et al. have compared his data with those reported in this article, and we have found that in this article (Amal H. Aly et al. Bioactive metabolites from the endophytic fungus Ampelomyces sp. isolated from the medicinal plant Urospermum picroides. Phytochemistry 2008, 69 1716-1725. doi:10.1016/j.phytochem.2008.02.013.), the molecular formula of this compound was determined by HRES-TOF-MS which confirmed the presence of a sodium atom. As to the structures of 155 (now 154) and 156 (now 155), we have altered their structures to be an acid form (instead of a sulfate anion). This also was based on the results of the HRESIMS of both compounds which gave their accurate molecular formulas.

Reviewer#1: Line 617-Structure 160 needs correcting (OH floating)

Reply: done

Reviewer#1: Lines 620-2-Not entirely sure the header “seco-anthraquinones”, of the opening sentence is accurate. These so named “seco-anthraquinones” are also known as “depsidones” and the biosynthesis can proceed via a Baeyer-Villiger-like enzymatic oxidation direct from the anthraquinone to the ring-expanded lactone – without proceed via a ring opened (ie seco) intermediate. In such a biogenesis the seco form arise via hydrolysis of the depsidone lactone. At the very least 161–164 would be better characterised as depsidones, and as 165–168 are not ring opened it seems odd to characterise them as seco, rather than reduced depsidones.

Reply: We wish to thank reviewer #1 for raising the discussion about “seco-anthraquinones” and “depsidones”. However, we disagree with reviewer#1’s opinion for the following reasons. First, a crucial step in fungal seco-anthraquinone biosynthesis is a cleavage of the C-10 and C-4a bond which has long been proposed as a classic Baeyer–Villiger oxidation. Moreover, most recently, Qi et al. have confirmed that the reductase GedF is responsible for the reduction of the keto group at C-10 in questin (3) to a hydroxyl group, with the aid of NADPH. The C-10–C-4a bond of the resulting questin hydroquinone is subsequently cleaved by the atypical cofactor-free dioxygenase GedK, giving rise to desmethylsulochrin. For more information, please see https://doi.org/10.1021/jacs.1c07182.

On the other hand, Ibrahim et al., in their review article of the fungal depsidone compounds (https://doi.org/10.1016/j.fitote.2018.04.012), have defined them as two 2,4-dihydroxybenzoic acid rings linked together by ether and ester bonds, while anthraquinones are biosynthesized from acetyl CoA  unit (as a starter) with malonyl CoA (as extenders). Moreover, if we look at the structures of seco-anthraquinones in the manuscript, i. e. 161-168 (now, 160-167), we can see that all of them have a methyl (or oxidized methyl group) para to the carbonyl (either ketone or lactone), while the methyl group in depsidones is ortho to the lactone carbonyl as in orsellinic acid or beta-orsellinic acid (their precursors). Additionally, Du et al. (Ref. 21) have elegantly postulated the biogenesis pathways of seco-anthraquinone scaffold from the anthraquinone  emodin (2).

Reviewer#1: As an interesting aside, it has recently become clear that depsidones exist as atropisomers, a hidden layer of chirality that has been overlooked for decades – and which clearly impacts on biological activity. For example, see RSC Advances, 2021, 11, 29661-29667 (spiromastimelleins) and J. Nat. Prod. 2014, 77 (4), 1021-1030 (spiromastixones). These seco/depsidones seem to have been overlooked in this current review.

Reply: We wish to thank reviewer #1 for this valuable information. We do agree with reviewer #1 that many researchers have overlooked the existence of atropisomers in many classes of compounds. One of the corresponding authors have recently identified atropisomers of bis-naphthoquinones by NMR methods (please see: https://doi.org/10.1016/j.phytochem.2020.112575).

Reviewer#1: Line 702-The structure diagrams for the semi synthetic acetates 183a and 187a are superfluous.

Reply: We do not think the structures of 183a (now 182a) and 187a (now 186a) superfluous because both compounds showed strong cytotoxicity against A549 cancer cell lines. As reviewer #1 can see that there are more acetylable hydroxyl groups in both 183 and 187. However, only the hydroxyl group in that specific position was acetylated.

Reviewer#1: Fig 20-3-H in the structure diagram for 230 is redundant.

Reply: H-3 in the structure of 230 (now 229) was removed.

Reviewer#1: Line 861: “mangrave”

Reply: The typo was corrected.

Reviewer#1: Line “anthraquines”

Reply: The typo was corrected.

Reviewer#1: Fig 24: Structure 253 is indecipherable.

Reply: The structure of 253 was redrawn to be more clearly decipherable.

Reviewer#1: Line 2048-formatting – blue coloured title and journal details?

Reply: The blue colour was changed to black.

Reviewer#1: Line 2086-ref 66 title is incorrect.

Reply: We thank reviewer#1 for detecting this error. Now, the title of Ref. 66 was corrected.

Reviewer 2 Report

This is a good and well-written review paper titled “Anthraquinones and Analogues from Marine-Derived Fungi: Chemistry and Biological Activities”. Authors described 297 anthraquinones and their derivatives, summarized their biological properties, and proposed biogenesis of some anthraquinone derivatives. The chemical structures of anthraquinones and their derivatives have been well categorized by their structures. The reviewer recommends this review to accept for Marine Drugs after a minor revision:

Comments:
1)   This review mainly consists of structural diversity of anthraquinones from marine fungi and their biological activities. As a reviewer and as a reader, it is not easy to follow up the structures of anthraquinones and their biological activities, because in the structural section, authors deal only the structures of anthraquinones without describing their activities, on the other hand, in biological activity section, authors describe the bioactivities of anthraquinones without showing their structures. So I would like to recommend authors to properly put some explanations on the bioactivities of anthraquinones for some representative compounds in the structural section and the structures of some representative compounds in the bioactivity section.

2) In concluding remarks and future perspectives section, future perspectives are not described well. Authors need to add more description on future perspectives on marine fugal resources in drug discovery.

For minor comments,

please find the comments which are marked on the pdf version of manuscript file.

Author Response

Reply to Reviewer #2

Reviewer#2:

This is a good and well-written review paper titled “Anthraquinones and Analogues from Marine-Derived Fungi: Chemistry and Biological Activities”. Authors described 297 anthraquinones and their derivatives, summarized their biological properties, and proposed biogenesis of some anthraquinone derivatives. The chemical structures of anthraquinones and their derivatives have been well categorized by their structures. The reviewer recommends this review to accept for Marine Drugs after a minor revision:

Reply: We wish to thank reviewer #2 for his/her appreciation of our manuscript. We are very pleased to have high opinion of the expert in the field.

Reviewer#2:

Comments:

1) This review mainly consists of structural diversity of anthraquinones from marine fungi and their biological activities. As a reviewer and as a reader, it is not easy to follow up the structures of anthraquinones and their biological activities, because in the structural section, authors deal only the structures of anthraquinones without describing their activities, on the other hand, in biological activity section, authors describe the bioactivities of anthraquinones without showing their structures. So I would like to recommend authors to properly put some explanations on the bioactivities of anthraquinones for some representative compounds in the structural section and the structures of some representative compounds in the bioactivity section.

Reply: We understand reviewer#2’s concern about the organization of this manuscript. As we have published several review papers in the past 5 years (please see, for example: https://doi.org/10.1039/c8np00043c, https://doi.org/10.3390/md18060317, https://doi.org/10.3390/ molecules26061754, https://doi.org/10.3390/md19080410, https://doi.org/10.3390/md20010003,  https://doi.org/10.3390/molecules27072351), we have tried to improve our next manuscript in terms of logic and readability to make them more and more attractive to the readers

In this concrete review, we approached the subject with the chemistry part which is fundamental for the readers to know what the main scaffolds of anthraquinids are. We then expand to more complex structures and discussed the methods of determination of the absolute configurations of the side chains. In some cases, where biogenesis is fundamental, we also give some concrete examples. In the second part, we show the importance of this class of compounds in terms of biological activities. As reviewer#2 can see, there are a myriad of biological activities exhibited by members of this class of compounds. Of note is that antibacterial and cytotoxic activities are dominating while other activities were detected in less number of compounds. This reflects a fundamental lacuna of studying bioactive compounds as a majority of the laboratories usual perform these traditional in vitro techniques. Another important aspect is that some common compounds, which are isolated from most of marine-derived fungi seem to exhibit more activities than any other rare structures which were isolated in small quantity and in one or two laboratories. As reviewer#2 can see, many compounds did not exhibit bioactivity in the assay platforms available in some laboratories. However, it doesn’t mean that they do not have other bioactivities.  Therefore, mixing the discussion of the structures of the compounds with biological activities will add even more confusion and inaccuracy. In my opinion, the review of naturally occurring specialized metabolites should have a different approach from the review manuscript of “Medicinal Chemistry” where the objective is the biological/pharmacological activities of the scaffolds.

However, we also realize that some readers might get confused if they read only the text. That’s why we have elaborated the Table (Table.1) to summarize the compounds which exhibit biological activities.  

  1. In concluding remarks and future perspectives section, future perspectives are not described well. Authors need to add more description on future perspectives on marine fugal resources in drug discovery.

Reply:

We have expanded in the section of future perspectives.

Reviewer#2: For minor comments, please find the comments which are marked on the pdf version of manuscript file.

Reply: We wish to thank reviewer#2 for his/her time to overhaul the whole text and gives valuable suggestions.